# LIE GROUP-INDUCED DYNAMICS IN SCORE-BASED GENERATIVE MODELING

## ABSTRACT

We extend score-based generative modeling by incorporating Lie group actions on the data manifold into the denoising diffusion process. Our approach yields a Langevin dynamics whose infinitesimal transformations decompose as a direct sum of Lie algebra representations, enabling generative processes that align with the underlying symmetry properties of the data. Unlike equivariant models, which restrict the space of learnable functions by quotienting out group orbits, our method incorporates both global and local symmetries and can model *any* target distribution. Standard score-matching, which minimizes the Fisher divergence, emerges as a special case of our framework when the Lie group is the translation group in Euclidean space. We prove that our generalized generative processes arise as solutions to a new class of reverse-time stochastic differential equations (SDEs), introduced here for the first time. We validate our approach through experiments on diverse data types, demonstrating its effectiveness in real-world applications such as $SO(3)$-guided molecular conformer generation and modeling ligand-specific global $SE(3)$ transformations for molecular docking. We show that an appropriate choice of Lie group enhances learning efficiency by reducing the effective dimensionality of the trajectory space and enables the modeling of transitions between complex data distributions, lifting the requirement of a Gaussian prior. Additionally, we demonstrate the universality of our approach by deriving how it extends to flow matching techniques.

## 1 INTRODUCTION

Deep probabilistic generative modeling amounts to creating data from a known tractable prior distribution. Score-based models (Hyvärinen & Dayan, 2005; Sohl-Dickstein et al., 2015; Ho et al., 2020; Huang et al., 2021) achieve this by learning to reverse a corruption process of the data. The sampling process is realized by a Markov chain Monte Carlo (MCMC) using Langevin dynamics (Song et al., 2021; 2020b), producing a probabilistic trajectory $\{\mathbf{x}_t\}_{t=T,\dots,0}$ in the data space $X$ guided by the score function $\mathbf{s}(\mathbf{x}) = \nabla_{\mathbf{x}} \log p(\mathbf{x})$: for each infinitesimal step $\mathbf{x}_{t+1} \mapsto \mathbf{x}_t - \mathbf{s}(\mathbf{x}_t) + \text{noise}$, the score induces a small translation in $X$ (Figure 1a), and the noise prevents the dynamics from collapsing into local minima (Welling & Teh, 2011; Neal, 2010). The observation that translations are just one of many transformations of Euclidean space leads to the central question of the present work: *given a group $G$ acting on $X$, can we construct a score-based dynamics whose steps correspond to the infinitesimal transformation of $G$?*

We will prove that the answer is positive when $G$ is a Lie group, as these are suitable to describe local continuous transformations. Specifically, we construct a Langevin dynamics that decomposes as a direct sum of representations of the Lie algebra $\mathfrak{g}$ of $G$, where each step is driven by tangent vectors associated with elements of $\mathfrak{g}$ (Figure 1)c. This construction employs the formalism of Generalized Score Matching (GSM) (Lyu, 2009; Lin et al., 2016), whose aim is to estimate a probability density $p(\mathbf{x}) = \exp(-U(\mathbf{x}))/Z$ through its *generalized score function* $\mathcal{L} \log p(\mathbf{x})$, where $\mathcal{L}$ is a suitable linear operator. As Figure 1b suggest, a generalized score enables the dynamics to follow curved paths, a crucial feature for our scope since group orbits typically have nonzero curvature. In this work we will systematically develop the connection between the transformation-generating Lie group $G$ and the linear operator $\mathcal{L}$. We will also show that such generalized dynamics admits a stochastic differential equations (SDE) interpretation: the $G$-induced generative process satisfies a continuous-(reverse-) time SDE involving the generalized score function.

Figure 1: Standard score-matching (a) vs. generalized score-matching (b); (c) Trajectories for $G = T(2)$ (dark blue, standard score-matching), and $G = \mathrm{SO}(2) \times \mathbb{R}_+$ (dark orange). (d) Illustration of invariant, equivariant, standard and generalized score functions for a radial vector field in $X = \mathbb{R}^2$.

At this point, the attentive reader might wonder whether all we are achieving is to merely introduce multiple ways, according to the choice of $G$, to achieve the same goal of generating samples from a target distribution. So, what's the real benefit? This lies in the ability to tailor $G$ to the specific transformation properties of the data and the learning task: the score components can be designed to align with the true degrees of freedom of the data, enhancing interpretability, sampling steerability, and enabling dimensionality reduction by leveraging the data's inherent symmetries.

Figure 1d illustrates this in a simple but instructive case. Consider $X = \mathbb{R}^2$ and let the true score be of the form $\mathbf{s}(\mathbf{x}) = f(r, \theta)\hat{\mathbf{r}}$. This is neither invariant nor equivariant since $f$ depends on both $r, \theta$, thus it cannot be learned by an invariant/equivariant network. A standard score function neglects the problem's symmetry and is required to inefficiently (and likely inaccurately) learn the correct point-dependent linear combination of both Cartesian components of the score. A $\mathrm{SO}(2)$ generalized score function, however, directly learns the radial component $f(r, \theta)$, effectively reducing the problem to a 1-dimensional formulation, as the angular component may be omitted from the leaning procedure.

In short, *we propose an exact SDE-based diffusion framework that enables Lie group-guided curved dynamics while operating entirely in Euclidean space*, thus combining the advantages of curved dynamics with the theoretical and practical effectiveness of Euclidean diffusion. Our framework in unconstrained, unlike equivariant models, but still leverages the information of the group action on the data space. Ours is the first methods we are aware of that realizes **simulation-free** training of Lie group-like diffusion models, and the **first result of denoising score-matching result for general non-Abelian groups** (unlike De Bortoli et al. (2022) and Huang et al. (2022)).

More specifically, we summarize the main contributions of this manuscript as follows:

**Generalized score matching via Lie algebras:** We develop the mathematical foundations for extending score-based generative modeling to incorporate Lie group-induced dynamics. We spell out the conditions for a suitable $G$ (valid for any differentiable manifold $X$) and demonstrate that the dynamics decomposes as a linear combination of Lie algebra representations. We also show that standard score-matching is recovered as a specific case of our framework, corresponding to the group $G = T(n)$ of translations on $X = \mathbb{R}^n$.

**Exact solution of a novel class of SDEs:** We introduce a *new class of solvable* SDEs that govern symmetry-aware processes in Euclidean space, significantly expanding the range of processes that can be addressed using diffusion modeling techniques. Additionally, we show that our approach extends naturally to flow matching (see Appendix D).

**Dimensionality reduction, bridging non-trivial distributions and trajectory disentanglement:** Through extensive experiments[*], we demonstrate that: (1) our approach can estimate, regardless of the choice of $G$, any probability density (Sections 5(2,3,4d distributions) and 5)(QM9); (2) by appropriately selecting $G$ to align with the data structure, the learning process is significantly simplified, effectively reducing its dimensionality (Section 5(MNIST)) (3) our framework enables solutions to processes that are challenging or unfeasible with standard score matching, such as bridging between complex data-driven distributions (Section 5(MNIST) and 5(CrossDocked));

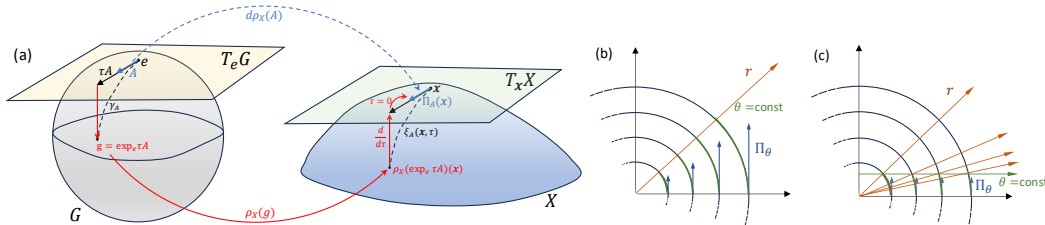

Figure 2: (a) Depiction of the fundamental vector field definition (1). Flow coordinates for a pair of commuting (b) and not-commuting ones vector fields (c).

## 2 DIFFUSION DYNAMICS THROUGH LIE ALGEBRAS

We start this section by setting up notation and review the connection between vector fields and Lie algebra actions on manifolds. A Lie group $G$ is a group that is also a finite-dimensional differentiable manifold, such that the group operations of multiplication $\cdot : G \times G \to G$ and inversion are $C^\infty$-functions[†]. A Lie algebra $\mathfrak{g}$ is a vector space equipped with an operation, the Lie bracket, $[,]$ : $\mathfrak{g} \times \mathfrak{g} \to \mathfrak{g}$, satisfying the Jacobi identity. Every Lie group gives rise to a Lie algebra as its tangent space at the identity, $\mathfrak{g} = T_e G$, and the Lie bracket is the commutator of tangent vectors, $[A, B] = AB - BA$. In this work, we are interested in how Lie groups and Lie algebras act on spaces. Given a manifold $X$, a (left) **group action** of $G$ on $X$ is an associative map $\rho_X : G \times X \to X$ such that $\rho_X(e) = \mathrm{Id}_X$. Fundamental concepts associated with a group action are the ones of orbits and stabilizers. The **orbit** of $\mathbf{x} \in X$ is the set of elements in $X$ which can be reached from $\mathbf{x}$ through the action of $G$, i.e., $G \cdot \mathbf{x} = \{\rho_X(g)(\mathbf{x}), \, g \in G\}$. The **stabilizer** subgroup of $G$ with respect to $\mathbf{x}$ is the set of group elements that fix $\mathbf{x}$, $G_x = \{g \in G | \rho_X(g)(\mathbf{x}) = \mathbf{x}\}$. The **action of a Lie algebra** on $X$, $\mathfrak{A} : \mathfrak{g} \to \mathrm{Vect}(X)$ is a Lie algebra homomorphism and maps elements of $\mathfrak{g}$ to vector fields on $X$ such that the map $\mathfrak{g} \times X \to TX, (A, \mathbf{x}) \mapsto \mathfrak{A}(A)(\mathbf{x})$ is smooth. Given $A \in \mathfrak{g}$ and a group action $\rho_X$, the **flow** on $X$ induced by $\rho_X$ is given by $\xi_A : X \times \mathbb{R} \to X, (\mathbf{x}, \tau) \to \rho_X(\exp(\tau A))(\mathbf{x})$, where the map $\exp : \mathfrak{g} \to G$ is defined by $\exp(A) = \gamma_A(1)$, where $\gamma_A : \mathbb{R} \to G$ is the unique one-parameter subgroup of $G$ whose tangent vector at the identity is $A$. The **infinitesimal action** of $\mathfrak{g}$ on $X$ is defined as the differential of the map $\rho_X$, that is

$$d\rho_X : \mathfrak{g} \to \mathrm{Vect}(X) : A \mapsto \frac{d}{d\tau}\bigg|_{\tau=0} \rho_X(\exp(\tau A))(\mathbf{x}) \equiv \Pi_A(\mathbf{x}) \,. \tag{1}$$

$\Pi_A$ is called the **fundamental vector field** corresponding to $A \in \mathfrak{g}$. Given a fixed point $\mathbf{x}_0 \in X$, we denote $\tau = \xi_A(\mathbf{x}_0)^{-1}(\mathbf{x})$ the **fundamental flow coordinate**, which is the parameter such that applying the flow to $\mathbf{x}_0$ gives $\mathbf{x}$. Central to our discussion is the fact that any smooth vector field $V : X \to TX$ on $X$ can be interpreted as a differential operator acting on smooth functions $f : X \to \mathbb{R}$. The operator $V(f)$ represents the directional derivative of $f$ at $\mathbf{x} \in X$ in the direction of $V(\mathbf{x})$. We denote $\mathcal{L}_A = \Pi_A \cdot \nabla$ the differential operator corresponding to $\Pi_A$. In the following we will use both $\Pi_\tau$ and $\Pi_A$ interchangeably, when no potential confusion arises. When $\dim \mathfrak{g} > 1$ we indicate as $\mathbf{\Pi}(\mathbf{x}) = (\Pi_{A_1} \quad \Pi_{A_2} \quad \cdots)$ the matrix of the collection of fundamental vector fields.

Let us work out the example for $X = \mathbb{R}^2$ and $G = \mathrm{SO}(2)$, the group of rotations in the plane. The Lie algebra $\mathfrak{so}(2)$ consists of all matrices of the form $A_\alpha = \begin{pmatrix} 0 & -\alpha \\ \alpha & 0 \end{pmatrix}$, where $\alpha \in \mathbb{R}$, and the Lie bracket is identically zero. The flow on $X$ induced by $\rho_X$ is given by the exponential map $\rho_{\mathbb{R}^2}(\exp(\tau A_\alpha))(\mathbf{x}) = \begin{pmatrix} \cos(\alpha\tau) & -\sin(\alpha\tau) \\ \sin(\alpha\tau) & \cos(\alpha\tau) \end{pmatrix} \mathbf{x}$, and without loss of generality we can set $\alpha = 1$. The infinitesimal action is computed as

$$d\rho_{\mathbb{R}^2}(A) = \frac{d}{d\tau}\bigg|_{\tau=0} \begin{pmatrix} \cos\tau & -\sin\tau \\ \sin\tau & \cos\tau \end{pmatrix} \begin{pmatrix} x_1 \\ x_2 \end{pmatrix} = \begin{pmatrix} -x_2 \\ x_1 \end{pmatrix} \,. \tag{2}$$

[*]The code to reproduce our results will be open-sourced upon publication.

[†]We restrict ourselves here to real Lie groups. It would be interesting to extend the analysis presented here to the complex case as well (Le et al., 2021).

and thus the fundamental vector field defines the derivation $\mathcal{L}_A(\mathbf{x}) = -x_2 \frac{\partial}{\partial x_1} + x_1 \frac{\partial}{\partial x_2}$. Let $\mathbf{x}_0 \in \mathbb{R}^2$ be a fixed point, then the flow equation $\mathbf{x}(\tau) \equiv \xi_A(\mathbf{x}_0, \tau) = \rho_{\mathbb{R}^2}(\exp(\tau A), \mathbf{x}_0)$ gives a system of two equations, which we can solve to find the expression of the fundamental flow coordinate

$$\begin{cases} \mathbf{x} \cdot \mathbf{x}_0 &= |\mathbf{x}_0|^2 \cos\tau , \\ \mathbf{x} \times \mathbf{x}_0 &= |\mathbf{x}_0|^2 \sin\tau , \end{cases} \Rightarrow \quad \tau = \arctan \frac{\mathbf{x} \times \mathbf{x}_0}{\mathbf{x} \cdot \mathbf{x}_0} . \tag{3}$$

where $\mathbf{x} \times \mathbf{y} = y_2 x_1 - x_1 y_2$. Note that $\frac{\partial}{\partial \tau} = \frac{\partial x_1}{\partial \tau} \frac{\partial}{\partial x_1} + \frac{\partial x_2}{\partial \tau} \frac{\partial}{\partial x_2} = -x_2 \frac{\partial}{\partial x_1} + x_1 \frac{\partial}{\partial x_2} = \Pi_A(\mathbf{x})^\top \nabla = \mathcal{L}_A$ .

## 2.1 Intuition behind Lie group-induced generalized score matching

Score matching aims at estimating a (log) probability density $p(\mathbf{x})$ by learning to match its score function, i.e., its gradient in data space. Generalized score matching replaces the gradient operator with a general linear operator $\mathcal{L}$. The learning objective is given by minimizing the generalized Fisher divergence

$$D_{\mathcal{L}}(p||q_\theta) = \int_X p(\mathbf{x}) |\mathcal{L} \log p(\mathbf{x}) - \mathbf{s}_\theta(\mathbf{x})|^2 \, d\mathbf{x} , \tag{4}$$

where $\mathbf{s}_\theta = \mathcal{L} \log q_\theta$. The requirement on the choice of $\mathcal{L}$ is that it preserves all the information about the original density. Formally, we require $\mathcal{L}$ to be *complete*, that is, given two densities $p(\mathbf{x})$ and $q(\mathbf{x})$, $\mathcal{L}p(\mathbf{x}) = \mathcal{L}q(\mathbf{x})$ (almost everywhere [*]) implies that $p(\mathbf{x}) = q(\mathbf{x})$ (almost everywhere).

Given a Lie group $G$ acting on $X$, the collection of fundamental fields $\Pi$ corresponding to a choice of basis $\mathbf{A} = (A_1, A_2, \dots)$ of $\mathfrak{g}$ is a linear operator, thus potentially suitable for score-matching. It is then natural to set $\mathcal{L}$ to the derivation associated with the fundamental fields $\Pi$, i.e., $\mathcal{L} = \Pi(\mathbf{x})^\top \nabla$. It then follows that $\mathcal{L} \log p(\mathbf{x})$ computes the directional derivatives of $\log p(\mathbf{x})$ with respect to the fundamental flow coordinates $\boldsymbol{\tau}$, and provided that $\Pi$ meets some consistency conditions (which we will address in the next section), we can employ $\mathcal{L} \log p(\mathbf{x})$ to sample from $p(\mathbf{x})$ using Langevin dynamics:

$$\mathbf{x}_{t+1} = \mathbf{x}_t - \mathbf{s}_\theta(\mathbf{x}_t) d\rho_X(\exp(\boldsymbol{\tau}\mathbf{A}))(\mathbf{x}_t) = \mathbf{x}_t - \sum_i \underbrace{\mathcal{L}_i \log p_t(\mathbf{x}_t)}_{\text{generalized scores}} \underbrace{\Pi_{A_i}(\mathbf{x}_t)}_{A_i \text{ directions}} \Delta t , \tag{5}$$

where $\Delta t$ is the step size and we have temporarily set aside stochasticity and denoising aspects. This process mirrors the example depicted in Figure 1c: each infinitesimal step of the dynamics corresponds to infinitesimal transformations along the flow on $X$ induces by the $G$-action, and each component of the generalized score is learned through maximum likelihood over the orbits $\xi_{A_i}$ of the corresponding transformations.

## 2.2 Sufficient conditions for Lie group-induced generalized score matching

We now address the properties our setup $(X, G, \mathfrak{g}\ \Pi)$ must satisfy to meet the sufficient conditions for score-matching and Langevin dynamics. We note that these result hold for any differentiable manifold $X$. Proofs for these results can be found in Appendix B.

**Condition 1: Completeness of $\Pi$.** We start by establishing an algebraic-geometric condition for $\Pi$'s completeness:

**Proposition 2.1.** *The linear operator induced by $\Pi$ is complete if $\Pi$ is the local frame of a vector bundle $E$ over $X$ whose rank is $n \geq \dim X$ almost everywhere. If rank $E = n$ everywhere, then $E = TX$, the tangent bundle of $X$.*

The following result specifies which Lie groups yield operators $\Pi$ satisfying the above proposition:

**Proposition 2.2.** *The operator $\Pi$ induced by $\mathfrak{g}$ is complete if and only if the subspace $U \subseteq X$ such that $\dim \frac{G}{G_\mathbf{x}} < n$ for $\mathbf{x} \in U$, where $n = \dim X$, has measure zero in $X$.*

---

[*]Almost everywhere means everywhere except for a set of points of measure zero, where we assume the standard Lebesgue measure.

As an example, consider standard score-matching on mass-centered point clouds. Here $X = \mathbb{R}^{3N-3}$, since the points' coordinates satisfy $\sum_{i=1}^{N} \mathbf{x}_i = 0$. Without loss of generality, $X$ can be parametrized by $\mathbf{x}_{1,\ldots,N-1}$, with $\mathbf{x}_N$ determined by the center of mass condition. The group $G = T(3N)$ acts transitively on $X$, with a 3-dimensional stabilizer subgroup $G_X = \{(0,\ldots,0,\mathbf{a})^\top \in \mathbb{R}^{3N}\}$ fixing the space. Thus, $\dim G/G_X = n$ for all $\mathbf{x} \in X$, satisfying Proposition 2.2.

**Condition 2: Homogeneity of $X$.** While the completeness of the operators is necessary for estimating the target density, it is not sufficient to ensure that the Langevin dynamics (5) will behave appropriately, as the following example illustrates. Let $X = \mathbb{R}$, and $G = \mathbb{R}_+^*$, the multiplicative group of non-zero positive real numbers. The orbits under the action $\rho_X(a,x) = ax$ are $\mathcal{O}_+ = (0,\infty)$, $\mathcal{O}_- = (-\infty,0)$, and $\mathcal{O}_0 = \{0\}$. If the dynamics begins within $\mathcal{O}_+$, it will be never be able to reach values in $\mathcal{O}_-$, as $G$-transformations cannot move the system outside its initial orbit. We therefore ask that each pair of points of $X$ is connected through the $G$ action. This amounts to require that $X$ is **homogeneous** for $G$, that is, $\forall \mathbf{x}, \mathbf{y} \in X$ there exists a $g \in G$ such that $\rho_X(g)\mathbf{x} = \mathbf{y}$. We note that this condition solely ensures the generation outcome is independent of the initial sampling condition. Beyond this, the formalism remains fully applicable in the non-homogeneous case.

**Condition 3: Commutativity of $\Pi$.** The final requirement is that $\Pi$ forms a (locally) **commuting** frame of vector fields, $[\mathcal{L}_A, \mathcal{L}_B]f(\mathbf{x}) = 0 \, \forall A, B$ and $\forall f \in C^\infty(X)$. In this case, the coordinates $\tau_i$'s are orthogonal, and their flows commute, meaning the orbits parametrized by $\tau_i$ correspond to $\{\tau_j = 0\}_{j\neq i}$. For non-commuting flows this is not the case, as Figure 2b-c illustrates: (b) $V_1 = x_1 \partial_{x_1} + x_2 \partial_{x_2}$, $V_2 = x_1 \partial_{x_2} - x_2 \partial_{x_1}$ satisfy $[V_1, V_2] = 0$, and the orbits parametrized by $\tau_1 = r$ correspond to subspaces with constant $\tau_2 = \theta$; (c) $W_{1,2} = V_{1,2}/|\mathbf{x}|$ do not commute, and the loci $\theta = $ const no longer coincide with the $r$-orbits, causing $\theta$ to vary along these, despite the fact that $r, \theta$ are still orthogonal at each point. This last condition ensures that the updates governed by the different elements $A_i$ of $\mathfrak{g}$ in (5) remain independent of one another. Notably, this does not exclude non-Abelian groups; even if $A_{1,2} \in \mathfrak{g}$ do not commute in the Lie algebra, their flows on $X$ can, as shown in the $\mathfrak{g} = \mathfrak{so}(3)$ example in Appendix A.3.

# 3 LIE ALGEBRA SCORE-BASED GENERATIVE MODELING VIA SDES

In this section we formalize the framework we developed above from the point of view of SDEs and derive a Lie group curved dynamics in Euclidean space. Namely, we show that there exists a class of SDEs, which, when reversed, can generate data according to dynamics similar to (5), guided by the generalized score of the fundamental vector fields of the Lie algebra $\mathfrak{g}$. Our main result is provided by the following

**Theorem 3.1.** *Let $G$ be a Lie group acting on $X$ satisfying the conditions of Section 2.2, and let $\mathfrak{g}$ be its Lie algebra. The pair of SDEs*

$$d\mathbf{x} = \left[\beta(t)\mathbf{\Pi}(\mathbf{x})\mathbf{f}(\mathbf{x}) + \frac{\gamma(t)^2}{2}\rho_X(\Omega)\right]dt + \gamma(t)\mathbf{\Pi}(\mathbf{x})d\mathbf{W} \, , \tag{6}$$

$$d\mathbf{x} = \left[\beta(t)\mathbf{\Pi}(\mathbf{x})\mathbf{f}(\mathbf{x}) - \frac{\gamma^2(t)}{2}\rho_X(\Omega) - \gamma^2(t)\mathbf{\Pi}(\mathbf{x})\nabla^\top \cdot \mathbf{\Pi}(\mathbf{x})\right.$$
$$\left. -\gamma(t)^2\mathbf{\Pi}(\mathbf{x})\mathcal{L}\log p_t(\mathbf{x})\right]dt + \gamma(t)\mathbf{\Pi}(\mathbf{x})d\mathbf{W} \, , \tag{7}$$

*where $\beta, \gamma : \mathbb{R} \to \mathbb{R}$ are time-dependent functions, $\mathbf{\Pi} : \mathbb{R}^n \to \mathbb{R}^{n\times n}$ the fundamental vector fields, $\mathbf{f} : \mathbb{R}^n \to \mathbb{R}^n$ the drift, $\Omega = \sum_i A_i^2$ is known as the quadratic Casimir element of $\mathfrak{g}$, and $\mathcal{L} = \mathbf{\Pi}(\mathbf{x})^\top \nabla$, is such that*

1. *The forward-time SDE (6) is exactly solvable, with solution*

$$\mathbf{x}(t) = \left(\prod_i O_i(\tau_i(t))\right)\mathbf{x}(0) = \left(\prod_{i=1}^{n} e^{\tau_i(t)A_i}\right)\mathbf{x}(0) \, , \tag{8}$$

*where $O_i = e^{\tau_i(t)A_i}$ is the finite group action and $\boldsymbol{\tau}(t)$ is the solution to the SDE*

$$d\boldsymbol{\tau}(\mathbf{x}) = \beta(t)\mathbf{f}(\mathbf{x})dt + \gamma(t)d\mathbf{W} \, . \tag{9}$$

2. *The SDE* (7) *is the reverse-time process of* (6).

3. *The Langevin dynamic of the above SDEs decomposes as a direct sum of $\mathfrak{g}$ infinitesimal actions* (1)*, each defining an infinitesimal transformation along the flows $\xi_{\boldsymbol{\tau}}$.*

We refer to Appendix C for the proof of the above result. Here we limit ourselves to a few comments regarding the extra terms that appear in the SDEs. The appearance of the Casimir element (we assume the identity as bilinear form on $\mathfrak{g}$ (Kac & Kac, 1983)) compensates for the deviation of the tangent vector from the orbit due to the curvature of the flow coordinates. This can be seen in the example of $SO(2)$ acting on $\mathbb{R}^2$ (which will be discussed thoroughly below). An infinitesimal transformation along the $\theta$ direction, represented by $\Pi_\theta$, moves any point $\mathbf{x}$ along a vector tangent to its $SO(2)$ orbit, a circle of radius $r = \sqrt{x_1^2 + x_2^2}$. Due to the orbit's non-zero curvature, this movement would shift the point to an orbit of radius $r' > r$. The term $\rho_X(\Omega)$ compensates for this displacement, ensuring the final point remains close to the original orbit. This is illustrated in Figure 3.

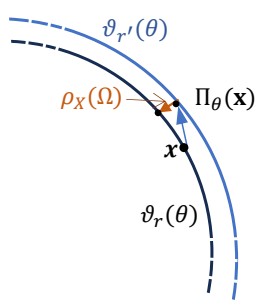

Figure 3: Quadratic Casimir for $G = SO(2)$.

With this result at hand we can formulate our procedure for our Lie group-induced score-based generative modeling with SDEs.

**Perturbing data through the SDE.** The forward-time SDE (6) defines a noising diffusion process respecting the decomposition of the Lie algebra $\mathfrak{g}$ infinitesimal actions on $X$. In fact, given a data sample $\mathbf{x}(0) \sim p_0$, the solution (8) takes the form of a product of finite group element actions $\mathcal{O}_i$ on $\mathbf{x}(0)$, where the specific order is irrelevant since the Lie algebra generators commute. For each factor, we first determine $\boldsymbol{\tau}(0) = \boldsymbol{\tau}(\mathbf{x}(0))$, and employ these as initial conditions for the forward SDE (9). By choosing appropriately the drift terms $f_i's$, for instance, to be affine in the flow coordinates $\tau_i$, we can solve for $\boldsymbol{\tau}(t)$ with standard techniques (Särkkä & Solin, 2019), as this will follow a Gaussian distribution. Alternatively, we can sample from $\boldsymbol{\tau}(t)$ by first simulating (9), then performing sliced score matching Song et al. (2020a); Pang et al. (2020) to sample from $p_t(\mathbf{x}(\boldsymbol{\tau}(t))|\mathbf{x}(0))$.

**Generating samples through the reverse SDE.** The time-reverse SDE (7) guides the generation of samples $\mathbf{x}(0) \sim p_0(\mathbf{x})$ starting from samples $\mathbf{x}(T) \sim p_T(\mathbf{x})$, provided we can estimate the generalized score $\mathcal{L} \log p_t(\mathbf{x})$ of each marginal distribution. To sample from $p_T$, we use the fact that the distribution in the flow coordinates $\boldsymbol{\tau}$ is tractable (with an appropriate choice of the drift terms and time-dependent functions $\beta, \gamma$ in (6)), and that (since $p_t(\mathbf{x})d\mathbf{x} = p_t(\boldsymbol{\tau})d\boldsymbol{\tau}$)

$$p_t(\mathbf{x}) = p_t(\boldsymbol{\tau}) \left| \frac{\partial \boldsymbol{\tau}}{\partial \mathbf{x}} \right| = p_t(\boldsymbol{\tau}) \left| \mathbf{\Pi}^{-1}(\mathbf{x}) \right| , \tag{10}$$

where the extra term corresponds to the determinant of the Jacobian of the coordinate transformation induced by the fundamental flow coordinates. In particular, when $\mathbf{f}(\boldsymbol{\tau})$ is affine, it follows that $p_T(\boldsymbol{\tau}) = \mathcal{N}(\boldsymbol{\tau} \mid \mathbf{0}, \boldsymbol{\Sigma})$, where $\boldsymbol{\Sigma} = \text{diag}(\sigma_1^2, \sigma_2^2, \dots, \sigma_n^2)$. Thus, we can sample $\boldsymbol{\tau}(T) \sim p_T(\boldsymbol{\tau})$ simply as a collection of independent Gaussian random variables, and use the flow map to obtain $\mathbf{x}(T) = \boldsymbol{\xi}_{\mathbf{A}}(\boldsymbol{\tau}(T), \mathbf{x}_0)$, which will follow the distribution (10) for $t = T$.

**Estimating the generalized score.** Analogously to standard score-matching, we train a time-dependent neural network $\mathbf{s}_{\boldsymbol{\theta}}(\mathbf{x}(t), t) : \mathbb{R}^n \times \mathbb{R} \to \mathbb{R}^n$ to estimate the generalized score $\mathcal{L} \log p_t(\mathbf{x}(t)|\mathbf{x}(0))$ at any time point, that is, we minimize the objective

$$\mathbb{E}_t \left\{ w(t) \mathbb{E}_{\mathbf{x}(0) \sim p_0(\mathbf{x})} \mathbb{E}_{\mathbf{x}(t) \sim p_t(\mathbf{x}|\mathbf{x}(0))} \left[ \left| \mathbf{s}_{\boldsymbol{\theta}}(\mathbf{x}(t), t) - \mathcal{L} \log p_t(\mathbf{x}(t)|\mathbf{x}(0)) \right|^2 \right] \right\} , \tag{11}$$

where $w : [0, T] \to \mathbb{R}_+$ is a time-weighting function. Now, from Condition 3 above and the property that $\mathcal{L}_{A_i}$ computes the direction derivative along the flow of $\Pi_{A_i}(\mathbf{x})$, it follows that $\mathcal{L} \log p_t(\mathbf{x}(t)|\mathbf{x}(0)) = \nabla_{\boldsymbol{\tau}(t)} \log p_t(\mathbf{x}(\boldsymbol{\tau})(t)|\mathbf{x}(\boldsymbol{\tau})(0))$. Under the above assumptions, $p_t(\boldsymbol{\tau}) = \mathcal{N}(\boldsymbol{\tau}|\boldsymbol{\mu}(\mathbf{x}(0), t), \boldsymbol{\Sigma}(t))$, where the form of the mean and the variance depends on the explicit form of (9). Using the parametrization $\boldsymbol{\tau}(t) = \boldsymbol{\mu}(\mathbf{x}(0), t) + \sqrt{\boldsymbol{\Sigma}(t)}\boldsymbol{\eta}_t$, where $\boldsymbol{\eta}_t \sim \mathcal{N}(\mathbf{0}, \boldsymbol{I})$, we obtain

$$\mathcal{L} \log p_t(\mathbf{x}(t)|\mathbf{x}(0)) = -\boldsymbol{\Sigma}^{-1}(\boldsymbol{\tau}(t) - \boldsymbol{\mu}(\mathbf{x}(0), t) = -\sqrt{\boldsymbol{\Sigma}(t)}^{-1}\boldsymbol{\eta}_t . \tag{12}$$

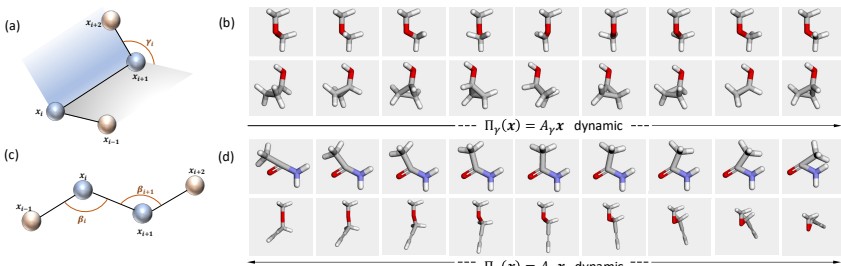

Figure 4: Lie algebra $\mathfrak{so}(2) \subset \mathfrak{so}(3)$ dynamics for torsion (a,b) and bond angles (c,d) in molecular conformers.

## 3.1 EXAMPLES

**Standard Score Matching.** Standard score matching can be recovered as a special case of our formalism by choosing $X = \mathbb{R}^n$ and $G = T(n)$. As we show explicitly in Appendix A.1, we have $\mathcal{L} = \nabla$ and the Lie algebra action $\mathbf{\Pi}(\mathbf{x}) = \mathbf{I}$, the identity on $X$. Since $\mathbf{\Pi}$ is $\mathbf{x}$-independent, its divergence vanishes, as well as the quadratic Casimir ($T(N)$ is Abelian), so that the SDEs (6) take the known form

$$d\mathbf{x} = \beta(t)\mathbf{f}(\mathbf{x})dt + \gamma(t)d\mathbf{W} , \qquad d\mathbf{x} = \left[\beta(t)\mathbf{f}(\mathbf{x}) - \gamma(t)^2 \nabla \log p_t(\mathbf{x})\right] dt + \gamma(t)d\mathbf{W} . \tag{13}$$

$G = \mathbf{SO(2)} \times \mathbb{R}_+$. A simple but non-trivial case in given by $G = \mathrm{SO}(2) \times \mathbb{R}_+$ describing rotations and dilations acting on $X = \mathbb{R}^2$. A basis for the 2-dimensional Lie algebra $\mathfrak{g} = \mathfrak{so}(2) \oplus \mathbb{R}$ is given by $A_r = I$ and $A_\theta = \begin{pmatrix} 0 & -1 \\ 1 & 0 \end{pmatrix}$, yielding $\mathbf{\Pi}(\mathbf{x}) = \begin{pmatrix} x & -y \\ y & x \end{pmatrix}$, which satisfies all the conditions of section 2.2. Following our discussion above and in Appendix A.1 we have (since $\rho(\Omega) = A_r^2 + A_\theta^2 = I - I = 0$)

$$d\mathbf{x} = \beta(t) \left( f_r(r)A_r\mathbf{x} + f_\theta(\theta)A_\theta\mathbf{x} \right) dt + \gamma(t) \left( dW_r A_r\mathbf{x} + dW_\theta A_\theta\mathbf{x} \right) , \tag{14}$$

and we see that the SDE splits into contributions from the two Lie algebra summands. To find an explicit solution, let $\gamma(t) = \sqrt{\beta(t)}$ and $f_r = -\frac{1}{4}\log(x^2 + y^2)$, $f_\theta = -\frac{1}{2}\arctan\frac{y}{x}$. This choice corresponds, in the flow coordinates system, to a 2d Ornstein-Uhlenbeck system (Gardiner, 1985) which has a Gaussian solution with mean $\begin{pmatrix} r(0) \\ \theta(0) \end{pmatrix} e^{-\int_0^t \beta(s)ds}$ and variance $\left( 1 - e^{-\int_0^t \beta(s)ds} \right) I$.

Let us define $\sigma(t) = \sqrt{1 - e^{-\int_0^t \beta(s)ds}}$, such that $r(t) = r(0) + \lambda(t) = r(0) - r(0)\sigma(t)^2 + \sigma(t)\eta_r$ and similarly $\theta(t) = \theta(0) + \varphi(t) = \theta(0) - \theta(0)\sigma(t)^2 + \sigma(t)\eta_\theta$ where $\eta_r, \eta_\theta \in \mathcal{N}(0,1)$, then it is an easy calculation to show that

$$\begin{pmatrix} x(t) \\ y(t) \end{pmatrix} = \begin{pmatrix} e^{r(t)}\cos(\theta(t)) \\ e^{r(t)}\sin(\theta(t)) \end{pmatrix} = \begin{pmatrix} e^{\lambda(t)} & 0 \\ 0 & e^{\lambda(t)} \end{pmatrix} \begin{pmatrix} \cos\varphi(t) & -\sin\varphi(t) \\ \sin\varphi(t) & \cos\varphi(t) \end{pmatrix} \begin{pmatrix} x(0) \\ y(0) \end{pmatrix} . \tag{15}$$

We can look at the asymptotic behavior of the solution. Assuming that $\beta(t)$ is a monotonous increasing function, that is, $\beta(t + \epsilon) > \beta(t)$ for $\epsilon > 0$, then $\lim_{t\to\infty}\sigma(t) = 1$ and hence

$$\lim_{t\to\infty} \mathbf{x}(t) = e^{-r_0 + \eta_r} \begin{pmatrix} \cos\theta_0 & \sin\theta_0 \\ -\sin\theta_0 & \cos\theta_0 \end{pmatrix} \begin{pmatrix} \cos\eta_\theta & \sin\eta_\theta \\ -\sin\eta_\theta & \cos\eta_\theta \end{pmatrix} \begin{pmatrix} e^{r_0}\cos\theta_0 \\ e^{r_0}\sin\theta_0 \end{pmatrix} = \begin{pmatrix} e^{\eta_r}\cos\eta_\theta \\ e^{\eta_r}\sin\eta_\theta \end{pmatrix} ,$$

where $\theta_0 = \theta(0)$, $r_0 = r(0)$. Note that, even if (16) is not Gaussian, we can still easily draw samples from it by sampling the two Gaussian variables $\eta_{r,\theta}$.

**Dihedral and bond angles.** The above formalism can be applied to obtain transformations of physically meaningful quantities, as bond and torsion angles for molecules' conformations. Let $\gamma_i$ be the dihedral angle between the planes identified by the points $\{\mathbf{x}_{i-1}, \mathbf{x}_i, \mathbf{x}_{i+1}\}$ and $\{\mathbf{x}_i, \mathbf{x}_{i+1}, \mathbf{x}_{i+2}\}$, respectively (Figure 4a). The Lie algebra element corresponding to an infinitesimal change in $\gamma_i$ is given by a $3N \times 3N$-dimensional $3 \times 3$-block diagonal matrix, whose $j = 1, \ldots, N$ block is given by $H(j - (i+1))\hat{\mathbf{x}}_{i+1,i} \cdot \mathbf{A})$, where $\mathbf{A} = (A_x, A_y, A_z)$ is the vector of the Lie algebra basis for $\mathfrak{so}(3)$, $\hat{\mathbf{x}}_{i+1,i} = (\mathbf{x}_{i+1} - \mathbf{x}_i)/|\mathbf{x}_{i+1} - \mathbf{x}_i|$ and $H(i) = 1$ if $i > 0$ and 0 otherwise is the Heaviside

Figure 5: (a) 2d mixture of Gaussians (top: ground truth, bottom: generated); (b) generating process using single scores for the subgroups $SO(2), \mathbb{R}_+$ with the corresponding vector field generating scores (c) (d,e) one-dimensional learning for a symmetric distributions; $3d$-distributions: torus (f) and Möbius strip (g) (top: ground truth, bottom: generated);(f) 4d mixture of Gaussian for the group $G = SO(4) \times \mathbb{R}_+$.

step function. For bond angles $\beta_i$ (Figure 4c) we construct the corresponding $\mathfrak{so}(2) \in \mathfrak{so}(3)$ algebra element blocks as $H(j - i)(\mathbf{x}_{i+1,i} \times \mathbf{x}_{i-1,i}) \cdot \mathbf{A}$. Examples of the dynamics generated by these operators are presented in Figure 4(b,d). This shows that the approach of Corso et al. (2023); Jing et al. (2022); Ketata et al. (2023) can be recovered as a particular case of our general formalism. Notably, our approach performs diffusion in a flat, rather than a curved Riemannian manifold.

## 4 RELATED WORK

Representation theory applied to neural networks has been studied both theoretically (Esteves, 2020; Chughtai et al., 2023; Puny et al., 2021; Smidt, 2021) and applied to a variety of groups, architecture and data type: CNNs (Cohen & Welling, 2016; Romero et al., 2020; Liao & Liu, 2023; Finzi et al., 2020; Weiler & Cesa, 2019; Weiler et al., 2018), Graph Neural Networks (Satorras et al., 2021), Transformers, (Geiger & Smidt, 2022; Romero & Cordonnier, 2020; Hutchinson et al., 2021), point clouds (Thomas et al., 2018), chemistry (Schütt et al., 2021; Le et al., 2022a). On the topic of disentanglement of group action and symmetry learning, Pfau et al. (2020) factorize a Lie group from the orbits in data space, while Winter et al. (2022) learn through an autoencoder architecture invariant and equivariant representations of any group acting on the data. Fumero et al. (2021) learns disentangled representations solely from data pairs. Dehmamy et al. (2021) propose an architecture based on Lie algebras that can automatically discover symmetries from data. Xu et al. (2022) predict molecular conformations from molecular graphs in an roto-translation invariant fashion with equivariant Markov kernels.

Partially related to our study is the field of diffusion on Riemannian manifolds. De Bortoli et al. (2022) propose diffusion in a product space – a condition which is not a necessary in our framework – defined by the flow coordinates in the respective Riemannian sub-manifolds. When the Riemannian manifold is a Lie group, their method yields dynamics similar to ours, as illustrated in an example in Section 3.1. In fact, our formalism could be integrated with their approach to create a unified framework for diffusion processes on the broader class of Riemannian manifolds admitting a Lie group action. These techniques has been applied in a variety of use cases (Corso et al., 2023; Ketata et al., 2023; Yim et al., 2023) for protein docking, ligand and protein generation. The works Zhu et al. (2024); Kong & Tao (2024) leverage trivialized momentum to perform diffusion on the Lie algebra (isomorphic to $\mathbb{R}^n$) instead of the Lie group, thereby eliminating curvature terms, although their approach is to date only feasible for Abelian groups. An interesting connection with our work is the work of Kim et al. (2022): the authors propose a bijection to map a non-linear problem to a linear one, to approximate a bridge between two non-trivial distributions. Our case can be seen as a bijection between the (curved) Lie group manifold and the (flat) Euclidean data space.

In the context of interpreting the latent space of diffusion models, Park et al. (2023) explores the local structure of the latent space (trajectory) of diffusion models using Riemannian geometry. Similarly, Haas et al. (2024) propose a method to uncover semantically meaningful directions in the semantic latent space ($h$-space) (Wang et al., 2023) of denoising diffusion models (DDMs) by PCA. Wang et al. (2023) propose a method to learn disentangled and interpretable latent representations of diffusion models in an unsupervised way. We note that the aforementioned works aim to extract meaningful latent factors in traditional DDMs, often restricting to human-interpretable semantic features and focusing on image generation.

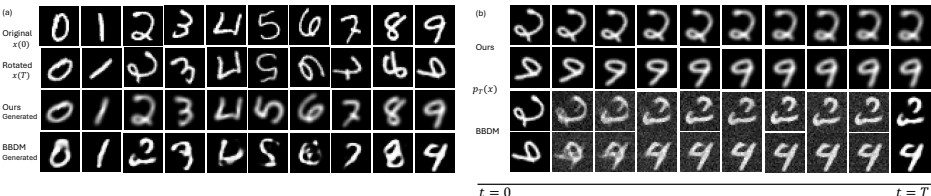

Figure 6: (a) Original and rotated MNIST samples with generated samples from our model and BBDM. (b) Reverse diffusion trajectories of our model against BBDM. Intermediate samples from BBDM resemble interpolation of mixed digits. For the last case, the 9-digit transitions into a 4-digit.

## 5 EXPERIMENTS

**2d, 3d and 4d distributions.** In Figure 5 we illustrate the framework for a variety of $d = 2, 3$-dimensional distributions. In all cases we take $G = \mathrm{SO}(d) \times \mathbb{R}_+$. Figure 5(a,b,c) displays a mixture of Gaussians: in (a) (bottom) we see that our generalized score-matching can learn any distribution, regardless of its inherent symmetry; (b) shows the output of the generation process using only one score (top $\mathfrak{g} = \mathfrak{so}(2)$, bottom $\mathfrak{g} = \mathfrak{r}_+$), while (c) shows the vector fields corresponding to the scores, where we color-coded the field directions. In Figure 5, we illustrate the framework for various $d = 2, 3$-dimensional distributions, and we choose $G = \mathrm{SO}(d) \times \mathbb{R}_+$. Figure 5(a,b,c) shows a mixture of Gaussians: (a) demonstrates that our generalized score-matching can learn any distribution, regardless of its symmetry; (b) shows the output of the generation process using a single score (top $\mathfrak{g} = \mathfrak{so}(2)$, bottom $\mathfrak{g} = \mathfrak{r}_+$) and in (c) we display the corresponding vector fields. Figures 5(d,e) depicts radial and angular distributions, where the score is learned using the respective Lie algebra elements. This reflects the ability to leverage the symmetry properties of the data and perform diffusion in a lower-dimensional space. We also show in Figure 5f ($G = \mathrm{SO})(4) \times \mathbb{R}_+$) that our method can be applied to higher dimensional Lie groups.

**Rotated MNIST.** In this experiment we show that our framework can be applied to effectively learn a bridge between two non-trivial distributions, adopting however only techniques from score-matching and DDM. Let $p_T(\mathbf{x})$ be the rotated MNIST dataset and $p_0(\mathbf{x})$ the original (non-rotated) MNIST dataset. We can learn to sample from $p_0$ starting from element of $p_T$ by simply modeling a $\mathrm{SO}(2)$ dynamic. Some examples of our results are shown in Figure 6. Notice that our formalism allows us to reduce the learning to a 1-dimensional score $\mathcal{L}_\theta = x_1 \partial_{x_1} - x_2 \partial_{x_2}$, which reflects the true dimensionality of the problem. We trained the model with $T = 100$ time-steps, but for sampling it suffices to set $T = 10$. As it can be seen in the example trajectories 6b, the model starts converging already at $t/T \sim 0.5$. We employ a CNN which processes input images $\mathbf{x}(t)$, and the resulting feature map is flattened and concatenated with a scalar input $t$, then passed through fully connected layers to produce the final output. We compare our approach to the Brownian Bridge Diffusion Model (BBDM) (Li et al., 2023). Unlike our method, BBDM operates unconstrained in the full MNIST pixel space ($\mathbb{R}^{28 \times 28}$), where intermediate states represent latent digits. As shown in Figure 6a, this can result in incorrect transitions, such as adding extraneous pixels or altering the original digit, sometimes generating entirely different digits (Figure 6b). Further modeling details can be found in Appendix E.

**QM9.** We use our framework to train a generative model $p_\theta(X|M)$ for conformer sampling of small molecules $M$ from the QM9 dataset (Ramakrishnan et al., 2014). We only keep the lowest energy conformer as provided in the original dataset, that is, for each molecule only one 3D conformer is maintained. Here $X = \mathbb{R}^{3N}$ and we choose $G = (\mathrm{SO}(3) \times \mathbb{R}_+)^N$, where each factor acts on the space $\mathbb{R}^3$ spanned by the Cartesian coordinates of the molecule's atoms, respectively. As Figure 7a shows, our generative process yields conformers that are energetically very similar to the ground truth conformers, while showing some variability, as it can be seen in the last example where the torsion angle is differently optimized. We train another model $p_\gamma(X|M)$ via standard Fisher denoising score-matching, i.e., choosing $G = T(3)^N$ as in Sec. 3.1, and generate 5 conformers per molecule for both models $p_\theta, p_\gamma$. We then compute the UFF energy (Rappe et al., 1992) implemented in the RDKit for all generated conformers and extract the lowest energy geometry as generated sample. To compare against the reference geometry, we compute the energy difference $\Delta = U_{\mathrm{true}} - U_{\mathrm{gen}}$ for both models. Figure 7b shows that both diffusion models tend to generate conformers that have

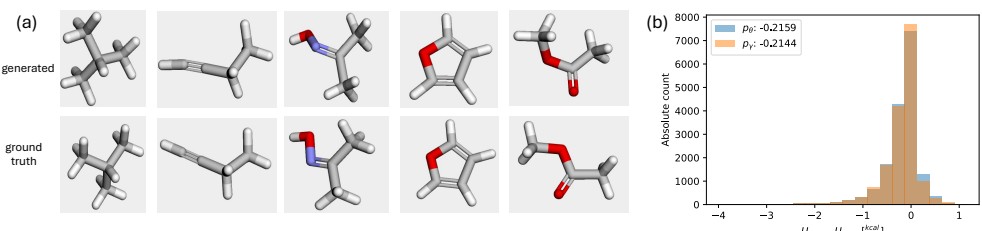

Figure 7: (a) Generated 3D conformer for several molecules from the QM9 validation set (top row) and ground truth conformer (bottom row). (b) Energy difference distribution between diffusion models $(p_\theta, p_\gamma)$ compared to the ground-truth energy. Both diffusion models generate conformers that have a similar $\Delta$ energy distribution.

lower energies than the ground true conformer according to the UFF parametrization, while the diffusion model that implements the dynamics according to $G = (\mathrm{SO}(3) \times \mathbb{R}_+)^N$ (colored in blue) achieves slightly lower energy conformers, mean $\Delta_\theta = -0.2159$ against mean $\Delta_\gamma = -0.2144$ for the standard diffusion model (colored in orange).

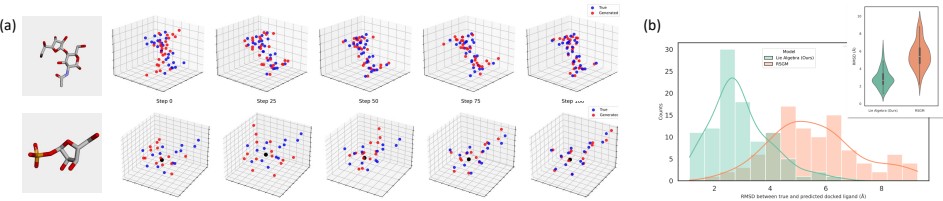

Figure 8: (a) Generated SE(3) trajectories for molecular docking. (b) Comparison with RSGM.

**CrossDocked2020: Global E(3) and Protein-Ligand Complexes.** In this final experiment, we train a generative model for global SE(3) transformations acting on small molecules. Specifically, given a pair consisting of a compound and a protein pocket in space, our goal is to generate the trajectory by which the ligand best fits into the pocket. Importantly, the internal structure of the compound remains fixed, which presents a challenge when using standard diffusion processes. Thus, while the SE(3) transformations are global with respect to the ligand, they do not represent global symmetries of the overall system. We derive in appendix A.4 the relevant operators that guide the dynamics (6), (7). Figure 8a shows examples of docked molecules using SE(3)-guided score-matching diffusion. The true and generated molecules at different generation steps are visualized as point clouds, showing a good agreement. Figure 8b shows that our model achieves a lower RMSD $(2.9 \pm 1.0$ Å vs $5.6 \pm 1.2$ Å) for the docked ligands than the method from RSGM (De Bortoli et al., 2022; Corso et al., 2023) (for implementation details we refer to Appendix E.2.1).

## 6 CONCLUSIONS AND OUTLOOK

We presented a framework for generative modeling that respects the action of *any* Lie group $G$ and derived its mathematical foundation. **Our framework generates a curved Lie group diffusion dynamics in flat Euclidean space, thus without the need to transform the data and of performing group projections.** Specifically, we introduced a new class of exactly-solvable SDEs that guide the corruption and generation processes. Traditional diffusion score-matching emerges as a special case when the group $G$ is set to the translation group in Euclidean space. Thus, our framework does not merely complement existing methods, but *expands* the space of exactly solvable diffusion processes. Our framework is particularly relevant given recent findings (Abramson et al., 2024) showing that unconstrained models outperform equivariant ones: with our framework there is no need of a trade-off, as we retain the expressivity of unconstrained models with the benefits of group inductive bias. Moreover, our techniques descend quite straightforwardly to flow matching (Lipman et al., 2022) through the Diffusion Mixture Representation Theorem (Peluchetti, 2023; Brigo, 2008). We spell out the connection in appendix D and we plan to expand on this in future work.

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

## A EXAMPLES OF LIE GROUPS AND LIE ALGEBRA ACTIONS

In this appendix we list some important Lie groups and Lie algebra actions, their corresponding fundamental vector fields as well as the fundamental flow coordinates. These will be useful in the main text.

### A.1 $T(N)$

Let $X = \mathbb{R}^N$ and $G = T(N)$, the group of translations in $N$-dimensional space. Element of $T(N)$ are represented by a vector $\mathbf{v} = (v_1, v_2, \ldots, v_N)^\top \in \mathbb{R}^N$, where $v_i$ are the translation components along the $x_i$ axes for $i = 1, \ldots, N$, thus $T(N) \simeq \mathbb{R}^N$. Explicitly, for a $\mathbf{x} \in X$ its action is given by $\rho_{\mathbb{R}^N}(\mathbf{v}, \mathbf{x}) = \mathbf{x} + \mathbf{v}$.

The corresponding Lie algebra $\mathfrak{t}(N)$ is also isomorphic to $\mathbb{R}^N$, and it consists of vectors $\mathbf{a} = (a_1, a_2, \ldots, a_N)^\top \in \mathbb{R}^N$. The Lie bracket of any two elements in $\mathfrak{t}(N)$ vanishes, as $T(N)$ is Abelian.

To derive the infinitesimal action, we first note that the exponential map is trivial, $\exp(\tau \mathbf{A}) = \tau \mathbf{A}$. Hence, we have

$$\Pi_A(\mathbf{x}) = \frac{d}{d\tau}\bigg|_{\tau=0} \rho_{\mathbb{R}^N}(\tau\mathbf{A}, \mathbf{x}) = \frac{d}{d\tau}\bigg|_{\tau=0} (\mathbf{x} + \tau\mathbf{A}) = \mathbf{A} . \tag{16}$$

Thus, the fundamental vector field $\Pi_A$ corresponding to $\mathbf{A} \in \mathfrak{t}(N)$ is the constant vector field:

$$\Pi_A = a_1 \frac{\partial}{\partial x_1} + a_2 \frac{\partial}{\partial x_2} + \cdots + a_N \frac{\partial}{\partial x_N} = \mathbf{A} \cdot \nabla .$$

### A.2 $X = \mathbb{R}^N, G = \mathbb{R}_+^*$ (GROUP OF DILATIONS)

Let us consider $X = \mathbb{R}^N$ and $G = \mathbb{R}_+^*$, the group of dilations in $N$-dimensional space. The group $\mathbb{R}_+^*$ consists of all positive scaling factors. Each element of $G = \mathbb{R}_+^*$ can be represented by a scalar $\lambda > 0$ that scales all vectors in $\mathbb{R}^N$ by this factor.

The action of $G = \mathbb{R}_+^*$ on $\mathbb{R}^N$ is a dilation, meaning that every vector $\mathbf{x} = (x_1, x_2, \ldots, x_N)^\top \in \mathbb{R}^N$ is scaled by the factor $\lambda$. Explicitly, the group action is given by

$$\rho_{\mathbb{R}^N}(\lambda, \mathbf{x}) = \lambda\mathbf{x} . \tag{17}$$

The Lie algebra $\mathfrak{g} = \mathbb{R}$ corresponding to the dilation group $G = \mathbb{R}_+^*$ consists of real numbers representing the logarithm of the scaling factor. Specifically, an element $A \in \mathfrak{g}$ corresponds to a generator of the dilation, and the exponential map $\exp : \mathfrak{g} \to G$ is given by: $\exp(\tau A) = e^{\tau A}$, where $\tau$ is a real parameter.

The infinitesimal action corresponds to taking the derivative at $\tau = 0$. For a vector $\mathbf{x} \in \mathbb{R}^N$ and $A \in \mathfrak{g}$, the fundamental vector field $\Pi_A$ is computed as:

$$d\rho_{\mathbb{R}^N}(A) = \frac{d}{d\tau}\bigg|_{\tau=0} \rho_{\mathbb{R}^N}(e^{\tau A}, \mathbf{x}) = \frac{d}{d\tau}\bigg|_{\tau=0} (e^{\tau A}\mathbf{x}) = A\mathbf{x} , \tag{18}$$

and

$$\Pi_A(\mathbf{x}) = A\mathbf{x} \cdot \nabla .$$

Now, solving the equation

$$\mathbf{x} = e^{\tau A}\mathbf{x}_0 \tag{19}$$

in terms of $\tau$ we obtain

$$\tau = \frac{1}{A} \log \frac{|\mathbf{x}|^2}{\mathbf{x} \cdot \mathbf{x}_0} = \frac{1}{A} \log \frac{|\mathbf{x}|^2}{|\mathbf{x}||\mathbf{x}_0|} = \frac{1}{A} \log \frac{|\mathbf{x}|}{|\mathbf{x}_0|} = \frac{1}{2A} \log \frac{|\mathbf{x}|^2}{|\mathbf{x}_0|^2} . \tag{20}$$

In the usual case of $A = 1$ (generator of the Lie algebra), $\mathbf{x}_0 = \frac{1}{\sqrt{N}}(1, 1, \ldots, 1)^\top$ to be the unit vector we obtain the usual expression

$$\tau = \frac{1}{2} \log(x_1^2 + x_2^2 + \cdots + x_N^2) \,. \tag{21}$$

## A.3  $X = \mathbb{R}^3$, $G = \mathrm{SO}(3) \times \mathbb{R}_+^*$

The dilation part is solved in the previous section, so we actually just focus on the action of $\mathrm{SO}(3)$ on $\mathbb{R}^3$. The orbits are given by spheres centered at the origin, and we can decompose the action of $\mathrm{SO}(3)$ by varying the azimuthal or the polar angle defined by a vector $\mathbf{x}$. Namely, we have the two actions

$$\rho_{\mathbb{R}^3}(\varphi, \mathbf{x}) = \begin{pmatrix} \cos\varphi & -\sin\varphi & 0 \\ \sin\varphi & \cos\varphi & 0 \\ 0 & 0 & 1 \end{pmatrix} \begin{pmatrix} x \\ y \\ z \end{pmatrix} \,,$$

$$\rho_{\mathbb{R}^3}(\theta, \mathbf{x}) = \left[ I + \sin\theta \begin{pmatrix} 0 & 0 & \cos\varphi \\ 0 & 0 & \sin\varphi \\ -\cos\varphi & -\sin\varphi & 0 \end{pmatrix} \right.$$

$$\left. + (1 - \cos\theta) \begin{pmatrix} -\cos^2\varphi & -\cos\varphi\sin\varphi & 0 \\ -\cos\varphi\sin\varphi & -\sin^2\varphi & 0 \\ 0 & 0 & -1 \end{pmatrix} \right] \begin{pmatrix} x \\ y \\ z \end{pmatrix} \,. \tag{22}$$

If we take the differentials

$$d\rho_{\mathbb{R}^3}(\varphi, \mathbf{x})|_{\varphi=0} = \begin{pmatrix} -\sin\varphi & -\cos\varphi & 0 \\ \cos\varphi & -\sin\varphi & 0 \\ 0 & 0 & 0 \end{pmatrix} \begin{pmatrix} x \\ y \\ z \end{pmatrix} \bigg|_{\varphi=0} = \begin{pmatrix} 0 & -1 & 0 \\ 1 & 0 & 0 \\ 0 & 0 & 0 \end{pmatrix} \begin{pmatrix} x \\ y \\ z \end{pmatrix} = A_z \mathbf{x} \,,$$

$$d\rho_{\mathbb{R}^3}(\theta, \mathbf{x}) = \left[ \cos\theta \begin{pmatrix} 0 & 0 & \cos\varphi \\ 0 & 0 & \sin\varphi \\ -\cos\varphi & -\sin\varphi & 0 \end{pmatrix} \right.$$

$$\left. - \sin\theta \begin{pmatrix} \cos^2\varphi & \cos\varphi\sin\varphi & 0 \\ \cos\varphi\sin\varphi & \sin^2\varphi & 0 \\ 0 & 0 & 1 \end{pmatrix} \right]_{\theta=0} \begin{pmatrix} x \\ y \\ z \end{pmatrix}$$

$$= \begin{pmatrix} 0 & 0 & \cos\varphi \\ 0 & 0 & \sin\varphi \\ -\cos\varphi & -\sin\varphi & 0 \end{pmatrix} \begin{pmatrix} x \\ y \\ z \end{pmatrix} = (\cos\varphi A_y - \sin\varphi A_x)\mathbf{x} \,, \tag{23}$$

where

$$A_x = \begin{pmatrix} 0 & 0 & 0 \\ 0 & 0 & -1 \\ 0 & 1 & 0 \end{pmatrix} \qquad A_y = \begin{pmatrix} 0 & 0 & 1 \\ 0 & 0 & 0 \\ -1 & 0 & 0 \end{pmatrix} \qquad A_z = \begin{pmatrix} 0 & -1 & 0 \\ 1 & 0 & 0 \\ 0 & 0 & 0 \end{pmatrix} \tag{24}$$

form a basis for $\mathfrak{so}(3)$. The corresponding differential operators are

$$\mathcal{L}_\varphi = x\partial_y - y\partial_x \,, \qquad \mathcal{L}_\theta = \frac{1}{\sqrt{x^2 + y^2}} \left[ zx\partial_x + zy\partial_y - (x^2 + y^2)\partial_z \right] \,, \tag{25}$$

and it is an easy calculation to show that they commute $[\mathcal{L}_\varphi, \mathcal{L}_\theta] = 0$. The attentive reader might have noticed that the commutation does not hold at the matrices level. While this is expected, since there is no 2-dimensional commuting subalgebra in $\mathfrak{so}(3)$, it is nonetheless quite puzzling since everything works out at the level of differential operators. This reflect the fact that the commutation properties are necessary at the level of the action of $\mathfrak{g}$ on $X$, and not necessarily at the Lie algebra level. In this case, however, we can elegantly resolve the puzzle indem we found a matrix representation for the action $d\rho_{\mathbb{R}^3}(\theta)\mathbf{x}$ which does commute with the $\varphi$ action. To do this we note that we can rewrite

$$\mathcal{L}_\theta = \frac{\cos\theta}{\sin\theta} x\partial_x + \frac{\cos\theta}{\sin\theta} y\partial_y - \frac{\sin\theta}{\cos\theta} z\partial_z \,, \tag{26}$$

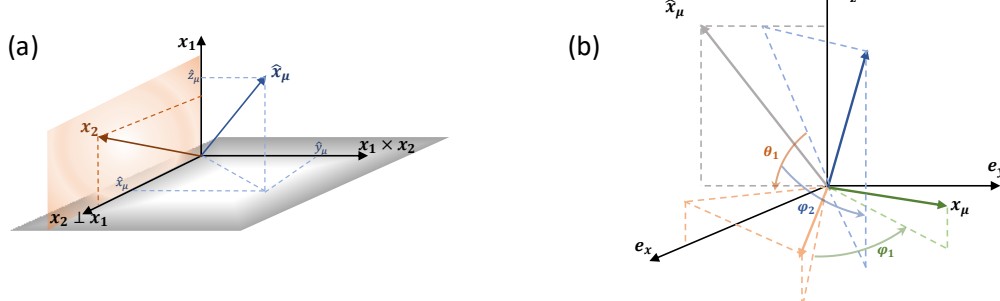

Figure 9: (a) The coordinates $\widehat{\mathbf{x}}_\mu$ are the coordinates in the coordinate system defined by $\mathbf{x}_1$, the orthogonal projection of $\mathbf{x}_2$ with respect to $x_1$. $\mathbf{x}_2 \perp \mathbf{x}_1 = \mathbf{x}_2 - \mathbf{x}_1 \cdot \mathbf{x}_2$, and $\mathbf{x}_2 \times \mathbf{x}_1$. (b) Graphical depiction of the global symmetry transformations parametrized by the three angles $\varphi_2, \theta_1, \varphi_1$.

which corresponds to simultaneous dilations, with different coefficient, in the $z$ axis and $x, y$-plane. The finite action takes the form

$$\widetilde{\rho}_{\mathbb{R}^3}(\theta, \mathbf{x}) = \exp\left[\log\sin\theta \begin{pmatrix} 1 & 0 & 0 \\ 0 & 1 & 0 \\ 0 & 0 & 0 \end{pmatrix} + \log\cos\theta \begin{pmatrix} 0 & 0 & 0 \\ 0 & 0 & 0 \\ 0 & 0 & 1 \end{pmatrix}\right], \tag{27}$$

and computing the first order term we obtain

$$d\widetilde{\rho}_{\mathbb{R}^3}(\theta, \mathbf{x}) = \begin{pmatrix} \frac{\cos\theta}{\sin\theta} & 0 & 0 \\ 0 & \frac{\cos\theta}{\sin\theta} & 0 \\ 0 & 0 & -\frac{\sin\theta}{\cos\theta} \end{pmatrix} \mathbf{x}. \tag{28}$$

This matrix is diagonal and it trivially commutes with $A_z$. The price we had to pay to realize a system of commuting matrices is that in $\widetilde{\rho}$ the flow parameter $\theta$ appear non-linearly, thus we traded-off commutativity at the level of the Lie algebra matrices for the linearity of the flow parameters at the group level. We remark that both give rise to the same differential operator on $X$, which is the relevant object for our purposes.

## A.4   $X = \mathbb{R}^{3N}$ AND GLOBAL SO(3)

Let $X = \mathbb{R}^{3N}$ be parametrized by $\mathbf{x}_{i=1,\dots,N}$. We can describe a global SO(3) action as follows

$$\mathbf{x}_1 = R_{\mathbf{e}_z}(\varphi_1)R_{\mathbf{e}_y}(\theta_1) \begin{pmatrix} 0 \\ 0 \\ \widehat{z}_1 \end{pmatrix},$$

$$\mathbf{x}_2 = R_{\mathbf{e}_z}(\varphi_1)R_{\mathbf{e}_y}(\theta_1)R_{\mathbf{e}_z}(\varphi_2) \begin{pmatrix} \widehat{x}_2 \\ 0 \\ \widehat{z}_2 \end{pmatrix},$$

$$\mathbf{x}_{\mu=3,\dots,N} = R_{\mathbf{e}_z}(\varphi_1)R_{\mathbf{e}_y}(\theta_1)R_{\mathbf{e}_z}(\varphi_2)\widehat{\mathbf{x}}_\mu, \tag{29}$$

where $R_{\mathbf{a}}(\omega)$ represents a rotation of an angle $\omega$ around the axis $\mathbf{a}$. We can then derive the operator $\mathbf{\Pi} \in \mathbb{R}^{3N \times 3N}$ as follows. Let $R'(\omega)$ be the matrix where we take the partial derivative with respect to $\omega$ of all elements of $R$. Then

$$\mathbf{\Pi}_{\varphi_1} = \left( [A_z\mathbf{x}_1]^\top \quad [A_z\mathbf{x}_2]^\top \quad \cdots \quad [A_z\mathbf{x}_N]^\top \right)^\top$$

$$\mathbf{\Pi}_{\theta_1} = \begin{pmatrix} (\cos\varphi_1 A_y - \sin\varphi_1 A_x)\mathbf{x}_1 \\ (\cos\varphi_1 A_y - \sin\varphi_1 A_x)\mathbf{x}_2 \\ \vdots \\ (\cos\varphi_1 A_y - \sin\varphi_1 A_x)\mathbf{x}_N \end{pmatrix}$$

$$\mathbf{\Pi}_{\varphi_2} = \begin{pmatrix} \mathbf{0} \\ (\sin\theta_1\cos\varphi_1 A_x + \sin\theta_1\sin\varphi_1 A_y + \cos\theta_1 A_z)\mathbf{x}_2 \\ \vdots \\ (\sin\theta_1\cos\varphi_1 A_x + \sin\theta_1\sin\varphi_1 A_y + \cos\theta_1 A_z)\mathbf{x}_N \end{pmatrix} \tag{30}$$

Notice that these do represent global rotations since it is easy to see that $(\sin\theta_1\cos\varphi_1 A_x + \sin\theta_1\sin\varphi_1 A_y + \cos\theta_1 A_z)\mathbf{x}_1 = \mathbf{0}$. Formally, the true Lie algebra elements are $3\times 3$ matrices of the form

$$
A_\varphi = \begin{pmatrix} A_z & 0 & 0 & \cdots & 0 \\ 0 & A_z & 0 & \cdots & 0 \\ 0 & 0 & A_z & \cdots & 0 \\ \vdots & \vdots & \vdots & \ddots & \vdots \\ 0 & 0 & 0 & \cdots & A_z \end{pmatrix}
\tag{31}
$$

and similarly for the other operators. Now, for the inverse relations we have

$$
\theta_1 = \arccos\frac{z_1}{(x_1^2 + y_1^2 + z_1^2)^{1/2}}\,,
$$
$$
\varphi_1 = \mathrm{sgn}(y_1)\arccos\frac{x_1}{(x_1^2 + y_1^2)^{1/2}}\,,
$$
$$
\varphi_2 = \arctan\frac{\widetilde{y}_2}{\widetilde{x}_2}\,,
\tag{32}
$$

where $\widetilde{\mathbf{x}}_2 = R_{\mathbf{e}_y}(\theta_1)^{-1}R_{\mathbf{e}_z}(\varphi_1)^{-1}\mathbf{x}_2 = R_{\mathbf{e}_y}(-\theta_1)R_{\mathbf{e}_z}(-\varphi_1)\mathbf{x}_2$.

## A.5 $X = \mathbb{R}^4, G = \mathrm{SO}(4)\times\mathbb{R}_+$

Now we look at the case of a higher dimensional Lie group, namely $G = \mathrm{SO}(4)\times\mathbb{R}_+$. The parametrization is given by

$$
x_1 = e^r\cos\varphi_1\,,
$$
$$
x_2 = e^r\sin\varphi_1\cos\varphi_2\,,
$$
$$
x_3 = e^r\sin\varphi_1\sin\varphi_2\cos\varphi_3\,,
$$
$$
x_4 = e^r\sin\varphi_1\sin\varphi_2\sin\varphi_3\,.
\tag{33}
$$

The Lie algebra elements corresponding to the $\mathrm{SO}(4)$ flow coordinates are

$$
A_{\varphi_1} = \begin{pmatrix} 0 & -\cos\varphi_2 & -\sin\varphi_2\cos\varphi_3 & -\sin\varphi_2\sin\varphi_3 \\ \cos\varphi_2 & 0 & 0 & 0 \\ \sin\varphi_2\cos\varphi_3 & 0 & 0 & 0 \\ \sin\varphi_2\sin\varphi_3 & 0 & 0 & 0 \end{pmatrix}\,,
$$
$$
A_{\varphi_2} = \begin{pmatrix} 0 & 0 & 0 & 0 \\ 0 & 0 & -\cos\varphi_3 & -\sin\varphi_3 \\ 0 & \cos\varphi_3 & 0 & 0 \\ 0 & \sin\varphi_3 & 0 & 0 \end{pmatrix}\,,
$$
$$
A_{\varphi_3} = \begin{pmatrix} 0 & 0 & 0 & 0 \\ 0 & 0 & 0 & 0 \\ 0 & 0 & 0 & -1 \\ 0 & 0 & 1 & 0 \end{pmatrix}\,.
\tag{34}
$$

### A.6 $G = \mathrm{SO}(N)$

We present here the formalism for the $G = \mathrm{SO}(N)$ for any $N \geq 4$. The parametrization is given by (Blumenson, 1960)

$$
\begin{aligned}
x_1 &= e^r \cos \varphi_1 \,, \\
x_2 &= e^r \sin \varphi_1 \cos \varphi_2 \,, \\
x_3 &= e^r \sin \varphi_1 \sin \varphi_2 \cos \varphi_3 \,, \\
&\vdots \\
x_j &= e^r \sin \varphi_1 \sin \varphi_2 \sin \varphi_3 \cdots \sin \varphi_{j-1} \cos \varphi_j \,, \\
&\vdots \\
x_{n-1} &= e^r \sin \varphi_1 \sin \varphi_2 \sin \varphi_3 \cdots \sin \varphi_{n-2} \cos \varphi_{n-1} \,, \\
x_n &= e^r \sin \varphi_1 \sin \varphi_2 \sin \varphi_3 \cdots \sin \varphi_{n-2} \sin \varphi_{n-1} \,.
\end{aligned}
\tag{35}
$$

The corresponding Lie algebra elements are given by

$$
A_{\varphi_{n-1}} = \begin{pmatrix} 0 & \cdots & 0 & 0 \\ 0 & \ddots & \vdots & \vdots \\ 0 & \cdots & 0 & -1 \\ 0 & \cdots & 1 & 0 \end{pmatrix} \,,\quad
A_{\varphi_{n-2}} = \begin{pmatrix} 0 & \cdots & & 0 & 0 & 0 \\ 0 & \ddots & & \vdots & \vdots & \vdots \\ 0 & \cdots & & 0 & -\cos\varphi_{n-1} & -\sin\varphi_{n-1} \\ 0 & \cdots & \cos\varphi_{n-1} & 0 & 0 \\ 0 & \cdots & \sin\varphi_{n-1} & 0 & 0 \end{pmatrix} \,,
$$

$$
A_{\varphi_{n-3}} = \begin{pmatrix} 0 & \cdots & & 0 & 0 & 0 & 0 \\ 0 & \ddots & & \vdots & \vdots & \vdots & \vdots \\ 0 & \cdots & & 0 & -\cos\varphi_{n-2} & -\sin\varphi_{n-2}\cos\varphi_{n-1} & -\sin\varphi_{n-2}\sin\varphi_{n-1} \\ 0 & \cdots & \cos\varphi_{n-2} & 0 & 0 & 0 \\ 0 & \cdots & \sin\varphi_{n-2}\cos\varphi_{n-1} & 0 & 0 & 0 \\ 0 & \cdots & \sin\varphi_{n-2}\sin\varphi_{n-1} & 0 & 0 & 0 \end{pmatrix} \,,
$$

$$
\vdots
$$

$$
A_{\varphi_j} = \frac{1}{x_j} \begin{pmatrix} 0 & \cdots & 0 & 0 & 0 & \cdots & 0 \\ 0 & \ddots & \vdots & \vdots & \vdots & \vdots & \vdots \\ 0 & \cdots & 0 & -x_{j+1} & -x_{j+2} & \cdots & -x_n \\ 0 & \cdots & x_{j+1} & 0 & 0 & \cdots & 0 \\ 0 & \cdots & x_{j+2} & 0 & 0 & \cdots & 0 \\ 0 & \cdots & \vdots & \vdots & \vdots & \ddots & \vdots \\ 0 & \cdots & x_n & 0 & 0 & \cdots & 0 \end{pmatrix} \,,
$$

$$
\vdots
$$

$$
A_{\varphi_1} = \frac{1}{x_1} \begin{pmatrix} 0 & -x_2 & -x_3 & -x_4 & \cdots & -x_n \\ x_2 & \ddots & \vdots & \vdots & \vdots & \vdots \\ x_3 & \cdots & 0 & 0 & 0 & 0 \\ x_4 & \cdots & 0 & 0 & 0 & 0 \\ \vdots & \cdots & 0 & 0 & 0 & 0 \\ x_n & \cdots & 0 & 0 & 0 & 0 \end{pmatrix} \,.
\tag{36}
$$

## B  Proofs of condition for suitable Lie group

Here we provide the statements with proofs of the results in Section 2.2.

**Proposition B.1.** *The linear operator induced by $\Pi$ is complete if $\Pi$ is the local frame of a vector bundle $E$ over $X$ whose rank is $n \geq \dim X$ almost everywhere. If rank $E = n$ everywhere, then $E = TX$, the tangent bundle of $X$.*

*Proof.* We start by noting that, given the expression of the fundamental fields as derivations, we can write $\mathcal{L}(\mathbf{x}) = \mathbf{\Pi}(\mathbf{x})^\top \nabla$. Let $\pi : E \to X$ be the projection map, then rank $\pi^{-1}(\mathbf{x}) = \min(\text{rank } \mathbf{\Pi}(\mathbf{x}), n)$, since rank $\nabla = n$. Now, consider $\mathcal{L} \log p(\mathbf{x}) = \mathcal{L} \log q(\mathbf{x})$, which implies $\mathcal{L} \log \frac{p(\mathbf{x})}{q(\mathbf{x})} = 0$. Let $U \subseteq X$ such that rank $\mathbf{\Pi} \geq n \; \forall \mathbf{x} \in U$, and by assumption $X \setminus U$ has measure zero. Then the above holds if and only if $\nabla \log \frac{p(\mathbf{x})}{q(\mathbf{x})} = 0$, which implies $\frac{p(\mathbf{x})}{q(\mathbf{x})} = c$, constant $\forall \mathbf{x} \in U$. Now, $p(\mathbf{x})$ and $q(\mathbf{x})$ are probability densities by assumption, thus $c = 1$, which proves the claim. $\qquad\square$

**Proposition B.2.** *The operator $\mathbf{\Pi}$ induced by $\mathfrak{g}$ is complete if and only if the subspace $U \subseteq X$ such that $\dim \frac{G}{G_\mathbf{x}} < n$ for $\mathbf{x} \in U$, where $n = \dim X$, has measure zero in $X$.*

*Proof.* First, we recall that the dimension of an orbit $\mathcal{O}_\mathbf{x}$ of $\mathbf{x} \in X$ equals the dimension of the image of the map $d\rho_\mathbf{x} : \mathfrak{g} \to T_\mathbf{x} X : \mathbf{A} \mapsto \mathbf{\Pi}(\mathbf{x})$. Suppose first that $\mathbf{\Pi}$ is complete. Then, from Proposition B.1 the rank of $\mathbf{\Pi}(\mathbf{x})$ is $\geq n$ almost everywhere, and therefore $\dim G/G_\mathbf{x} \geq n$ almost everywhere, which implies one direction of the claim. The reverse is quite straightforward. Assume that the rank of $\mathbf{\Pi}(\mathbf{x})$ is $\geq n$ almost everywhere. As $\mathbf{\Pi}$ represent the action of the infinitesiamal transformations of $G$, it means that locally $G$ cannot fix points in $X$, thus proving the claim. $\qquad\square$

## C    PROOF OF MAIN THEOREM

Here we provide the full proof of Theorem 3.1:

**Theorem C.1.** *Let $G$ be a Lie group acting on $X$ satisfying the conditions of Section 2.2, and let $\mathfrak{g}$ be its Lie algebra. The pair of SDEs*

$$d\mathbf{x} = \left[\beta(t)\mathbf{\Pi}(\mathbf{x})\mathbf{f}(\mathbf{x}) + \frac{\gamma(t)^2}{2}\rho_X(\Omega)\right] dt + \gamma(t)\mathbf{\Pi}(\mathbf{x})d\mathbf{W} \;, \tag{37}$$

$$d\mathbf{x} = \Big[\beta(t)\mathbf{\Pi}(\mathbf{x})\mathbf{f}(\mathbf{x}) - \frac{\gamma^2(t)}{2}\rho_X(\Omega) - \gamma^2(t)\mathbf{\Pi}(\mathbf{x})\nabla^\top \cdot \mathbf{\Pi}(\mathbf{x})$$
$$-\gamma(t)^2\mathbf{\Pi}(\mathbf{x})\mathcal{L}\log p_t(\mathbf{x})\Big] dt + \gamma(t)\mathbf{\Pi}(\mathbf{x})d\mathbf{W} \;, \tag{38}$$

*where $\beta, \gamma : \mathbb{R} \to \mathbb{R}$ are time-dependent functions, $\mathbf{\Pi} : \mathbb{R}^n \to \mathbb{R}^{n \times n}$ the fundamental vector fields, $\mathbf{f} : \mathbb{R}^n \to \mathbb{R}^n$ the drift, $\Omega = \sum_i A_i^2$ is the quadratic Casimir element of $\mathfrak{g}$, and $\mathcal{L} = \mathbf{\Pi}(\mathbf{x})^\top \nabla$ is such that*

1. *The forward-time SDE (37) is exactly solvable, with solution*

$$\mathbf{x}(t) = \left(\prod_i O_i(\tau_i(t))\right)\mathbf{x}(0) = \left(\prod_{i=1}^n e^{\tau_i(t)A_i}\right)\mathbf{x}(0) \;, \tag{39}$$

   *where $O_i = e^{\tau_i(t)A_i}$ is the finite group action and $\boldsymbol{\tau}(t)$ is the solution to the SDE*

$$d\boldsymbol{\tau}(\mathbf{x}) = \beta(t)\mathbf{f}(\mathbf{x})dt + \gamma(t)d\mathbf{W} \;. \tag{40}$$

2. *The SDE (7) is the reverse-time process of (6).*

3. *The Langevin dynamic of the above SDEs decomposes as a direct sum of $\mathfrak{g}$ infinitesimal actions (1), each defining an infinitesimal transformation along the flows $\xi_{\boldsymbol{\tau}}$.*

*Proof.* We start by proving 3. We start by rewriting (37) in terms of the fundamental flow coordinates $\tau_i = \xi_{A_i}^{-1}(\mathbf{x}_0)(\mathbf{x}) : X \to \mathbb{R}$. For this we employ Itô's Lemma for the multivariate case: given the SDE (37) and a transformation $\boldsymbol{\tau}(\mathbf{x})$, it is given by

$$d\boldsymbol{\tau}(\mathbf{x}) = (\nabla_\mathbf{x}\boldsymbol{\tau})^\top \left[\beta(t)\mathbf{\Pi}(\mathbf{x})\mathbf{f}(\mathbf{x}) + \frac{\gamma^2(t)}{2}\rho_X(\Omega)\right] dt + \frac{\gamma^2(t)}{2}\text{Tr}\left[\mathbf{\Pi}(\mathbf{x})^\top\left(H_\mathbf{x}\boldsymbol{\tau}\right)\mathbf{\Pi}(\mathbf{x})\right] dt$$
$$+ \gamma(t)(\nabla_\mathbf{x}\boldsymbol{\tau})^\top\mathbf{\Pi}(\mathbf{x})d\mathbf{W}$$
$$= \beta(t)\mathbf{f}(\mathbf{x}) + \frac{\gamma^2(t)}{2}\left[(\nabla_\mathbf{x}\boldsymbol{\tau})^\top\Delta_{\boldsymbol{\tau}}\mathbf{x} + \text{Tr}\left[\mathbf{\Pi}(\mathbf{x})^\top\left(H_\mathbf{x}\boldsymbol{\tau}\right)\mathbf{\Pi}(\mathbf{x})\right]\right] dt + \gamma(t)d\mathbf{W} \tag{41}$$

since $\nabla_{\mathbf{x}}\boldsymbol{\tau} = \mathbf{\Pi}^{-1}(\mathbf{x})$ as matrices. Now, the second term can be rewritten in components as

$$\left\{(\nabla_{\mathbf{x}}\boldsymbol{\tau})^{\top}\Delta_{\boldsymbol{\tau}}\mathbf{x} + \mathrm{Tr}\left[\mathbf{\Pi}(\mathbf{x})^{\top}\left(H_{\mathbf{x}}\tau\right)\mathbf{\Pi}(\mathbf{x})\right]\right\}_{k,l}$$

$$= \sum_i \sum_j \frac{\partial x_j}{\partial \tau_k}\left(\frac{\partial}{\partial x_j}\frac{\partial x_i}{\partial \tau_l}\right)\frac{\partial \boldsymbol{\tau}}{\partial x_i} + \sum_i \sum_j \frac{\partial x_j}{\partial \tau_k}\frac{\partial x_i}{\partial \tau_l}\frac{\partial^2 \boldsymbol{\tau}}{\partial x_i \partial x_j}$$

$$= \sum_j \frac{\partial x_j}{\partial \tau_k}\frac{\partial}{\partial x_j}\left(\sum_i \frac{\partial x_i}{\partial \tau_l}\frac{\partial \boldsymbol{\tau}}{\partial x_i}\right)$$

$$= \frac{\partial}{\partial \tau_k}\left(\sum_i \frac{\partial x_i}{\partial \tau_l}\frac{\partial \boldsymbol{\tau}}{\partial x_i}\right)$$

$$= \frac{\partial}{\partial \tau_k}\frac{\partial \boldsymbol{\tau}}{\partial \tau_l}$$

$$= H_{\boldsymbol{\tau}}\boldsymbol{\tau}\ , \tag{42}$$

which vanishes. Thus we proved that

$$d\boldsymbol{\tau}(\mathbf{x}) = \beta(t)\mathbf{f}(\mathbf{x})dt + \gamma(t)d\mathbf{W}\ , \tag{43}$$

and provided that is chosen so that $f_i(\mathbf{x}(\boldsymbol{\tau})) = f_i(\tau_i)$, this corresponds to a system of independent SDEs, as claimed.

Now, to prove 1, let $\boldsymbol{\tau}(t)$ be a solution to (43) and $\mathbf{x}(t)$ as in (8). Then a Taylor expansion yields

$$\mathbf{x}(t) = \left[I + \sum_i \tau_i(t)A_i + \frac{1}{2}\left(\sum_i \tau_i(t)A_i\right)^2 + \mathcal{O}(\tau_i^3)\right]\mathbf{x}(0) \tag{44}$$

since $[A_i, A_j] = 0$ and where $\mathcal{O}(\tau_i^3)$ represents terms of third order in $\tau_i$'s. Then taking the differential and dropping higher order terms

$$d\mathbf{x}(t) = \left[\sum_i d\tau_i(t)A_i + \frac{1}{2}\left(\sum_i d\tau_i(t)A_i\right)^2\right]\mathbf{x}(0)$$

$$= \left[\sum_i \left[\beta(t)\mathbf{f}(\mathbf{x})dt + \gamma(t)d\mathbf{W}\right]A_i + \frac{1}{2}\left(\sum_i \left[\beta(t)\mathbf{f}(\mathbf{x})dt + \gamma(t)d\mathbf{W}A_i\right]\right)^2\right]\mathbf{x}(0)$$

$$= \left[\beta(t)\mathbf{\Pi}(\mathbf{x})\mathbf{f}(\mathbf{x})dt + \gamma(t)\mathbf{\Pi}(t)d\mathbf{W}\right] + \frac{1}{2}\left(\sum_i \gamma(t)d\mathbf{W}A_i\right)^2\mathbf{x}(0)$$

$$= \left[\beta(t)\mathbf{\Pi}(\mathbf{x})\mathbf{f}(\mathbf{x})dt + \gamma(t)\mathbf{\Pi}(t)d\mathbf{W}\right] + \frac{\gamma(t)^2}{2}\left(\sum_i A_i^2 dt\right)\mathbf{x}(0)$$

$$= \left[\beta(t)\mathbf{\Pi}(\mathbf{x})\mathbf{f}(\mathbf{x}) + \frac{\gamma(t)^2}{2}\rho_X(\Omega)\right]dt + \gamma(t)\mathbf{\Pi}(\mathbf{x})d\mathbf{W}\ , \tag{45}$$

which in the forward SDE (6), proving our claim, where we used the relations $dW_i^2 = dt$ and $dW_i dW_j = 0$ for $j \neq i$.

Finally, we prove 2. To do this it suffices to apply Anderson's result (Anderson, 1982)

$$d\mathbf{x} = \left[\beta(t)\mathbf{\Pi}_i(\mathbf{x})\mathbf{f}(\mathbf{x}) + \frac{\gamma^2(t)}{2}\rho_X(\Omega) - \gamma^2(t)\nabla\cdot(\mathbf{\Pi}(\mathbf{x})\mathbf{\Pi}(\mathbf{x})^{\top})\right.$$

$$\left. -\gamma(t)^2\mathbf{\Pi}(\mathbf{x})\mathbf{\Pi}(\mathbf{x})^{\top}\nabla_{\mathbf{x}}\log p_t(\mathbf{x})\right]dt + \gamma(t)\mathbf{\Pi}(\mathbf{x})d\mathbf{W}_i\ , \tag{46}$$

and note that $\mathbf{\Pi}(\mathbf{x})^\top \nabla_\mathbf{x} = \mathcal{L}$, the generalized score, and

$$\left[\nabla_\mathbf{x} \cdot (\mathbf{\Pi}(\mathbf{x})\mathbf{\Pi}(\mathbf{x})^\top)\right]_i = \frac{\partial}{\partial x_k}\left(\Pi_{ij}\Pi_{kj}\right)$$

$$= \frac{\partial}{\partial x_k}\left(\Pi_{ij}\right)\Pi_{kj} + \Pi_{ij}\frac{\partial}{\partial x_k}\Pi_{kj}$$

$$= \frac{\partial x_k}{\partial \tau_j}\frac{\partial}{\partial x_k}\left(\frac{\partial x_i}{\partial \tau_j}\right) + \Pi_{ij}[\nabla^\top \cdot \mathbf{\Pi}(\mathbf{x})]_j$$

$$= \frac{\partial}{\partial \tau_j}\left(\frac{\partial x_i}{\partial \tau_j}\right) + \Pi_{ij}[\nabla^\top \cdot \mathbf{\Pi}(\mathbf{x})]_j$$

$$= [\text{Tr}\, H_{\boldsymbol{\tau}}(\mathbf{x})]_i + \Pi_{ij}[\nabla^\top \cdot \mathbf{\Pi}(\mathbf{x})]_j \qquad (47)$$

where we recall that the divergence of a matrix is a vector whose components are the divergence of its rows. Recalling the relationship between the trace of the Hessian and the Laplacian we can write in operator form

$$\nabla_\mathbf{x} \cdot (\mathbf{\Pi}(\mathbf{x})\mathbf{\Pi}(\mathbf{x})^\top) = \mathbf{\Pi}(\mathbf{x})\nabla^\top \cdot \mathbf{\Pi}(\mathbf{x}) + \rho_X(\Omega)\,, \qquad (48)$$

Plugging this back in into the previous expression we obtain our claim

$$d\mathbf{x} = \left[\beta(t)\mathbf{\Pi}(\mathbf{x})\mathbf{f}(\mathbf{x}) - \frac{\gamma^2(t)}{2}\rho_X(\Omega) - \gamma^2(t)\mathbf{\Pi}(\mathbf{x})\nabla^\top \cdot \mathbf{\Pi}(\mathbf{x})\right.$$

$$\left. -\gamma(t)^2\mathbf{\Pi}(\mathbf{x})\mathcal{L}\log p_t(\mathbf{x})\right]dt + \gamma(t)\mathbf{\Pi}(\mathbf{x})d\mathbf{W}\,. \qquad (49)$$

$\square$

# D   LIE GROUP-INDUCED FLOW MATCHING MODELING

We briefly summarize the formalism of flow matching. Given a target distribution $p_0(\mathbf{x})$ and a vector field $u_t$ generating the distribution $p_t(\mathbf{x})$, i.e., if it satisfies $p_t(\mathbf{x}) = [u_t]_* p_0(\mathbf{x})$ where $[u_t]_*$ is the push-forward map, the flow matching objective is defined as

$$\mathcal{L}_{\text{FM}}(\theta) = \mathbb{E}_{t,\mathbf{x}_t \sim p_t(\mathbf{x})} \left| v_{t;\theta}(\mathbf{x}_t) - u_t(\mathbf{x}_t) \right|^2 . \tag{50}$$

Marginalizing over samples $\mathbf{x}_0 \sim p_0(\mathbf{x})$ we obtain the conditional flow matching objective

$$\mathcal{L}_{\text{CFM}}(\theta) = \mathbb{E}_{t,\mathbf{x}_0 \sim p_0(\mathbf{x}), \mathbf{x}_t \sim p_t(\mathbf{x}|\mathbf{x}_0)} \left| v_{t;\theta}(\mathbf{x}_t) - u_t(\mathbf{x}_t|\mathbf{x}_0) \right|^2 . \tag{51}$$

Now, under the assumptions for learning the generalized score through the objective 11 we have that $p_t(\boldsymbol{\tau}(\mathbf{x})) = \mathcal{N}(\boldsymbol{\tau}|\boldsymbol{\mu}(\boldsymbol{\tau}(0), t), \boldsymbol{\Sigma}(t))$, where $\boldsymbol{\tau}(0) = \mathbf{x}(\boldsymbol{\tau})(0)$. Then the solution of the SDE from Theorem 3.1

$$\mathbf{x}(t) = \left( \prod_{i=1}^{n} e^{\tau_i(t) A_i} \right) \mathbf{x}(0) , \tag{52}$$

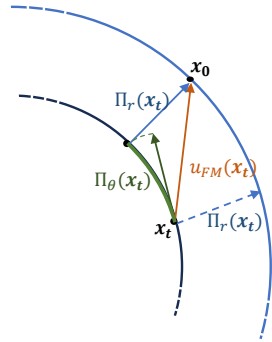

Figure 10: $\mathfrak{so}(2)$ (green and blue) vs. $\mathfrak{t}(2)$ (orange) induced flows.

is a flow inducing the distribution $p_t(\boldsymbol{\tau}(\mathbf{x}))$. Thus, the vector field that generates the conditional probability path is obtained by differentiating the path above with respect to $t$, yielding

$$
\begin{aligned}
u_t(\mathbf{x}(t)|\mathbf{x}(0)) = \frac{d}{dt}\mathbf{x}(t) &= \sum_i \frac{\partial \mathbf{x}(t)}{\partial \tau_i} \frac{\partial \tau_i}{\partial t} \\
&= \sum_i A_i \mathbf{x}(t) \left( \mu'_{t,i}(\tau_i(0)) + \sigma'_t(\tau_i(0))\eta_i \right) \\
&= \boldsymbol{\Pi}(\mathbf{x}(t)) \left[ \boldsymbol{\mu}'_t(\tau_i(0)) + \frac{\sigma'_t(\tau_i(0))}{\sigma_t(\tau_i(0))} \left( \boldsymbol{\tau}(t) - \boldsymbol{\mu}_t(\boldsymbol{\tau}(0)) \right) \right] ,
\end{aligned}
\tag{53}
$$

where we used the fact that

$$\boldsymbol{\tau}(t) = \mu_t(\boldsymbol{\tau}(0)) + \sigma_t(\boldsymbol{\tau}(0))\boldsymbol{\eta} , \tag{54}$$

where $\boldsymbol{\eta} \sim \mathcal{N}(\mathbf{0}, \mathbf{1})$. Thus, we see that the unique vector field that defines the flow (8) is again proportional to the fundamental vector field $\boldsymbol{\Pi}(\mathbf{x})$ of the Lie algebra $\mathfrak{g}$ of $G$. In figure 10 we illustrate the flow generated by our formalism in the case of $SO(2)$ in comparison with the traditional flow matching of $T(2)$. The orange path depicts the linear (in Euclidean metric) displacement given by the traditional flow matching, assuming $G = T(2)$. In green and blue we depicted the orbits trajectories resulting from generalized flow matching with $G = SO(2) \times \mathbb{R}_+$. Although the start and end points are the same, the path is decomposed into transformations along the orbits of the two group factors. This is particularly useful when these correspond to meaningful degrees of freedom in the system. For example, when flowing between conformers of the same molecule, the intermediate states produced by traditional flow matching are often unphysical, as they involve linear interpolation between the Cartesian coordinates of the atoms. However, generalized score matching, following the degrees of freedom given by bond and torsion angles as described in Section 3.1, would not only yield efficient learning but also produce chemically meaningful intermediate states, as the path is broken down into updates of chemically relevant degrees of freedom.

# E   EXPERIMENTS

In this final section we present some further details regarding our experiment in Section 5. We provide the code to replicate our experiments in the Supplementary Information (SI). Following publication we will open-source our code.

### E.1 MNIST

We parametrize the noising process through the SDE

$$d\boldsymbol{\tau} = \sqrt{\beta(t)}d\mathbf{W} \,, \tag{55}$$

where we set the drift term to zero. Notice that this choice is consistent with a 2d-rotation of a function over the grid $\mathbf{x}_{i,j}$, given by $f(\mathbf{x}_{i,j}) = f_{i,j}$, denoting the value of the pixel of image $f$ at the location $i, j$. We train a convolutional neural network (CNN) with three convolutional layers followed by fully connected layers that outputs a single value, being the score for the flow coordinate $\tau$. For the specific details of the implementation we refer to the code-base in the SI. In sampling, we apply a smoothing function to compensate interpolation artifacts due to rotations on a discretized grid. We choose $T = 100$ time-steps in training but only need $T = 10$ time-steps during sampling.

#### E.1.1 BBDM

We implement the Brownian Bridge Diffusion Model (BBDM) (Li et al., 2023) and train it on the rotated MNIST dataset. The BBDM operates on the full pixel space $\mathbb{R}^{784}$ of the $28 \times 28$ MNIST digits and indicates a continuous time stochastic process conditioned on the starting $\mathbf{x}(0)$ and end point $\mathbf{x}(T)$ which are pinned together as paired data. In this case, we assume $\mathbf{x}(T) \sim p(\mathbf{x}_T)$ to be a randomly augmented MNIST digit obtained from an original MNIST digit $\mathbf{x}(0)$. During training, we sample an intermediate point $\mathbf{x}(t) \sim N(x_t|\mu_t(x(0), x(T)), \Sigma_t)$ where the mean function $\mu_t(t)(\mathbf{x}(0), \mathbf{x}(T))$ is a linear interpolation between the endpoints $(\mathbf{x}(0), \mathbf{x}(T))$ and use the score-network to predict the original data point $\widehat{\mathbf{x}(0)} = s_\theta(x_t, t, x_T)$ as opposed to the noise or difference paramterization proposed in the original BBDM paper. We noticed that predicting the original data point led to better sampling quality including the inductive bias that MNIST digits are represented as binary tokens. Furthermore, we observe that the sampling quality is also better when the prior image $x_T$ is input as context into the score network, enforcing a stronger signal throughout the trajectory. As opposed to our model, we trained the BBDM on $T = 1000$ diffusion timesteps using the sin-scheduler from BBDM.

### E.2 QM9 & CROSSDOCKED2020

**QM9** The conformer generation tasks is about learning a conditional probabilistic map $\mathbf{x} \sim p_\theta(X|M)$, where $\mathbf{x} \in \mathbb{R}^{3N}$ for a molecule with $N$ atoms. We implement a variant of EQGAT (Le et al., 2022b) as neural network architecture where input features for the nodes consist of atom types and atomic coordinates, while edge features are encoded to indicate the existence of a single-, double, triple or aromatic bond based on the adjacency matrix. We use $L = 5$ message passing layers with $s_{\text{dim}} = 128$ , $v_{\text{dim}} = 64$ scalar and vector features, respectively. To predict the scores for each atom, we concatenate the hidden scalar and vector embeddings $s \in \mathbb{R}^{128}$ , $v \in \mathbb{R}^{3\times64}$ into one output embedding $o = \mathbb{R}^{128+3*64}$ which is further processed by a 2-layer MLP with three output units. Notice that the predicted scores per atom are neither invariant nor equivariant since the scalar and vector features are transformed with an MLP.

We choose the drift $f$ with its scaling $\beta$ and the diffusion coefficients $\gamma$ in such way that the forward SDE for the flow coordinates $\boldsymbol{\tau}$ in (9) has the expression

$$d\boldsymbol{\tau} = -\frac{1}{2}\beta(t)\boldsymbol{\tau}dt + \sqrt{\beta(t)}d\mathbf{W} \,, \tag{56}$$

where for clarity we have omitted the dependency between the flow coordinates and the original data in Cartesian coordinates, i.e. $\boldsymbol{\tau}(\mathbf{x})$, since the coordinate transformations with Lie algebra representation are described in A.3. The forward SDE in (56) is commonly known as *variance-preserving* SDE (Song et al., 2020b). We use the cosine scheduler proposed by Dhariwal & Nichol (2021) and $T = 100$ diffusion timesteps.

**CrossDocked** For this experiment we adopt again an SDE of the form (56) for the three SO(3) flow coordinates $\theta_1, \varphi_1, \varphi_2$ and the three $T(3)$ center of mass Cartesian flow coordinates. The SO(3) flow coordinates are always computed and applied in the ligand center of mass. In this way there is no ambiguity regarding the non-commutativity of SE(3), as rotation around the origin commute with translations of the system. We train a variant of EQGAT as in the QM9 case, but now including

also node and edge features of the protein pocket. Specifically, the adjacency matrix for the GNN is computed dynamically at each time step, according to the relative distance between ligand and protein. For this, we choose a cut-off of 5 Å. We also use in this experiment a cosine scheduler and $T = 100$ diffusion timesteps. Since this learning problem is 6-dimensional, we aggregate the last layer's node embeddings from the ligand atoms into a global representation through summation. This embedding is fed as input into a 2-layer MLP to predict the six scores.

### E.2.1   RSGM ON CROSSDOCKED

We utilized the framework of Riemannian Score-Based Generative Models (RSGM) by (De Bortoli et al., 2022) to model rigid-body motions on $G = (\mathrm{SO}(3) \times T(3))$, in similar fashion to (Corso et al., 2023; Yim et al., 2023) by choosing a variance exploding SDE for the rotation dynamics and variance preserving SDE for the global translations. The terminal distribution for the rotation is designed to converge to an isotropic Gaussian distribution on SO(3) (Leach et al., 2022), while the terminal distribution for the translation component converges to an isotropic Gaussian in $\mathbb{R}^3$. To obtain the tractable scores for rotation and translation, we use the code by the authors from DiffDock and SE(3)-Diffusion for Protein Backbone Modeling in `https://github.com/gcorso/DiffDock/blob/main/utils/so3.py` and `https://github.com/jasonkyuyim/se3_diffusion/blob/master/data/se3_diffuser.py` and make sure that the score outputs for rotation and translation are SO(3) equivariant using the same EQGAT model architecture. The (variance-preserving) scheduler for the translation dynamics is chosen in similar fashion to our experiment using the cosine scheduler, while the (variance-exploding) scheduler for the rotation dynamics is implemented as an linear increasing sequence in $\log_{10}$ space with $\sigma_{min} = 0.001$ and $\sigma_{max} = 2.0$ and $T = 100$ discretized diffusion steps as $\sigma(t) = 10^t$ for $t \in (\log_{10}(\sigma_{min}), \log_{10}(\sigma_{max}))$.

To compare both modeling approaches with respect to the dynamics using the same network architecture, we perform 5 dockings per protein-ligand complex in the CrossDocked test dataset comprising 100 complexes and compute the mean RMSD between ground-truth coordinates and predicted coordinates.

## F   FURTHER OUTLOOK

In the context of generative chemistry, particularly for modeling interactions within protein pockets, our methods could be employed to decouple the intrinsic generation of ligands from the global transformations required to fit the ligand into the pocket. This approach can also be extended beyond 3D coordinates, for instance, by working with higher-order representations, such as modeling electron density (Rackers et al., 2023).

Moreover, for more complex problems, it is feasible that an optimal generation process can be achieved by combining different choices of $G$ along the trajectory. In the context of ligand generation, we propose a **time-dependent group action** $G_t = tT(3N) + (1 - t)(\mathrm{SO}(3) \times \mathbb{R}_+)^N$: at the beginning of the diffusion process, when the point cloud is still far from forming a recognizable conformer, we can leverage the properties of a true Gaussian prior. As the point cloud is gradually optimized to "resemble a molecule", we progressively transition to a generalized score-guided process. This shift allows us to fine-tune chemically relevant properties, such as bonds and torsion angles, ensuring that the intermediate and final conformers are chemically valid and accurate. This will be the focus of our forthcoming work.

