# OpenReview forum: "Lie Group-Induced Dynamics in Score-Based Generative Modeling"
_ICLR.cc/2025/Conference — Submitted to ICLR 2025_

### Official Review · Reviewer_hBkJ · 2024-10-29

**Soundness:** 3
**Presentation:** 2
**Contribution:** 2
**Rating:** 5
**Confidence:** 3

**Summary:**

In this paper the authors propose a new diffusion model framework to disentangle symmetries in the data. Assuming that there is a group action (where the group action is given by a Lie group) associated with the space of interest, they build a diffusion model which leverages the action of the group on the data space. This class of diffusion model is valid under certain assumptions on the relationship between the group action and the data space. For instance, one must assume that the group action is complete. In that case, the forward diffusion is linked with an (explicit) forward diffusion on the Lie group. In order to learn the quantity appearing in the backward diffusion, they propose a generalized score loss function, which in practice corresponds to a preconditioning of the score function. The proposed diffusion model is then tested on rotated image data, small molecule dataset (QM9) and molecular docking.

**Strengths:**

* I appreciate the rigor of the paper (with some caveats, see the Weaknesses section). Most of the quantities used in the paper are correctly introduced and I found the paper mostly easy to follow.

* I think the proposed forward diffusion is interesting and worth studying.

* I found the dynamics identified in Theorem 3.1 to be interesting. This is a nice link between some latent diffusion work and Riemannian geometry. However, I have several concerns regarding the novelty and applicability of such dynamics.

**Weaknesses:**

There are several limitations to this work:

* Lack of baseline experiments: while there is merit in this work I find the lack of baseline experiments to be concerning. There are many existing works on diffusion on Riemannian manifolds and I think a comparison with such works is necessary to assess the applicability of the method. To be more specific, in the case of rotated MNIST, why not compare the performance with a classical diffusion model on MNIST augmented with rotation? Similarly for QM9, why not compare with a diffusion model on QM9 where the QM9 dataset is augmented with the symmetries of the $G = (\mathrm{SO}(3) \times \mathbb{R}_+)^N$ and $G = T(3)^N$. Regarding Section 5.4 on CrossDocked2020, I know that the authors are space limited but I would suggest cutting some other parts of the paper to better explain CrossDocked2020. Such examples are important because, to my understanding, this is the only example in the paper dealing with a dataspace $X$ which is actually different from the classical Euclidean space. In these settings, it would be good to compare to other Riemannian based diffusion models such as [1, 2, 3]

* Motivation and link with existing literature: Looking at Theorem 3.1 I find that this result is extremely related to existing results on latent diffusion, see Section 4.1. of [4]. Broadly speaking, it seems to me that the main result of the paper is to translate a diffusion on a dataspace X to a Lie group G, the same way a bijective function would (or any minimal assumption needed to get a change of variable). The case of a bijective transformation is exactly the one treated in [4]. It would be great to acknowledge that fact if it is indeed the case. I understand that the settings and motivations are different but such works on latent diffusion cannot be ignored. Regarding works on Lie groups, there is a line of work devoted to the implementation of diffusion models on Lie groups, see for instance [5, 6] and references therein. In particular [5] focuses on diffusion models on SE(3) which seems to be related to the application presented in Section 5.4.
Mathematical correctness: I think the paper would highly benefit from a clarification between the general setting of a data space X and the special case of a Euclidean space. Indeed, while I understand that a few conditions are enforced to apply Theorem 3.1 (completeness, homogeneity, commutativity), the central mapping $\Pi$ should transfer a vector field defined on the Lie algebra of G into a vector field which is a section of the tangent bundle of X. In Theorem 3.1, we have that $\Pi$ is defined on Euclidean spaces. While there is a local identification between the tangent space of $X$ at any point and a Euclidean space, I am not clear where this is explicit in the paper. A similar remark holds for all the quantities introduced in Theorem 3.1. In general, I found the key quantity $\Pi$ to be poorly defined. The definition of this quantity given in line 173 “the collection of fundamental fields $\Pi$” is not enough to understand how it is defined truly.

*Limitation: I don’t think the limitations of the work are discussed in enough depth. For instance, a central and essential point for the procedure described in the paper to work is the access to the mapping $\Pi$, its inverse and the ability to compute its determinant (according to the sampling paragraph, “Generating samples through the reverse SDE”). When is that the case? The authors have presented a few cases when that is possible but one of the crucial assumption is basically that G and X have the same dimensionality (in order to compute the inverse mapping). See Section 5.1, Section 5.2 and Section 5.3 to verify that the examples of the authors match this assumptions. There are many cases where the Lie group acting on the dataspace $X$ is not of the same dimension. In that case it seems very hard, if not impossible, to construct the diffusion process on $G$.

[1] De Bortoli et al. – Riemannian Score-Based Generative Modelling

[2] Huang et al. – Riemannian Diffusion Models

[3] Chen and Lipman – Flow Matching on General Geometries

[4] Kim et al. – Maximum Likelihood Training of Implicit Nonlinear Diffusion Models

**Questions:**

* I might be missing something but I think Vect is TX in (1)

* What is the difference between x and bolded x (for example in (1)). Both seem to denote vector data.

* Please unify the notation (x,y) and (x_1, x_2) in the fleshed out example on the rotation on the plane

* What action does A represent in (5)? It is not defined in this section. I am guessing that it is associated to L but for some reason the dependency of L with respect to A is not highlighted.

* Proposition 2.1. Can we have > dim X? This is suggested by the phrasing of Proposition 2.1. If such an example exists please provide it.

* Is homogeneity really important? “ensure that the Langevin dynamics (5) will behave appropriately,” > what does appropriately mean here. For instance I could think of densities supported on only O_+

* I would move line 173 after Equation (1) to introduce the linear operator right away.

* A more general (open) question is the relevance of constrained generative models. In the light of the new Alphafold paper for instance [1], it seems that constrained generative models do not fare well compared to simpler models fed with a lot of data (and augmentations). I think running such comparisons and explaining the author's stance on this point would be very valuable for the paper and the community.

[1] Abramson et al. – Accurate structure prediction of biomolecular interactions with AlphaFold 3

---

> ### Author Response · Authors · 2024-11-20
>
> First of all, we thank the reviewer for taking the time to read our manuscript and provide valuable feedback. We look forward to a constructive rebuttal period and are confident that the paper will be improved through this exchange.
>
> In the following, we address the points raised in both the "Weaknesses" and "Questions" sections.
>
> -----------------
>
> **Weaknesses**
> -----------------
> _____________
> *Motivation*
>
> **TL;DR: We propose a diffusion process in Euclidean space that retains all the characteristics of curved Lie group diffusion, enabling unconstrained dynamics (unlike equivariant neural networks) while preserving the group action on the data.**
>
> The key idea is that while data is typically represented in Euclidean space (e.g., 3D coordinates for point clouds or pixel values for images), its underlying "true" signal coordinates may reside in a curved space (e.g., bond angles for molecules or global transformations of images/solids). However, performing diffusion in curved spaces poses significant challenges:
>
> - *Data transformation/preprocessing*: Extracting curved coordinates requires extensive preprocessing, which is generally labor-intensive and complex.
> - *Challenging diffusion dynamics*: Diffusion on curved manifolds is challenging due to curvature-related terms and operations (covariant derivatives, etc..). While promising, this area is less developed compared to Euclidean diffusion, where mature theoretical and computational tools are readily available.
> -  *Projection requirement*: Ensuring diffusion paths remain on a curved manifold $X$ requires projections (e.g., the exponential map for Lie groups), which are computationally expensive.
> - *Need of sampling*: Exact solutions to the SDE are unavailable for general curved spaces, necessitating simulation to obtain $p(x(t)|x(0))$. In contrast, we provide an exact solution for **any**  Lie group $G$ (see Equation 8).
>
> Thus, our approach aims at comprising the "best of both worlds": having a "curved-like"-dynamics (as if it was on a Lie group) but purely in Euclidean space, so that we do not need to modify the data and we can take over any computational method developed for standard diffusion models.
>
> The "curved dynamics" determined by the Lie group enters in the SDE through a first-order term (the Lie algebra elements) and one second-order term (the Casimir element). In general, the projection map is a Taylor expansion with infinite terms.
>
> Moreover, we wish to point out that **standard Euclidean diffusion also corresponds to a choice of Lie group**, so it can also be seen as part of our formalism. So, our proposal is not an alternative to standard Euclidean diffusion, but rather an extending framework.
>
> Finally, when the Lie group $G$ satisfies the constraints of Section 2.2, the approach generate an **uncostrained dynamics**, that is, it is able to learn any distributions with support on $X$, thus achieving the maximum expressiveness while maintaining the inductive bias given by the group action on the data space.

---

> > ### Author Response · Authors · 2024-11-20
> >
> > *Lack of baseline experiments.*
> >
> > Following the motivation above, we restate that our framework does not learn a distribution augmented with the Lie group transformations. Our framework learn **any data distribution**, regardless of whether it possesses the symmetry of $G$ or not.
> > This is demonstrated explicitly in the experiments in Section 5.1, where we successfully reconstruct symmetry-free distributions for any chosen (suitable) group. Similarly, in the QM9 example, we showcase the generation of molecules using either $G = T(3N)$ or $G = SO(3) \times \mathbb{R_+})^N$. This can be seen as a comparison to a baseline experiment, since $G=T(3N)$ corresponds to the standard Euclidean diffusion.
> >
> > The advantage of our framework comes with the flexibility in choosing $G$ such that it aligns with the inner degrees of freedom of the system, or when we wish to learn a specific transformation of that data.
> > An example that exemplifies the utility of our approach is the MNIST example in section 5.2.
> > We wanted to learn a mapping between $p_T=$ rotated MNIST and $p_0=$ standard MNIST. Each digit is represent by a picture $x=\mathbb{R}^{28\times28}$, and we need to learn a $28\times28=784$ dimensional bridge problem (since we need to map two non-trivial distributions), even though we intuitively know it is a 1-dimensional problem. Indeed, with standard diffusion it is not possible to solve the problem. If we learned an augmented distribution with random rotation of images, all we could do with standard diffusion is generate images which will have a random rotation.
> >
> > If we wanted to perform diffusion on a curved manifold, we would need to transform the MNIST images $x$ in such a way that they lie in a curved $SO(2)$ manifold, which is very complicated (we are not sure how to do that).
> >
> > With our methods, we can simply apply a $G=SO(2)$ generalized diffusion on the original data, leveraging the data structure we have and the true dimensionality of the problem.
> >
> > The only other option for a baseline approach is to train a bridge-model in a $784$ dimensional space. Therefore, **we trained a Brownian Bridge Diffusion model (BBDM) [1] on the rotated MNIST dataset from the experimental Section 5.2.**
> > The BBDM implements a continuous-time stochastic
> > process in which the probability distribution during the diffusion process is conditioned on the start $x(0) \sim q(x_0)$ and end point $x(T) \sim p(x_T)$, where an intermediate point $x(t) \sim p(x_t| x_0, x_T, t) = N(x_t, \mu(t, x_0, x_T), \Sigma(t)$ can be simply drawn since the mean function depends on the pairs $(x_0, x_T)$, see Eq. 3 and 4 in reference [1].
> > We leveraged the $x_0$-parametrisation to train the BBDM which states that given a noisy interpolated image $x(t)$, the network is tasked in predicting the original data point $x(0)$. Note that we do not train the BBDM on exact paired data, since the augmented MNIST digits $x(T) \sim p(x_T|x_0)$ are computed on the fly, similar to the experiment in Section 5.2, and therefore on input image $x(0)$ can have multiple random augmented samples.
> >
> > We observe that the BBDM model is, in some cases, not able to reconstruct a rotated MNIST digit, see Figure 6a, digits: (2,4,5,6,7) but removes/adds pixels in the image and in some cases it transitions from a rotated digit to another digit as shown in the example (from 9 to 4) in the last row of Figure 6b in the updated manuscript. This artifact is most likely due to the way the BBDM is trained by pinning two endpoints and allowing to change the (latent) represenation in the full 784-dimensional space.
> >
> > [1] BBDM: Image-to-image Translation with Brownian Bridge Diffusion Models. Bo Li, Kaitao Xue, Bin Liu, Yu-Kun Lai, CVPR 2023
> >
> > -------------
> > *Link with existing literature.*
> >
> > We agree with the reviewer that the connection with the subfield of latent diffusion needs to be mentioned, as well as the connection with the bijective mapping of normalizing flow. We restructured the related work section and include these lines of work as well a more comprehensive list of work on diffusion on Lie groups.
> >
> > -------------
> >
> > *Mathematical correctness.*
> >
> > The reviewer is correct that this distinction is not well incorporated into the various parts of the work. As mentioned above, our goal is to have a (curved) Lie group dynamics in Euclidean space, but in doing so, we derived sufficient conditions for generalized score matching which are valid also if $X$ is curved. Thus, to summarize:
> >
> > - The conditions in section 2.2 "SUFFICIENT CONDITIONS FOR LIE GROUP-INDUCED GENERALIZED SCORE MATCHING" are valid for a curved $X$
> > - For the main results Theorem 3.1, and the rest of the paper (as well as its focus) $X$ is flat/Euclidean.
> >
> > We made this clear in the introduction and in the various sections, so that no more confusion should arise.

---

> > > ### Author Response · Authors · 2024-11-20
> > >
> > > *Limitations.*
> > >
> > >  We clarify here the relationship between the dimension of the Lie group $G$ and the dimension of that data space $X$. Let $n_G = \dim G$ and $n_X = \dim X$.
> > > - ${n_G > n_X}$. This case $X$ is in general "overparametrized" (of course it still needs to satisfy the conditions of section 2.2), and there is no issues with it. We presented some examples of such cases in the answers to the "Questions" section. It might be potentially inefficient, but it is possible to learn any distribution $p(x)$.
> > > - ${n_G = n_X}$. Also in this case, if $G, X$ satisfy the conditions of Section 2.2, it is possible to learn any distributions $p(x)$. This is the most efficient case because we can do it with  $n_X$ scores.
> > > -  ${n_G < n_X}$. Here we cannot possibly learn any distributions, as the conditions are violated. However, we can still learn specific conditional distributions obtained from the joint $p(x)$. For instance, in the MNIST example $n_G = 1$, $n_X=784$ and we learned the rotation degree of freedom. In the CrossDocked example, we learned the 6-dimensional degrees of freedom of the ligand *given* a protein and a ligand.
> > >
> > >
> > > Thus, the conditions in Section 2.2 are describing in which cases we can learn any unconstrained distribution $p(x)$ with support on $X$ using generalized score matching. If the conditions are not satisfied we can still learned interesting problems as the examples for $n_G < n_X$ show.
> > >
> > > _________________
> > > **Questions**
> > > ------------------
> > > ________________
> > >
> > > *I might be missing something but I think Vect is TX in (1)*
> > >
> > > $Vect$ is not quite $TX$. $TX$ is the tangent bundle. $Vect$ denote the space of (smooth) vector fields on $X$, that is, of maps $V: X \rightarrow TX$, that is, of sections of the tangent bundle. In the more mathematical community sections are often denoted as $\Sigma$, so $Vect(X) = \Sigma(TX)$.
> > >
> > > ---------------
> > > *What is the difference between x and bolded x*
> > >
> > > This is a typo. Thank you very much for catching that!
> > >
> > > ---------------
> > > *Please unify the notation (x,y) and (x_1, x_2)*
> > >
> > > Also this is a remanence effect of a change of notation. Thank you very much for reading the draft carefully and pointing this out!
> > >
> > > --------------
> > > *What action does A represent in (5)?*
> > >
> > > $\boldsymbol{A}$ is a "matrix" that collects the basis elements of the Lie algebra $\boldsymbol{A} = (A_1, A_2, etc)$. We denoted "matrix" since the single basis elements are often represented by matrices themselves, but if we see each element of $\mathfrak{g}$ as a vector (living in the tangent space $T_eG$) then the notation of matrix is justified. We address this in the text.
> > >
> > > -----------
> > > *Proposition 2.1. Can we have > dim X?*
> > >
> > > Yes we can. One example is provided in the main text in the lines 201-206. Another example is the following.
> > >
> > > Consider $X=S^2$ the 2-sphere and $G=SO(3)$. Note that $\dim S^2=2$ and $\dim SO(3)=3$. This satisfies Proposition 2.2 (thus condition 1), as the stabilizer $G_x$ at each point $x\in X$ is 1-dimensional (rotations around that line defined by the origin and $x$ fix $x$), and while $\dim G =3$ we have that $\dim G/G_x = 2 $ and $n=dim X =2$ thus, $\dim G/G_x < n$ holds nowhere on $S^2$.
> > >
> > > Homogeneity also holds since each two points on the sphere can be obtained through a $SO(3)$ rotation (around the axis perpendicular to the place containing the great circle connecting the two points).
> > >
> > > Finally, the vector fields on $S^2$ generated by the Lie algebra elements commute (even if the algebra elements do not!). This is shown in appendix A.3 (equation (25)).
> > >
> > > ------------
> > > *Is homogeneity really important?*
> > >
> > > Homogeneity simply states that any two points in $X$ can be connected by a group transformation, since these determined the dynamics. If the opposite would be the case, it could happen that the success of the dynamics to reach the desired sample would depend on the starting point, which is something we do not necessarily want to introduce. Other than that, everything would work even without this constraints. We will make this clearer in the manuscript.
> > >
> > > In the case that the reviewer is suggesting, everything would work exactly the same. From our perspective though, we would see $X = O_+$, so that problem can be translated to an homogeneous space as well.
> > >
> > > But, in case we do not have a problem with the fact that we are only able to learn mapping between specific orbits and specific sub-supports of the density, we can leave out the assumption of homogeneity.
> > >
> > > The reviewer's point is well-taken, and we will make sure to include it in the main text.
> > >
> > > -----------------
> > > *I would move line 173 after Equation (1) to introduce the linear operator right away.*
> > >
> > > Also this point of the reviewer is well-taken. This would also avoid the confusion from the point above.

---

> ### Author Response · Authors · 2024-11-20
>
> *A more general (open) question is the relevance of constrained generative models.*
>
> We thank the reviewer for giving us a chance to explain this. Our strategy **is not constrained**:
> - By selecting $G$ according to the sufficient condition of section 2.2, we obtain a **fully unconstrained** dynamics, whose sub-dynamics though follows the orbits of the given Lie subgroups.
> - This is shown explicitly in the experiments in section 5.1 (where we are able to reconstruct symmetry-free distributions with any suitable group), as well as in the QM9 example, where we can generate molecules either with $G=T(3N)$ or with $G=(S(3)\times \mathbb{R})^N$.
> - Also recall that we proved that traditional diffusion models are a special case of our formalism with $G=T(N)$, so our formalism includes all known unconstrained models as well.
> - In Figure 1(c,d) we tried to depict in a simple case the difference between our generalized Lie Group-induced dynamics and a constrained dynamics (equivariant or invariant).
>
> In this sense, we believe that our approach could bring the benefits of inductive bias into such models (for instance, aligning Lie group subfactor to the desired degrees of freedom of the data, as in Figure 4 for bond and torsion angles), without loosing the expressivity of unconstrained generative models.
>
> This is a very interesting and important point, and we will include it in the final discussion.

---

> > ### Author Response · Authors · 2024-11-26
> > **Additional baseline experiment for CrossDocked**
> >
> > Following the suggestion of the reviewer, we performed a **comparison between our proposed method and diffusion on Lie groups based on the strategy of "Riemannian score-based generative modelling" (RSGM) [1] in the CrossDocked experiment.** We re-implemented the strategy of [1,2,3] for modelling the $T(3)\times SO(3)$ dynamics corresponding to global transformations (translation and rotations) of the ligand. We compared the Root mean square deviation (RMSD) for all the 100 molecules in the test dataset from the two approaches. For each molecule we generated 5 poses and we compared the RMSD with respect to the ground-truth docked ligand. The results are presented in the updated Figure 8. In summary, our framework achieves a lower (i.e., better) RMSD, namely in average RMSD=$2.9\pm 1.0$ Å vs RMSD=$5.6\pm 1.2$ Å for RSGM.
> >
> > [1] De Bortoli et al, ”Riemannian score-based generative modelling”
> >
> > [2] Huang, Chin-Wei, et al. ”Riemannian diffusion models.”
> >
> > [3] Corso, Gabriele, et al. "Diffdock: Diffusion steps, twists, and turns for molecular docking."
> >
> > ------------------
> >
> > Thank you again for your valuable feedback and the time you spent reviewing our submission. Your comments have been instructive and key in helping us improve the paper, and we feel we addressed carefully all the points you raised.
> >
> > We hope that our revisions answered your questions and resolve your concerns, and if so, we would greatly appreciate it if you could update your review to reflect this. Of course, we remain available and ready to address any further questions or comments you may have.
> >
> > Thank you for your input and for helping us strengthen the paper.

---

> > > ### Author Response · Authors · 2024-12-02
> > >
> > > Dear Reviewer hBkJ,
> > >
> > > We sincerely thank you for the time and effort you invested in reviewing our paper and for your valuable comments, which we took seriously and which have contributed to a significantly improved version of the manuscript.
> > >
> > > As the rebuttal period comes to an end, we would like to take this opportunity to ask the reviewer if there are any remaining concerns that we may address.
> > >
> > > In our response to the first round of comments, we carefully addressed all the points raised by the reviewer, including:
> > >
> > > - Lack of baseline experiments: We conducted **three additional experiments**, including Lie group diffusion on CrossDocked,  Brownian Bridge Diffusion model on MNIST and a SO(4) experiment.
> > > - Motivation and connection to existing literature: We expanded the related work section and clarified the scope of the paper. Specifically, we emphasize that $X$ being Euclidean and still being able to sample from a Lie group is not a weakness but rather the main advantage achieved by of our work.
> > > - Presentation and notation: We improved the overall presentation of the paper and corrected notation issues, which was also acknowledged by Reviewer T87W.
> > >
> > > We would like to kindly request the reviewer to reevaluate our work in light of these discussions, the substantial revision of the manuscript and the steps we have taken to address all concerns raised.
> > >
> > > If there are any further questions requiring clarification, we would be more than happy to provide additional explanations.
> > >
> > > Best regards,

---

### Official Review · Reviewer_75NJ · 2024-10-30

**Soundness:** 3
**Presentation:** 4
**Contribution:** 1
**Rating:** 3
**Confidence:** 5

**Summary:**

A traditional generative model is a (stochastic) flow of a vector field in Euclidean space. This paper expands it to a new case: it considers a vector field on the manifold and the vector field comes from a Lie group acting on the manifold. The paper is well-written with well-illustrated numerical experiments provided. However, there are some important weaknesses, please see the weaknesses section. I will increase my score if the weakness is explained.

**Strengths:**

The interesting idea of using the infinitesimal generator of a group action to generate a vector field on the manifold is new to me. The author claims that this approach enables us to design diffusion models on manifolds. The author claims the algorithm handles the non-Abelian Lie groups as long as conditions in Sec. 2.2 is satisfied. (see more discussion about Sec 2.2 in weakness)

The numerical experiments are interesting since it seems the manifold structure is well-learned in some synthetic examples. Applications on real datasets are also provided (MNIST and QM9).

**Weaknesses:**

My main concern is: Can X be curved? Here are the details for this question:

First, notice that in all examples, X is Euclidean, even in the example in line 223-226 and Sec A.3 arguing the conditions are not that strong. Could you please explicitly address whether your theoretical framework and algorithm can be applied to the cases where X is a curved manifold? If so, could you demonstrate this by applying your example to a curved X?

Secondly, in Eq. 8 in the algorithm, the order of multiplication of $O(\tau(t))$'s does not matter, which means they exchange. This means your algorithm highly depends on Condition 3.

Finally, assume condition 3 is given, then we can see $[L_A, L_B]f=0$ ->$L_{[A, B]}f=0$ -> $[A, B]=0$ -> the Lie bracket vanishes on X (by condition 2) -> X is flat. This seems to be short proof showing that condition 2+3 infers that X is flat. Is this true? If not, could you provide a counterexample or explain how the framework can accommodate curved spaces while satisfying the conditions in Sec 2.2?

**Questions:**

If the weakness I stated is incorrect, could you please give me an example that X is non-flat with all conditions in Sec 2.2 is satisfied?

If the weakness I stated is true, I have the following question:

1. If the proposed method is indeed limited to Euclidean spaces, could you elaborate on the specific advantages it offers over traditional algorithms in these cases? Are there particular types of problems or data structures where your approach would be more efficient or effective?

2. Given that Condition 2 assumes a homogeneous space, how does this impact the expressiveness of your model compared to traditional Euclidean methods? Could you clarify if there are scenarios where your approach provides benefits even within these constraints?

---

> ### Author Response · Authors · 2024-11-21
>
> We thanks the reviewer for taking the time to read and provide comments to our manuscript. We are looking forward to a constructive rebuttal period and we are confident that the paper will improve through the exchange.
>
> **Weaknesses**
> -------------
> __________
> *Can X be curved?*
>
> The very short answer to this question is as follows:
> - The conditions in section 2.2 "SUFFICIENT CONDITIONS FOR LIE GROUP-INDUCED GENERALIZED SCORE MATCHING" are valid for a curved $X$
> - For the main results Theorem 3.1, and the rest of the paper $X$ is flat/Euclidean.
>
> Now, this is not a restriction on our part but a truly motivating factor for our work. Let us be more explicit about it.
>
> Our goal is to define a dynamics which can be represented by any continuous transformation (Lie group) but acting on flat space.
> In other words: *How can we obtain a diffusion process that behaves like it is curved (as the orbits/trajectories of the generalized score functions are not in general "lines", i.e., geodesics of Euclidean space) while remaining on flat space?*
>
> The motivation for the above goal is as follows: while data is typically represented in Euclidean space (e.g., 3D coordinates for point clouds or pixel values for images), its underlying "true" signal coordinates may reside in a curved space (e.g., bond angles for molecules or global transformations of images/solids). However, performing diffusion in curved spaces poses significant challenges:
>
> - *Data transformation/preprocessing*: Extracting curved coordinates requires extensive preprocessing, which is generally labor-intensive and complex.
> - *Challenging diffusion dynamics*: Diffusion on curved manifolds is challenging due to curvature-related terms and operations (covariant derivatives, etc..). While promising, this area is less developed compared to Euclidean diffusion, where mature theoretical and computational tools are readily available.
> -  *Projection requirement*: Ensuring diffusion paths remain on a curved manifold $X$ requires projections (e.g., the exponential map for Lie groups), which are computationally expensive.
> - *Need of sampling*: Exact solutions to the SDE are unavailable for general curved spaces, necessitating simulation to obtain $p(x(t)|x(0))$. In contrast, we provide an exact solution for **any**  Lie group $G$ (see Equation 8).
>
> Our approach, therefore, aims at obtaining an equivalent dynamics to curved space diffusion, while remaining in Euclidean space. Thus we have the property of the former, with all the benefits of the latter. In particular, the data does not need to be transformed, we can use all the algorithm developed for Euclidean diffusion, and there is no need for an expensive projection map.
>
> The "curved dynamics" determined by the Lie group enters in the SDE through a first-order term (the Lie algebra elements) and one second-order term (the Casimir element). In general, the projection map is a Taylor expansion with infinite terms.
>
> Moreover, we wish to point out that **standard Euclidean diffusion also corresponds to a choice of Lie group**, so it can also be seen as part of our formalism. So, our proposal is not an alternative to standard Euclidean diffusion, but rather an extending framework.
>
> Finally, when the Lie group $G$ satisfies the constraints of Section 2.2, the approach generate an **uncostrained dynamics**, that is, it is able to learn any distributions with support on $X$, thus achieving the maximum expressiveness while maintaining the inductive bias given by the group action on the data space.
>
> We will make this clearer in the introduction where we are motivating our work.
>
> ----------
> *Counterexample to $X$ only being flat*
>
> While the main Theorem is derived for flat $X$ for the reasons above, we derived the sufficient conditions for generalized score matching in Section 2.2 for a general $X$.
>
> A counterexample as the reviewer asks is the following. Consider $X=S^2$ the 2-sphere and $G=SO(3)$. This satisfies Proposition 2.2 (thus condition 1), as the stabilizer $G_x$ at each point $x\in X$ is 1-dimensional (rotations around that line defined by the origin and $x$ fix $x$), and while $\dim G =3$ we have that $\dim G/G_x = 2 $ and $n=dim X =2$ thus, $\dim G/G_x < n$ holds nowhere on $S^2$.
>
> Homogeneity holds because any two points on the sphere can be connected by an SO(3) rotation around the axis perpendicular to the plane of the great circle between them.
>
> Additionally, the vector fields on $S^2$ generated by the Lie algebra elements commute (even if the Lie algebra elements themselves do not!), as shown in Appendix A.3. These actions, restricted to $SO(3)$, preserve the radius and descend naturally to $S^2 \subset \mathbb{R}^3$.
>
> The reviewer's proof fails because condition 3 must hold only **locally**, not globally; the orthogonal tangent vectors spanning $G/G_x$ vary with the point $\mathbf{x}$ on $S^2$, as explicitly shown in Equation (25).

---

> ### Author Response · Authors · 2024-11-21
>
> **Questions**
> ----------------
> ____________
>
> *Elaborate on the specific advantages it offers over traditional algorithms.*
>
> An example that exemplifies the utility of our approach is the MNIST example in section 5.2.
> We wanted to learn a mapping between $p_T=$ rotated MNIST and $p_0=$ standard MNIST. Each digit is represent by a picture $x=\mathbb{R}^{28\times28}$, and with traditional diffusion we need to learn a $28\times28=784$ dimensional bridge problem (since we need to learn a map between two non-trivial distributions), even though we know intuitively it is a 1-dimensional problem.
>
> If we wanted to perform diffusion on a curved manifold, we would need to transform the MNIST images $x$ in such a way that they lie in a curved $SO(2)$ manifold, which is very complicated.
>
> With our methods, we can simply apply a $G=SO(2)$ generalized diffusion on the original data, leveraging the data structure we have and the true dimensionality of the problem.
>
> The only other option for a baseline approach is to train a bridge-model in a $784$ dimensional space. Therefore, **we trained a Brownian Bridge Diffusion model (BBDM) [1] on the rotated MNIST dataset from the experimental Section 5.2.**
> The BBDM implements a continuous-time stochastic
> process in which the probability distribution during the diffusion process is conditioned on the start $x(0) \sim q(x_0)$ and end point $x(T) \sim p(x_T)$, where an intermediate point $x(t) \sim p(x_t| x_0, x_T, t) = N(x_t, \mu(t, x_0, x_T), \Sigma(t)$ can be simply drawn since the mean function depends on the pairs $(x_0, x_T)$, see Eq. 3 and 4 in reference [1].
>
> We observe that the BBDM model is, in some cases, not able to reconstruct a rotated MNIST digit, see Figure 6a, digits: (2,4,5,6,7) but removes/adds pixels in the image.  In some cases it even transitions from a rotated digit to another digit as shown in the example (from 9 to 4) in the last row of Figure 6b in the updated manuscript. This artifact is most likely due to the way the BBDM is trained by pinning two endpoints and allowing to change the (latent) represenation in the full 784-dimensional space.
>
> [1] BBDM: Image-to-image Translation with Brownian Bridge Diffusion Models. Bo Li, Kaitao Xue, Bin Liu, Yu-Kun Lai, CVPR 2023
>
>
> Other advantages of our method compared to Riemannian diffusion were highlighted in our summary of the motivation for this work above. For example, compared to GeoDiff [1], our trained model for conformer generation on QM9 is faster, requiring only T=100 diffusion timesteps, whereas GeoDiff was trained with T=5000 (a speed increase of 50 times!). This is significant because a higher number of timesteps is a major drawback of diffusion models and has driven the development of alternative approaches like flow matching.
>
> [1] Xu, Minkai, et al. "Geodiff: A geometric diffusion model for molecular conformation generation."
>
> --------------
> *Homogeneity*
>
> First a comment about why we impose homogeneity: homogeneity simply states that any two points in $X$ can be connected by a group transformation, since these determined the dynamics. If the opposite would be the case, it could happen that the endpoint of the dynamics would depend on the starting point, which is something we do not necessarily want to introduce. Other than that, everything would work even without this constraints. We will make this clearer in the manuscript.
>
> To the specific points:
>
> **Traditional Euclidean methods also satisfy homogeneity**. We prove that traditional diffusion model are a subclass of our formalism where we take $G=T(N)$, and $X=\mathbb{R}^N$, but $X$ is homogeneous with respect to $G$ for this choice: given $x,y\in \mathbb{R}^n$, there is always a translation element $v=y-x \in T^(N)\simeq \mathbb{R}^N$ connecting the two points.
>
> Thus, homogeneity is not an extra requirement with respect to traditional Euclidean methods.

---

> > ### Comment · Reviewer_75NJ · 2024-12-01
> >
> > Dear authors
> >
> > I appreciate your reply. I appreciate your `Counterexample to $X$ only being flat` and I am fully convinced for that question. I have some other questions:
> >
> > Q1: You wrote `For the main results Theorem 3.1, and the rest of the paper $X$ is flat/Euclidean.` That means you confirmed that your algorithm only works in Euclidean space.
> >
> > My understanding is Sec 2.2 is just making assumptions and there exist cases that curved $X$ satisfies your assumptions in Sec 2.2 (thank you for the example). However, the main contribution, which is the algorithm in Sec 3, is still restricted to Euclidean $X$. This significantly weakens the contribution. Also, it would be misleading to claim $X$ can be a manifold since your algorithm in Sec 3 won't work in curved spaces.
> >
> > Q2: Your reply shows `An example that exemplifies the utility of our approach is the MNIST example in section 5.2.`. I am a little confused by this MNIST example in Sec 5.2 and I found in your reply to hBkJ, $X=\mathbb{R}^{28\times28}$ (you may have a small typo in the reply) and $G=SO(2)$. In this case, $dim G=1$ and $dim X=784$, and you admitted that condition 1 is not satisfied. (Small advice: could you add what X is in the MNIST example to the paper in the next version, please?)
> >
> > The experiment in which you apply your algorithm without meeting the requirement is not very convincing. Your algorithm is fully based on Thm 3.1 and Thm 3.1 requires all the conditions in Sec 2.2. Could you elaborate more about when indeed this algorithm can be applied?
> >
> > Q3: You wrote `even though we know intuitively it is a 1-dimensional problem.`. This is a good intuition and gives me the following understanding: what you hope to generate is indeed an element $g\in SO(2)$, and the picture generated is g act on $R^2$. If my understanding is true, the existing Riemannian generative model (e.g., RSGM) will solve this problem and I would think it would be convincing to provide a better baseline. In other words, I think adding an algorithm that generates elements from $SO(2)$ directly as a comparison would be helpful.
> >
> > Q4: How do you check the conditions from Sec 2.2 for a given Lie group $G$ and some space $X$? As you can see, in the updated version, it is hard to check even for $G=SO(d)$ and $X=R^n$ and your calculation seems still unfinished due to time constraints. If some users have something different tasks, what should they do? For example, 1. $G=U(d)$ and $X=C^n$; 2. $G:=GL$ 3. More examples in Wikipedia https://en.wikipedia.org/wiki/Lie_group#More_examples_of_Lie_groups
> >
> > Q5: Based on all the above questions, the main contribution is still unclear. The theoretical analysis and the experiment are not aligned. Many claims, even the ones added in the rebuttal period, are either too strong or incorrect. E.g., `first result of denoising score-matching result for general non-Abelian groups` (you are not generating points from a Lie group), `valid for any differentiable manifold X` (the algorithm only works on Euclidean X), `Dimensionality reduction`(cannot be achieved under condition 1), `2d, 3d and 4d distributions` (the verification of the conditions for SO(4) is not finished yet in Sec A.4), `with our framework there is no need of a tradeoff, as we retain the expressivity of unconstrained models with the benefits of group inductive bias.` (In rotate MNIST, you definitely lose the expressivity since your model only learns rotation. I think it would be better if you could clarify that dim reduction and expressivity cannot be achieved at the same time); `that our method can be applied to higher dimensional Lie groups.` (The most complicated Lie group you performed was SO(3), due to the fact it is hard to verify condition 3 for even SO(4))

---

> > > ### Author Response · Authors · 2024-12-02
> > >
> > > Dear Reviewer 75NJ,
> > >
> > > Thank you very much for your response and for acknowledging that we have successfully addressed some of your concerns!
> > >
> > > In the following, we will address each point you raised individually. However, we believe there may still be some misunderstanding regarding the main objective of the paper. Specifically, you repeatedly refer to the fact that $X$ is Euclidean as a weakness, when in fact this is central to the contribution of the paper. We think that agreeing on the scope of the paper is a necessary condition for having a constructive rebuttal discussion.
> > >
> > > We will attempt to clarify this point more explicitly below, but allow us to first summarize our motivation and main contribution:
> > >
> > > Our main contribution is the derivation of the conditions and mathematical formulation (via the SDE approach) for a curved diffusion process (in the sense of Lie group orbits), while remaining within Euclidean space. **In fact, our approach removes the requirement for the space being in curved space for solving the task of generating points on a (curved) Lie group through diffusion, but enable/extends it to Euclidean spaces.**
> > > Therefore, $X$ being Euclidean is actually a strength, as our method enables Lie group diffusion (in the style of RSGM) without the need to perform diffusion on the Lie group itself. In other words, since this is mentioned in another question: **When a Lie group $G$ acts on a vector space $X$, our approach generates points on the Lie group $G$ while performing diffusion merely on the Euclidean $X$ itself.**
> > >
> > > ---------------------------
> > >
> > > **Q1**
> > >
> > > As highlighted above, our scope is to perform a generative dynamics similar to diffusion on Lie groups but while remaining on Euclidean space. We listed the advantages of this approach in our previous responses but for convenient we summarize them here:
> > > - *Data transformation/preprocessing*. The data needs to be preprocessed and transformed to extract the curved coordinates. This is laborious and difficult to do in general
> > > - Diffusion on curved manifolds includes additional difficulties that related to the curvature of the manifold itself. This is a very interesting field, but it is at its infancy compared to Euclidean diffusion, where lots of theoretical and computation results are readily available and we can leverage them.
> > > - *Need of projection*. In order to perform diffusion on a general manifold $X$, we need to impose the condition that the diffusion path is contained within $X$ at all times. This implies the need of a projection (the exponential map in case of Lie groups), which is in general very expensive to compute.
> > > - *Need of sampling*. For general curved space there is no exact solution to the SDE. That means that the forward solution (also in training) must also be simulated in order to obtain $p(x(t)|x(0))$. We derived an exact solution for **any** group $G$ (equation 8).
> > >
> > > Now, there are two main challenges that we had to solve:
> > > - **Score matching needs to be generalized.** We showed that the action on a group on a manifold manifests itself through a generalized score function in the context of diffusion. We needed to provide conditions such that such generalized score matching is successful. These are the conditions in Section 2.2 (which are satisfied in all our experiments! more details about it below). Even though our objective is generating Lie group elements through Euclidean space, it just turned out to be that these conditions are valid for more general differential manifolds, and we consider them a result on its own.
> > > - The second challenge consists in having a mathematical framework (via SDEs) that allows us **noising without the need of simulation** (e.g., the geodesic random walk in RSGM Algorithm 1), and a **tractable denoising score** (without the need of implicit score matching). This is what Theorem 3.1 achieves, as mentioned, for Euclidean $X$ since that is what we aimed for, that is, removing the complication of a curved manifold but maintaining the property of a curved diffusion dynamics according to a Lie group.
> > >
> > > These two challenges are tackled in Section 2 and 3 respectively.
> > >
> > > In the updated version of the manuscript we made sure to highlight in each section which assumptions are made. We hope that this helps clarifying the scope and motivation of our work and that there is agreement on the fact that **$X$ being Euclidean is not a weakness of our work but its strength and main scope.**
> > >
> > >
> > > We kindly request the reviewer to provide their feedback on this point and let us know if there is alignment regarding the scope and contribution of the paper. Establishing this common ground is essential for continuing the discussion on the remaining points.

---

> > > > ### Author Response · Authors · 2024-12-02
> > > >
> > > > **Q2**
> > > >
> > > > We thank the reviewer for giving us the opportunity to clarify our setup for the MNIST experiment.
> > > > - MNIST is a dataset of pictures with distribution $p_{MNIST}(\mathbf{x})$, where $\mathbf{x}\in \mathbb{R}^{28\times 28}$. We call this data space $X_{MNIST} = \mathbb{R}^{28\times 28}$
> > > > - RotatedMNIST is a dataset of pictures with distribution $p_{RotMNIST}(\mathbf{x})$, where $\mathbf{x}\in  \mathbb{R}^{28\times 28}$, whose elements $\mathbf{x}$ consists of random rotation of original MNIST picture, that is, $\mathbf{x}\in p_{RotMNIST}(\mathbf{x})$ iff there exists $R\in SO(2)$ such that $\mathbf{x}= R(\mathbf{x}')$ for $\mathbf{x}'\in p_{MNIST}(\mathbf{x'})$
> > > > - In our experiment, we want to learn a diffusion process that models the process $\mathbf{x}\rightarrow \mathbf{x}'$, where $\mathbf{x}= R(\mathbf{x}')$. Thus, the distribution from which we need to sample to achieve this, and therefore the distribution we wish to learn is the conditional distribution $p(\mathbf{x}'| \mathbf{x}), \mathbf{x}\in p_{RotMNIST}, \mathbf{x}' \in p_{MNIST}$. Note that this corresponds to learn (generate) the $SO(2)$ element $R_{\mathbf{x}\rightarrow\mathbf{x}'}$ that transforms $\mathbf{x}$ into $\mathbf{x}'$. We model this by a one-dimensional process, thus $X_{diff}= \mathbb{R}^1$ and $G=SO(2)$. Note that with this setup all the conditions are satisfied ($\dim G = \dim X_{diff}=1$ and the stabilizers vanishes, homogeneity is not an issue since $\mathbb{R}$ is the universal cover of $S^1=SO(2)$  and commutativity neither since $G$ is Abelian).
> > > >
> > > > So, our experiment satisfies all conditions of Section 2.2. The confusion arose since there needs to be distinction between the data space $X_{MNIST} =  \mathbb{R}^{28\times 28}$ and the actual diffusion space
> > > > $X_{diff} =  \mathbb{R}^{1}$, where our modeling takes place and all conditions are satisfied.
> > > >
> > > > In the Brownian Bridge Diffusion Model (BBDM) benchmark, we are forced however to use the full space $X_{data}$ as diffusion space, learning therefore a much more complicated dynamics that leads to the problems we highligted in Figure 6.
> > > >
> > > > We thank the reviewer for forcing us to be precise and avoid confusion. The final version will include all spaces $X_{diff}$ where the modeling actually takes place (in case of conditional learning this will differ from $X_{data}$, where the data live).
> > > >
> > > > > *In this case, [...] and you admitted that condition 1 is not satisfied*
> > > >
> > > > The reviewer is referring to the following bullet point while answering the question "what happens when the dimensions of $G$ and $X$ do not coincide"?
> > > >
> > > > > $n_G<n_X$ Here we cannot possibly learn any distributions, as the conditions are violated.
> > > >
> > > > Here is again a bit of disambiguous notation the cause of confusion. In the answer to the question of Reviewer hBkJ we were referring to unconditional learning, thus to the following question: given $X$, can we learn any distribution on $X$ with a given $G$?
> > > >
> > > > If $n_G<n_{X_{data}}$ that is not possible becasue our conditions are violated, in accordance with our claims. However, if we wish to perform conditional learning, we restrict the diffusion space $X_{diff}$ such that the condition $n_G\geq n_{X_{diff}}$ is satisfied. Thus, we are again in accordance with our conditions and the learning is possible. This is what happens in the MNIST experiment ($X_{data}= \mathbb{R}^{28 \times 28}, X_{diff}= \mathbb{R}^1$) and CrossDocked ($X_{data}= \mathbb{R}^{3 \times (Nr. atoms ligands + Nr. atoms pocket)}$, $X_{diff} =  \mathbb{R}^6$ modelling $SE(3)$).
> > > >
> > > > In summary, for all our experiments the conditions are satisfied, as we will be clear in every setting which are the correct spaces and the corresponding groups.

---

> > > > > ### Author Response · Authors · 2024-12-02
> > > > >
> > > > > **Q3**
> > > > >
> > > > > The reviewer is absolutely correct in their description of our scope in the MNIST example (we described that in a detailed fashion above in the answer to Q2), that is, we wish to generate an element $g\in SO(2)$. The reviewer is also correct that such problem can be solved also through the RSGM approach.
> > > > >
> > > > > We compared our method with the RSGM approach in the revised CrossDocked experiment, which in spirit is similar to the MNIST experiment (also there we are trying to learn a conditional distribution where $X_{diff}$ is smaller than $X_{data}$). In that experiment, we want to learn the global SE(3) transformation to dock a ligand in a protein pocket. Figure 8b shows that the generated samples from our method achieve a better RMSD (with respect to the ground truth of docked ligands), namely in average RMSD=$2.9\pm 1.0$ Å vs RMSD=$5.6\pm 1.2$ Å for RSGM.
> > > > >
> > > > > With respect to the choice of benchmarks, we chose different strategies for different experiments to show the benefits of our approach with respect to a variety of competitors approaches, namely:
> > > > > - MNIST: benchmark with respect to Brownian Bridge Diffusion Model (BBDM) for conditional learning between two non-trivial distributions
> > > > > - QM9: benchmark with respect to standard Euclidean diffusion (flat $G=T(3n)$ vs curved $G=(SO(3)\times \mathbb{R})^n$)
> > > > > - CrossDocked: benchmark with respect to Lie group diffusion (RSGM)
> > > > >
> > > > >
> > > > > That being said, the reviewer is correct that RSGM is also applicable to the MNIST setup. We are unfortunately unable to update the manuscript at this point, but we wish to take upon the reviewer suggestion and perform the benchmark they are suggesting, namely, diffusion on SO(2) on the MNIST dataset. We will add the results of the experiment in the final version of the paper.

---

> > > > > > ### Author Response · Authors · 2024-12-02
> > > > > >
> > > > > > **Q4**
> > > > > >
> > > > > > We thank the reviewer for a good question and for giving us a chance to clarify this point.
> > > > > >
> > > > > > First we address the group $SO(4)$. The calculations are not unfinished, we merely thought that presenting the parametrization (i.e., group action) together with the Lie algebra element (i.e., inifitesimal algebra action) suffices to make the point, since then everything else proceeds as in the cases $SO(2), SO(3)$. We are more than happy to provide more details about the general case $G=SO(n)\times\mathbb{R_+}$, $X=\mathbb{R}^n$ as well.
> > > > > >
> > > > > > Here we discuss how the group $G=SO(4)\times \mathbb{R_+}$, and $X=\mathbb{R}^n$ satisfy the conditions in Section 2.2.
> > > > > >
> > > > > > Given the Lie algebra matrices (34), we obtain the differential operators as described in the text
> > > > > > $$
> > > > > > L_{\varphi_k} = A_{\varphi_k}\boldsymbol{x} \cdot \nabla
> > > > > > $$
> > > > > > The fact that these commute $[L_{\varphi_k}, L_{\varphi_s}]=0$ is a rather lengthy calculation but can be shown explicit, similar to the SO(3) case. We are happy to include those in the appendix.
> > > > > >
> > > > > > Homogeneity is also given, since the parametrizations (33) and (35) are one-to-one and cover the whole space $\mathbb{R}^4$. This is the statement that two vectors in 4d can be transformed into each other by a dilation ($r$ transformation) and a 4d rotation $\varphi_{1,2,3}$.
> > > > > > More generally, it is a known fact in mathematics that the (oriented) n-sphere $S^{n-1} = SO(n)/SO(n-1)$, and thus $S^{n-1}$ is homogeneous with respect to $SO(n)$. Now, taking the product with $\mathbb{R_+}$ we obtain $X = S^{n-1} \times \mathbb{R_+}=\mathbb{R}^{n}$ is homogeneous with respect to $G=SO(n)$ since $X=\mathbb{R_+} \times SO(n)/SO(n-1)$. For $SO(4)$, we see that the stabilizer is $SO(3)$ which has indeed dimension 3.
> > > > > >
> > > > > > Finally, Condition 1 is also satisfied, since $\dim G = \dim \mathbb{R_+} + \dim SO(4)=1+6=7$ and for $SO(4)$ there is a $3$ dimensional stabilizers at each point in $\mathbb{R}^4$. In equation (34) we explicitely provided the 3 elements that do not vanish. Thus, condition 1 is satisfied.
> > > > > >
> > > > > > Now, to the other cases the reviewer mentioned:
> > > > > > - $G=U(d)$ and $X=\mathbb{C}^n$. In this case we have a complex Lie group. Our paper deals with the real case only (we explicitely point this out in the footnote on page 3), as this is by far the most relevant case for generative modeling. Extending this analysis to complex Lie group would be however a very interesting endevour for future work.
> > > > > > - $G=GL(3)$ is the general linear group in dimension 3. One possibility here is to take $X=\mathbb{R}^3$. Then the action of $GL(3)$ on $R^3$ is simply matrix multiplication. All the conditions are trivially satisfied since $G'=SO(3)\times \mathbb{R_+}$ is a subgroup, which we have discussed at length in the main paper.
> > > > > >
> > > > > >
> > > > > > In general, given any Lie group and a space $X$ on which $G$ admits an action, we need to known the action explicitly. This is not surprising since we also need that in case of Lie group diffusion, when we need to transform the data into the Lie group manifold. Once the group action is known, everything follows since we can obtain the infinitesimal action of $\mathfrak{g}$ on $X$ through differentiation, once we have these we have the differential operators and therefore we can compute the generalized score function.

---

> > > > > > > ### Author Response · Authors · 2024-12-02
> > > > > > >
> > > > > > > **Q5**
> > > > > > >
> > > > > > > We address individually the points that are still unclear to the reviewer.
> > > > > > >
> > > > > > >  > The theoretical analysis and the experiment are not aligned
> > > > > > >
> > > > > > >  We hope that our answer to the reviewer's questions Q2-Q3 address this point. The theory and the experiments are indeed aligned, we just need to be careful in defining which space we are performing our diffusion on.
> > > > > > >
> > > > > > > > valid for any differentiable manifold X (the algorithm only works on Euclidean X).
> > > > > > >
> > > > > > > The results we are referring to here is for **score-matching**, and not for **generative modeling**. The first  focuses on the accurate estimation of the score function (the gradient of the log density) in a given data distribution, whereas the latter aims at generating of new samples from a learned distribution.
> > > > > > >
> > > > > > > To summarize: Our contributions in the context of score-matching is valid for differentiable manifold (also beyond Lie groups). (This is what the above sentences is referring to)
> > > > > > >
> > > > > > > Our contribution for generative modeling is, as addressed above, focused on Euclidean $X$ to lift the necessity of performing diffusion in curved space but still be able to sample from the Lie group of interest.
> > > > > > >
> > > > > > > > Dimensionality reduction(cannot be achieved under condition 1)
> > > > > > > This is address in Q2-Q3 where we explicitely show that the condition is indeed achieved.
> > > > > > >
> > > > > > > > 2d, 3d and 4d distributions (the verification of the conditions for SO(4) is not finished yet in Sec A.4
> > > > > > >
> > > > > > >  We provided the proof that the conditions for SO(4) (and for SO(n)) are satisfied in our answer to Q4.
> > > > > > >
> > > > > > > > with our framework there is no need of a tradeoff, as we retain the expressivity of unconstrained models with the benefits of
> > > > > > > group inductive bias. (In rotate MNIST, you definitely lose the expressivity since your model only learns rotation. I think
> > > > > > > it would be better if you could clarify that dim reduction and expressivity cannot be achieved at the same time
> > > > > > >
> > > > > > > We hope we could clarify above the different scenarios between purely unconstrained generation and conditional generation.
> > > > > > >
> > > > > > > We wish to state again, within the assumption of the Conditions in Section 2.2, we do not have any constrained learning (as long as we use an unconstrained networks). This fact follows from the proofs from Section 2.2, which have been accepted by all reviewers.
> > > > > > >
> > > > > > > For the MNIST experiment, these conditions are satisfied for $X=\mathbb{R}^1$ and $G=SO(2)$, and in this setup we do not indeed lose any expressivity (exactly like RSGM, also their framework does not loose any expressivity), as we are able to learn any distribution on $SO(2)$.
> > > > > > >
> > > > > > > > that our
> > > > > > > method can be applied to higher dimensional Lie groups. (The most complicated Lie group you performed was SO(3), due
> > > > > > > to the fact it is hard to verify condition 3 for even SO(4))}
> > > > > > >
> > > > > > > Again, in our revision we provided an experiment on SO(4) and we provided the proof that such setup satisfied our conditions above.

---

> > > > > > > > ### Author Response · Authors · 2024-12-02
> > > > > > > >
> > > > > > > > >    First result of denoising score-matching for general non-Abelian groups (you are not generating points from a Lie group).
> > > > > > > >
> > > > > > > > We would like to clarify that we are indeed generating points from a Lie group. While we have addressed this point above (and the reviewer also acknowledges it in Q3), we will reiterate it here explicitly. From our experiments, it is evident that we are sampling from an angular distribution ($G = SO(2)$ for MNIST) or an $SE(3)$ distribution for CrossDocked. Specifically, for MNIST, we observe that **the intermediate states correspond to infinitesimal rotations of the starting image**, rather than other forms of interpolation, such as those produced by the BBDM method.
> > > > > > > >
> > > > > > > > Perhaps the confusion lies in how this process can be achieved while remaining in Euclidean space. The key idea, in a conceptual sense, is that we access the group indirectly through its action on the space (see Figure 2a). Formally, given $x = \rho_X(g)x_0$, where $x, x_0 \in X$ (we can assume that $x_0$ is fixed) and $\rho_X$ represents the action of $G$ on $X$, sampling $x$ is equivalent to sampling $g$ (again, since $x_0$ is fixed). Of course, the conditions outlined in Section 2.2 are crucial for this approach to work. For instance, every $x$ can be written like this for a fixed $x_0$ given the property of homogeneity.
> > > > > > > >
> > > > > > > > While Lie group diffusion explicitly models $g$, we indirectly model $g$ by capturing how the infinitesimal action of $G$ appears on $X$. The updates in our method follow paths determined by the Lie algebra action, ensuring that the diffusion trajectory mirrors the group’s intrinsic motion. This is important not only because the "final sample" $x$ corresponds to a group element $g$ (that is, we can recover $g$ from $x$ from inverting the flow $\xi(\mathbf{x})$ on $X$ induced by $\rho_X$, thus recovering $g$ as the fundamental flow coordinates, -- please see the discussion in lines 142-144), but also because the entire diffusion trajectory does as well. Our updates in Euclidean space precisely align with the trajectory of an equivalent motion on the Lie group itself (see Figure 6b, where our model produces trajectories equivalent to group transformations while always remaining in Euclidean space).
> > > > > > > >
> > > > > > > > Furthermore, we can extend our framework to fully describe Lie group diffusion. This requires using an auxiliary space $X$ such that the stabilizers of $G$ are trivial (i.e., the action on $X$ retains all the group’s information). This allows us to generate group elements $g \in G$ through $X$, leveraging the one-to-one correspondence provided by the group action.
> > > > > > > >
> > > > > > > > For instance, in the case of $SO(3)$, we can use its action on $\mathbb{R}^3$ and parametrize it via Euler angles to generate samples from $SO(3)$. This approach satisfies all the necessary conditions (the stabilisers are zero and therefore each point corresponding to a frame rotation of $\mathbb{R}^3$ correspond to a specific rotation matrix which in turn corresponds to a specific $SO(3)$ element) and demonstrates the capability of our method to sample from distributions on $SO(3)$.

---

> ### Author Response · Authors · 2024-12-02
>
> We hope to have addressed the remaining concerns raised by the reviewer, particularly those related to the scope of the paper. Specifically, we emphasize that $X$ being Euclidean is not a weakness but rather the main advantage achieved by of our work. The reviewer has also acknowledged that we successfully resolved their other concerns, including the assumptions required for the validity of the conditions in Section 2.2 (including the counterexample) and the need for homogeneity.
>
> Additionally, we conducted several new experiments to address the reviewer’s request to elaborate on the specific advantages our framework offers over traditional algorithms. Since these points were not revisited in the second round of comments, we assume the reviewer is satisfied with our revision.
>
> We would like to kindly ask the reviewer to reevaluate our work in light of the discussions and the fact that we feel we have thoroughly addressed all the concerns raised.
>
> Should the reviewer have any further questions or require additional clarifications, we would be more than happy to provide them.
>
> Best regards

---

> ### Comment · Reviewer_75NJ · 2024-12-03
>
> Dear Authors
>
> I deeply appreciate appreciate for your reply.
>
> Some of my concerns are fully resolved:
>
> Q2 for X in rotateMNIST: The explanation helps a lot and now I fully understand your setting. I would think it would significantly help the readers if you could add those to the final version. I agree the conditions are not violated.
>
>
> Q3: this fully solves my concerns, thank you! The new experiment is really important and helpful.
>
> Q5:
> > The theoretical analysis and the experiment are not aligned
>
> The authors are totally correct.
>
> > Dimensionality reduction
>
> The authors replies for the MNIST experiment they do not lose expressivity. I totally agree with this. However, if you consider under this setting, then RSGM also has the same benefits of `no need of a tradeoff, as we retain the expressivity of unconstrained models with the benefits of group inductive bias.`. I think at least in the rotate MNIST experiment, this is because of the experiment setting but not the algorithm. But I am partially convinced.
>
>
> I still have the following concerns:
>
> Q1: I am so sorry I am still hard to follow. I spent a lot of time reading the interesting paper and I am getting lost during this rebuttal period. The main contribution is still not unclear to me, due to the fact that the authors are updating the paper, e.g., $X_{diff}$ is newly introduced and did not show up anywhere in the original version. The main contribution troubles me:
>
> Rotated MNIST and CrossDocked2020 are low dim problems, since if you use RSGM, they are 1-dim and 4-dim, respectively.
>
> QM9 and 2d, 3d and 4d distributions: both these 2 tasks can be solved by standard DSM in Euclideans and the improvement is unclear to me.
>
>
> Q4:
>
> This is my main concern: it is unclear to me what G and what group action enables me to use this algorithm.
>
> - $G=U(d)$ and $X=C^d$: the authors claim `In this case we have a complex Lie group`. I agree this is not a good example. However, the difficulty is not from complex numbers but complicated group structures. Let's focus on the next examples.
>
> - $G=GL$: The author states `All the conditions are trivially satisfied since $G'=SO(3)\times R_+$ is a subgroup`. No this is definitely not that trivial. Please take a look at condition 3. GL(n) has dim $n^2$. In the case $n=3$, you need to check all the 9 vector fields commutes. Simply saying the vectors in subgroup G' commutes is not enough.
>
> - What's more, the group action can be very different from matrix-vector product, e.g., G=GL(n) and $X=R^{n\times n}$, the group action is $(A, x)\to Ax$ where $A\in G$ and $x\in X$ and $Ax$ means matrix multiplication of a $n\times n$ matrix (A) and another $n\times n$ matrix (x reshaped to $n\times n$ matrix). Note that X is still an Euclidean space and I am only defining the group action. How can you check this case? I can also have more examples, like conjugation: $(A, x)\to AxA^{-1}$. See https://math.stackexchange.com/questions/1970623/showing-that-conjugation-is-a-group-action
>
>
>
> Q5:
>
> > Higher dim cases `Again, in our revision we provided an experiment on SO(4) and we provided the proof that such setup satisfied our conditions above.` `We provided the proof that the conditions for SO(4) (and for SO(n)) are satisfied in our answer to Q4.`
>
> I am so sorry but are you claiming the calculation in Sec A.5 is finished? I only see the parametrization but not checking the vector field commutes.
>
>
> I am so sorry I cannot improve my rating. I am grateful for your effort but I am still confused in many aspects. I believe some readers may have similar confusion.
>
>
>
> Summary to AC:
>
> Dear AC,
>
> I appreciate your service and hard work in holding such a great conference. In case of I cannot edit my comment after the discussion ends, I will summarize it here:
>
> Strengths: I think the experiment shows the work performed better than RSGM in some cases, and the formulation of the problem is novel.
>
> Weaknesses:
>
> - The writing needs to be significantly improved. I spend a lot of time trying to read the paper and the main contribution is unclear to me until now. The paper is still being updated during the rebuttal period.
>
> - It is unclear in what cases the conditions are satisfied and the conditions are hard to check. This fact also stops higher dim Lie groups and other more complicated Lie groups. The paper only checked SO(3) act on $R^3$ for matrix-vector product, but not higher-dim SO group nor other more complicated cases.
>
> - I am still confused about the problem setup and the main contribution. To me, it seems all the problems solved here can be solved by existing tools. The numerical experiment is good enough to show the algorithm works, but not enough to show the performance is better than existing algorithms, e.g., Euclidean DSM in QM9.
>
> - I would say the work provides a good result. however, this is fully based on a strong assumption (Condition 3) that is hard to check.

---

> > ### Author Response · Authors · 2024-12-03
> >
> > >  I only see the parametrization but not checking the vector field commutes.
> >
> > The reviewer wishes to see explicitly that the commutors of the differential operators for the $SO(4)$ case vanishes, so we present here the result (we emphasise that we cannot update our manuscript anymore since the November 27th deadline).
> >
> > We present here the three non-trivial commutators (note that an operators with itself always commute). (We apologise for the formatting, which is not that trivial using multi-lines equations in OpenReview)
> >
> > First, we list the differential operators
> > $$
> > L_{\varphi_1} = \frac{1}{\sqrt{x_2^2+x_3^2+x_4^2}} \left[ x_1x_2 \partial_2 + x_1x_3\partial_3 +x_1x_4 \partial_4 - (x_2^2+x_3^2+x_4^2)\partial_1  \right]
> > $$
> > $$
> > L_{\varphi_2} = \frac{1}{\sqrt{x_3^2+x_4^2}} \left[  x_2x_3\partial_3 +x_2x_4 \partial_4 - (x_3^2+x_4^2)\partial_2  \right]
> > $$
> > $$
> > L_{\varphi_3} = x_3\partial_4 -x_4 \partial_3~.
> > $$
> > where we used the notation $\partial_i = \partial_{x_i}$. These can be obtain from the Lie algebra matrices we listed in the appendix (equation (34)) and the equation $L_{\varphi_i} = A_{\varphi_i} \boldsymbol{x} \cdot \nabla$, and using the relations
> > $$
> > \sin\varphi_3= \frac{x_4}{\sqrt{x_3^2+x_4^2}}, \quad \cos\varphi_3= \frac{x_3}{\sqrt{x_3^2+x_4^2}},
> > \quad \cos\varphi_2= \frac{x_2}{\sqrt{x_2^2+x_3^2+x_4^2}}, \quad \sin\varphi_2= \frac{\sqrt{x_3^2+x_4^2}}{\sqrt{x_2^2+x_3^2+x_4^2}},
> > $$
> > The first commutator is
> > $$
> > [L_{\varphi_2}, L_{\varphi_3}]  =\frac{1}{\sqrt{x_3^2+x_4^2}} [  -x_2x_4\partial_3 +x_2x_3 \partial_4]   - \frac{-x_3x_4+x_4x_3}{({x_3^2+x_4^2})^{1/2}} \left[  x_2x_3\partial_3 +x_2x_4 \partial_4 - (x_3^2+x_4^2)\partial_2  \right]  - \frac{1}{\sqrt{x_3^2+x_4^2}}\left[ x_2x_3\partial_3 -2x_3x_4\partial_2 - x_2x_4\partial_4 +2x_3x_4\partial_1   \right]
> > $$
> > $$
> >   =\frac{1}{\sqrt{x_2^2+x_3^2+x_4^2}} \left[  x_1x_3\partial_4 -x_1x_4 \partial_3\right] - \frac{1}{\sqrt{x_2^2+x_3^2+x_4^2}}\left[ x_1x_3\partial_4 -2x_3x_4^2\partial_1 - x_1x_4\partial_3 +2x_3x_4\partial_1  \right]=0
> > $$
> > The second one is
> > $$
> > [L_{\varphi_1}, L_{\varphi_3}]  =\frac{1}{\sqrt{x_2^2+x_3^2+x_4^2}} \left[  x_1x_3\partial_4 -x_1x_4 \partial_3\right]- \frac{-x_3x_4+x_4x_3}{({x_2^2+x_3^2+x_4^2})^{3/2}}\left[ x_1x_2 \partial_2 + x_1x_3\partial_3 +x_1x_4 \partial_4 - (x_2^2+x_3^2+x_4^2)\partial_1  \right]
> > $$
> > $$
> >  \qquad\qquad- \frac{1}{\sqrt{x_2^2+x_3^2+x_4^2}}\left[ x_1x_3\partial_4 -2x_3x_4\partial_1 - x_1x_4\partial_3 +2x_3x_4\partial_1   \right] $$
> > $$
> >  \quad\qquad =\frac{1}{\sqrt{x_2^2+x_3^2+x_4^2}} \left[  x_1x_3\partial_4 -x_1x_4 \partial_3\right] - \frac{1}{\sqrt{x_2^2+x_3^2+x_4^2}}\left[ x_1x_3\partial_4 -2x_3x_4\partial_1 - x_1x_4\partial_3 +2x_3x_4\partial_1  \right]=0
> > $$
> > and finally
> > $$
> > [L_{\varphi_1}, L_{\varphi_2}]  = \frac{1}{\sqrt{x_2^2+x_3^2+x_4^2}} \frac{x_1x_2}{\sqrt{x_3^2+x_4^2}} \left[ x_3\partial_3+x_4\partial_4 \right] +\frac{1}{\sqrt{x_2^2+x_3^2+x_4^2}}  \frac{-x_1x_3^2-x_1x_4^2}{(x_3^2+x_4^2)^{3/2}}  \left[  x_2x_3\partial_3 +x_2x_4 \partial_4 - (x_3^2+x_4^2)\partial_2  \right]
> > $$
> > $$
> >  \qquad\qquad+\frac{1}{\sqrt{x_2^2+x_3^2+x_4^2}}  \frac{1}{(x_3^2+x_4^2)^{1/2}}  \left[  x_1x_2x_3\partial_3  -2x_1x_3^2\partial_2-2x_1x_4^2\partial_2+ x_1x_2x_4 \partial_4   \right]
> > $$
> > $$
> >  \qquad\quad- \frac{1}{(x_3^2+x_4^2)^{1/2}} \frac{-x_2x_4^2-x_2x_3^2+(x_3^2+x_4^2)x_2}{({x_2^2+x_3^2+x_4^2})^{3/2}}\left[ x_1x_2 \partial_2 + x_1x_3\partial_3 +x_1x_4 \partial_4 - (x_2^2+x_3^2+x_4^2)\partial_1  \right]
> > $$
> > $$
> > \qquad\quad- \frac{1}{(x_3^2+x_4^2)^{1/2}}\frac{1}{\sqrt{x_2^2+x_3^2+x_4^2}}\left[ x_1x_2x_4\partial_4 -2x_2x_4^2\partial_1 + x_1x_2x_3\partial_3 -2x_2x_3^2\partial_1
> >  -(x_3^2 +x_4^2)(x_1\partial_2 - 2x_2 \partial_1)  \right]
> > $$
> > $$
> >   \qquad\quad = \frac{1}{\sqrt{x_2^2+x_3^2+x_4^2}}  \frac{-x_1(x_3^2+x_4^2)}{(x_3^2+x_4^2)^{3/2}}  \left[  x_2x_3\partial_3 +x_2x_4 \partial_4 - (x_3^2+x_4^2)\partial_2  \right] + \frac{1}{\sqrt{x_2^2+x_3^2+x_4^2}}  \frac{1}{(x_3^2+x_4^2)^{1/2}}  \left[  x_1x_2x_3\partial_3  -2x_1x_3^2\partial_2-2x_1x_4^2\partial_2+ x_1x_2x_4 \partial_4   \right]
> > $$
> > $$
> >   \qquad\qquad- \frac{1}{(x_3^2+x_4^2)^{1/2}}\frac{1}{\sqrt{x_2^2+x_3^2+x_4^2}}\left[  -2x_2x_4^2\partial_1 -2x_2x_3^2\partial_1
> >  -(x_3^2 +x_4^2)(x_1\partial_2 - 2x_2 \partial_1)  \right]
> > $$
> >    $$
> > \qquad\quad = \frac{1}{\sqrt{x_2^2+x_3^2+x_4^2}}  \frac{-x_1}{(x_3^2+x_4^2)^{1/2}}  \left[  - (x_3^2+x_4^2)\partial_2  \right]+\frac{1}{\sqrt{x_2^2+x_3^2+x_4^2}}  \frac{1}{(x_3^2+x_4^2)^{1/2}}  \left[   -2x_1x_3^2\partial_2-2x_1x_4^2\partial_2  \right]
> > $$
> > $$\qquad\qquad - \frac{1}{(x_3^2+x_4^2)^{1/2}}\frac{1}{\sqrt{x_2^2+x_3^2+x_4^2}}\left[
> >  -(x_3^2 +x_4^2)x_1\partial_2   \right]=0
> > $$

---

> ### Author Response · Authors · 2024-12-03
>
> Regarding your statements:
> > *It is unclear in what cases the conditions are satisfied, and the conditions are hard to check.*
>
> > *[...] this is fully based on a strong assumption (Condition 3) that is hard to check.*
>
> We find these comment a bit difficult to reconcile with the rest of your feedback. The conditions are stated explicitly in the form of Theorems and Lemmas, complete with rigorous proofs. Furthermore, you have indicated a confidence level of 5, suggesting that you have carefully reviewed and are satisfied with the correctness of these results. Additionally, you acknowledged in earlier comments that our counterexample resolved your concerns about the validity of these conditions.
>
> We also wish to highlight that the conditions were explicitly verified for all the groups and examples analyzed in the paper. These include SO}(2), SO(3), SO(4) (please refer to our latest comment above), T(N), the dilation group in $N$-dimensions, and SE(3). These groups are the standard settings considered in almost all the works on diffusion on manifolds (SO(4) being a relatively novel inclusion). Given this, we feel that the assertion that the conditions are "hard to check" is somewhat unfair, as we have demonstrated their applicability to all common cases and **explicitly verified them in every experiment included in the paper.**
>
> If the formal conditions are too abstract for the reviewer, here is a more descriptive summary of what they imply:
>
> 1. **Identify the space $X$:**
>    Consider $X$ as the space where diffusion is performed. This is not necessarily the space where the data resides, but rather the space where the network outputs. We called this $X_{diff}$ in our previous replies while we named $X_{data}$ where the data live ($\dim X_{data}$ is the dimension for the input of the network) For instance:
>    - $X$ has dimension 1 for the MNIST experiment,
>    - $X$ has dimension 6 for \(\mathrm{SE}(3)\) in CrossDocked,
>    - $X$ has dimension \(3 \times \text{\# nodes}\) for QM9.
>
> 2. **Dimension matching of the group action:**
>    Ensure that the dimension of the image of the group action satisfies $\dim \rho_X(G) \geq \dim X$. This step is often straightforward, as it involves computing the kernel of the group action (i.e., the group elements whose action is the identity element on $X$) and ensuring the dimension of its complement matches or exceeds $\dim X$, $\dim \rho_X(G) \geq \dim X$.
>
> 3. **Group coverage of the space:**
>    Verify that the group "covers" the entire space $X$. For example, T(N) generates all of $\mathbb{R}^N$, and SO(2) generates all rotations of an image. This property is usually so obvious that it doesn’t require explicit mention.
>
> 4. **Lie algebra differential operator commutators:**
>    Compute the action of the Lie algebra differential operators and check that the commutators vanish. While this calculation can sometimes be lengthy, it follows systematically from the representation of the group action.
>
> We hope that the points above, while slightly informal and not capturing the full mathematical rigor of our proofs, help convey the intuition behind our conditions. Most importantly, we aim to show that these conditions are *natural* and not inherently "strong" or "hard to check."
>
> ----------------------------
>
> Your suggestion to include additional examples involving more unusual groups, such as GL(n) or the Unitary group, is certainly interesting. However, this suggestion comes very late in the rebuttal process, at a point when we can no longer update the paper. While we acknowledge the merit of exploring such examples, we also believe this request may go beyond the scope of what is reasonable within the current review cycle, especially considering the extensive effort we have already made to address your concerns and perform additional experiments.

---

> > ### Author Response · Authors · 2024-12-03
> >
> > Regarding your statement:
> > > *[...] To me, it seems all the problems solved here can be solved by existing tools.*
> >
> > We fully agree that the problems addressed by our work can also be solved by other methods. At no point have we claimed otherwise. In fact, we have made a substantial effort to benchmark our method against other strategies. The focus of our work is to present an additional, novel tool for the generative AI practitioner’s toolkit and to demonstrate the theoretical and practical advantages of our approach compared to existing methods.
> >
> > For instance, compared to RSGM, our method achieves the same tasks without requiring the sampling of trajectories during training and fully relies on explicit score matching instead of implicit score matching. Additionally, in our CrossDocked experiment we found that the task was better performed by our model with respect to RSGM.
> > Compared to Brownian Bridge Diffusion Model (BBDM) we solve a problem better and in a much lower dimensionality.
> >
> > Regarding the statement:
> > > *The numerical experiment is good enough to show the algorithm works, but not enough to show the performance is better than existing algorithms, e.g., Euclidean DSM in QM9.*
> >
> > We are unclear why the comparison with Euclidean DSM on QM9 is being emphasized in a negative connotation, as we have repeatedly stated that *the objective of that experiment was not to demonstrate superior performance over Euclidean DSM*. As noted in our responses, **Euclidean DSM already falls within our framework** with the choice $G = T(N)$, the translation group. The purpose of the QM9 experiment (as well as the 2d,3d) was to illustrate that different choices of the group $G$, provided they satisfy the conditions for generalized score matching, can successfully solve the problem. This demonstrates the versatility of our approach rather than an attempt to claim superiority in that specific case.
> >
> > Additionally, as you acknowledged:
> >
> > > *[...] the experiment shows the work performed better than RSGM in some cases.*
> >
> > This highlights that our method outperforms RSGM in the CrossDocked experiment. Furthermore, it is evident from the results that our method also outperforms BBBB in the MNIST experiment.
> >
> > We hope this clarifies the scope and intent behind our experiments.

---

> > > ### Author Response · Authors · 2024-12-04
> > >
> > > Despite our multiple attempts to clarify the motivation and main contribution of our work throughout this rebuttal, the reviewer appears to still misunderstand the main point of the paper.
> > >
> > > In response to the reviewer’s initial comments, we kindly requested feedback on whether our explanation was clear enough and we could agree on the main goal of the paper, as establishing common ground is critical for a constructive rebuttal. However, the reviewer did not engage in this discussion, responding instead with:
> > >
> > > > *I am still confused about the problem setup and the main contribution.*
> > >
> > > We will now attempt to address this point again in a very direct and concrete manner, even at the risk of oversimplifying our method.
> > >
> > > ### Main Contribution: Lie Group Diffusion in Euclidean Space
> > >
> > > **We achieve Lie group diffusion while staying in Euclidean space.** Every problem solvable by RSGM (when the manifold is a Lie group $G$) can also be solved using our method. The key difference lies in **how** the problem is solved and the computational tools required.
> > >
> > >
> > > Current methods for Lie group diffusion learn distributions with support on the Lie group $G$ and the entire noising/denoising process and sampling trajectories take place on $G$. However, in most cases, the data is initially presented in Euclidean space (e.g., point clouds for molecules, pixels for images), so these methods implicitly or explicitly rely on a group action on flat space during processing.
> > >
> > > Our method achieves the same goal of learning distributions with support on the Lie group $G$, but with a critical difference:
> > > - The noising/denoising process and sampling trajectories are supported in a Euclidean space $X$ on which $G$ admits an action.
> > > - Importantly, the sampling trajectories approximately follow the same paths they would on the Lie group itself.
> > >
> > > ### How do we achieve this?
> > >
> > > Specifically, **our approach can be seen as a second-order approximation of Lie group diffusion in Euclidean space through the group action**, where the Lie algebra update provides the first-order approximation, and the Casimir terms introduce a second-order correction. Explicitly, the group action gives us a map between $G$ and $X$, and its differential between $\mathfrak{g}=TG$ and $TX$. **We use these maps to construct a diffusion process on $X$ analogous to the one on $G$.**
> > >
> > > Despite this approximation, our method effectively learns distributions on Lie groups, as shown in the examples and experiments in the paper. The generated trajectories closely follow paths induced by Lie group elements, approximating the exact trajectories of Lie group diffusion. Moreover, our approach has important advantages:
> > > - It allows for an exact solution of the forward SDE, eliminating the need to sample trajectories during training.
> > > - It fully utilizes explicit score matching for the backward process, avoiding the complexities of implicit score matching
> > >
> > > We will make this distinction more explicit in the main text when discussing the relationship between our method and Lie group diffusion.
> > >
> > > As mentioned above, this intuitive description of our contribution may slightly understate the full scope of the results presented in our paper. However, it should make it very clear in which scenarios our method is applicable and why it is valuable.

---

> > > > ### Author Response · Authors · 2024-12-04
> > > >
> > > > Dear Reviewer 75NJ,
> > > >
> > > > Thank you very much for your second response and for acknowledging that we have successfully addressed several of your concerns.
> > > >
> > > > If we understand correctly, your remaining concerns pertained to the motivation and applicability of the method, as well as some technical aspects related to how easily the conditions for generalized score-matching can be verified. We have addressed these points again in detail in our latest replies, including presenting the SO(4) commutation relations as requested.
> > > >
> > > > Despite this, we are finding it difficult to understand why the significant improvements to the manuscript and the resolution of issues raised during the rebuttal period are not being reflected in a re-evaluation of our work.
> > > >
> > > > We would also like to note that we are unable to update the manuscript further (we will have a change however to make a final revision for the final version). As per the guidelines, the last date for uploading a revised PDF was November 26th. After this date, we have only been permitted to post replies on the forum without any changes to the manuscript itself. Unfortunately, we received your comments on December 1st, leaving us with just two days to respond during the rebuttal period.
> > > >
> > > > We would like to highlight the following statements from your comments:
> > > >
> > > > - > *"Experiment shows the work performed better than RSGM in some cases, and the formulation of the problem is novel."*
> > > > - > *"The work provides a good result."*
> > > >
> > > > These remarks seem to indicate that we have successfully conveyed the value of our contribution. Combined with your assessment of a score of 4 for presentation and 3 for soundness, we believe that an overall score of 3 does not objectively align with a fair evaluation of the individual aspects of the paper.
> > > >
> > > > We sincerely hope you will reconsider these points and reevaluate our submission in light of the discussions during the rebuttal period and the improvements we have made to address your concerns.
> > > >
> > > > Finally, we would like to point out that while reviewers cannot post further comments on the forum, it is still possible to modify the original review and adjust scores accordingly.
> > > >
> > > > Thank you once again for your feedback and for the time you have dedicated to reviewing our work.
> > > >
> > > > Best regards,
> > > > The authors

---

### Official Review · Reviewer_T87W · 2024-11-03

**Soundness:** 3
**Presentation:** 3
**Contribution:** 4
**Rating:** 8
**Confidence:** 4

**Summary:**

The paper introduces score-based generative models whose solutions abide to some symmetries, known as Lie groups. To achieve this, the authors show that diffusions in a space $ X $, which follow these symmetries, can be expressed using a directional derivative aligned with the symmetry. In particular, they shows that the stochastic differential equations of such diffusions includes a directional derivative of the log-density rather than the standard log-density gradient. Consequently, they propose a method for training the directional scores.

A symmetry group is a set of (closed) transformations on a space $ X $ that maintains certain symmetries, such as translation, rotation, and reflection. Specifically, a Lie group is a symmetry group with continuous symmetries. Each element (transformation/mapping) in a Lie group is known as a (Lie) group action. Since Lie group actions are continuous, they are differentiable with respect to the symmetries, such as rotations or translations. The derivative of a Lie group action is called a Lie algebra action, which is commonly viewed as a linearized form of the group action (analogous to any other derivative).

The paper highlights that the directional derivative at $ x \in X $, along group actions, can be represented by what is called the fundamental vector field. The fundamental vector field at $ x \in X $ is the linear mapping at $ x $, representing the Lie algebra actions (the derivatives of the symmetries) with respect to a basis of $ X $, and it is expressed through exponential mappings of Lie algebra actions. As a result, the directional derivative on $ X $ along the symmetries is defined by the inner product of the fundamental vector field and the ordinary derivative.

With this derivative representation of using fundamental vector fields, the authors show that forward and reverse diffusions under Lie groups can be represented in terms of the directional derivative of log-density. Based on this, the paper proposes generalized score matching (including a denoising variant) that learns the directional derivative of log-density in a manner similar to standard (denoising) score matching.

The authors argue that the proposed method offers an advantage by allowing the use of more general network architectures, without the need for designing network structures specifically invariant or equivariant to Lie groups.

Finally, the paper demonstrates the effectiveness of this approach through various experiments.

----
I have updated the overall rating from 6 to 8 and the soundness from 3 to 4 after the authors' rebuttal.

**Strengths:**

While the paper may not be entirely novel, it is original, beneficial, and contributes positively to the community. It bears some resemblance to diffusion or flow matching on Riemannian manifolds because a Lie group is a type of Riemannian manifold. Therefore, one might suggest using the previously proposed Riemannian diffusion models. However, the paper does an excellent job of explaining how the new differential equation contains directional derivatives along Lie groups and why the directional derivatives to the group action have a Jacobian-like term, i.e., the fundamental vector field. Such discussions can be considered original because it differs from simply showing a special case as an example from a general method.

The paper is written in a way that is easy for others to understand. It is beneficial, interesting, and well-explained.

**Weaknesses:**

I find the paper original and significant overall. However, the presentation should be improved.

In particular, the explanations of essential concepts in the preliminary section are too similar to the definitions and phrases found on Wikipedia. For example, the sections on Lie groups and Lie algebras closely mirror Wikipedia's wording. Wikipedia, while useful for quick references, is not always reliable or sufficiently detailed for scholarly work. It may lack the rigorous explanations required to fully grasp complex mathematical ideas. By closely copying Wikipedia's definitions—including any inaccuracies or omissions—the paper may fail to convey the nuances of the concepts involved.

The presence of typos in the equations is another area for improvement. Accuracy in mathematical expressions is crucial, as even minor errors can lead to misunderstandings. Careful proofreading and attention to detail in the equations would improve the clarity and overall quality of the paper.

**Questions:**

N/A

---

> ### Author Response · Authors · 2024-11-21
>
> We thanks the reviewer for taking the time to read and provide comments to our manuscript. We are looking forward to a constructive rebuttal period and we are confident that the paper will improve through the exchange.
>
> We have improved the presentation of the paper. The changes can be seen in the newly updated version of the paper (highlighted in blue). Among them:
> - We unified the notation for action of a group (representation) on a manifold.
> - We defined $\boldsymbol(\Pi)$ more clearly in line 143.
> - We defined $\boldsymbol{A}$ in line 178
> - We corrected the notation $(x,y) \rightarrow (x_1,x_2)$ in the SO(2) example around equations (2) and (3)
> - We unified the notation $x \rightarrow \boldsymbol{x}$ for vector quantities.
> - We made clear the assumptions about the curvature of $X$. The conditions of Section 2.2 hold for any differentiable $X$, while the Theorem 3.1 shows how we can obtain a curved Lie group dynamics while remaining in Euclidean space.
> - **New Experiments**
>    - Higher dimension Lie group: We added an experiment with G=SO(4) to show that our formalism works with any Lie group beyond the usual SO(2) and SO(3)
>    - We added a benchmark to the MNIST experiment. We trained a Brownian Bridge Diffusion model (BBDM) [1] on the rotated MNIST dataset from Section 5.2.
> The BBDM implements a continuous-time stochastic process in which the probability distribution during the diffusion process is conditioned on the start $x(0) \sim q(x_0)$ and end point $x(T) \sim p(x_T)$, where an intermediate point $x(t) \sim p(x_t| x_0, x_T, t) = N(x_t, \mu(t, x_0, x_T), \Sigma(t)$ can be simply drawn since the mean function depends on the pairs $(x_0, x_T)$, see Eq. 3 and 4 in reference [1].  We observe that the BBDM model is in some cases not able to reconstruct a rotated MNIST digit, see Figure 6a, digits: (2,4,5,6,7) but removes/adds pixels in the image and in some cases transitions from a rotated digit to another digit as shown in the example from 9 to 4 in the last row of Figure 6b in the updated manuscript. This artifact is most likely due to the way the BBDM is trained by pinning two endpoints and allowing to change the (latent) representation in the full 784-dimensional space.
>
> We added these new experiments in the main text of the paper.
>
>
> [1] BBDM: Image-to-image Translation with Brownian Bridge Diffusion Models. Bo Li, Kaitao Xue, Bin Liu, Yu-Kun Lai, CVPR 2023

---

> > ### Comment · Reviewer_T87W · 2024-11-25
> >
> > Thank you for addressing my concerns in your revised manuscript. I acknowledge the improvements made, particularly in enhancing the clarity of the paper's presentation. I appreciate your efforts and am pleased to reflect these revisions in my updated evaluation.

---

> > > ### Author Response · Authors · 2024-11-26
> > >
> > > We thank the reviewer for their feedback and for recognizing the improvements made in the revised manuscript. Your comments have been instructive and key in helping us improve the paper and we truly appreciate your acknowledgment of our efforts. We are glad the revisions meet the expectations of the reviewer, and we look forward to any further comments you may have.

---

### Official Review · Reviewer_3xv7 · 2024-11-04

**Soundness:** 2
**Presentation:** 2
**Contribution:** 2
**Rating:** 3
**Confidence:** 3

**Summary:**

This paper proposes a new class of diffusion generative models, where the forward process is defined through infinitetismal action of Lie algebra, to better model the underlying symmetry of data distribution. This also generalize the standard denoising score matching. Experiments are performed on rotated MNIST images, molecular conformer generation and molecular docking to demonstrate the effectiveness of the approach.

**Strengths:**

* The authors use nice and informative figures (Fig 1 - Fig 3) to demonstrate the complicated geometric object related to Lie groups and have provided some level of intuition with them.
* This paper has technically sound mathematical formulation and proof. The introduction of new SDEs based on Lie algbera is novel.
* This paper provides numerical experiments on molecular conformer generation and molecular docking, which are important tasks related to structural biology.

**Weaknesses:**

* The paper does not provide a clear motivation on why and in what cases we should use this new class of SDE instead of standard Euclidean Diffusion models. Is it when the data distribution is invariant under some Lie group action? Is it when the data distribution is defined on Lie groups? This part is not clearly explained, which further makes it confusing under what scenarios using Lie-induced generalized score matching are more beneficial. The paper misses a detailed discussion on this part, which I believe is very important.

* The paper does not discuss how restrictive the technical assumptions are. In section 2.2, three conditions are presented as sufficient conditions to use generalized score matching. However, the general feasibility of these conditions is not well discussed. Furthermore, while the general setting assumes the data state space $X$ to be general manifolds, all the examples and calculations in the paper seem to be done in cases where $X$ is essentially Euclidean. This makes me wonder about the applicability of this approach to more general settings.

* The paper does not discuss the tractability of different score matching objectives in the general setting. In lines 293-304, seemingly only settings of Euclidean data are presented, where the generalized conditional score has an analytic closed-form expression. It's unclear how things are if the data space is not Euclidean.

* The numerical experiments, while are important real-world applications, are not performed enough and comparison with benchmarks in the literature are missing. Molecular conformer generation and molecular docking respectively have a rich literature and many of them are based on generative modeling methods. Without comparison, it's hard to see the utility and advantage of the proposed approach.

* Paper writing clarity can be improved. Multiple typos and citations in inappropriate form could decrease paper readability. Missing some figure captions (e.g., Figure 5)

**Questions:**

* Can you elaborate more on in what scenarios, the proposed Lie algbera based SDE and Lie-induced generalized score matching are more beneficial than other approaches, e.g. vanilla Euclidean diffusion (with or without equivariance/invariance enforced), manifold diffusion models, etc?
* Can you provide more examples where the proposed framework is feasbile and the data space $X$ is not Euclidean?
* How to choose the drift function $f(x)$ in eq. (6), (7), and how does the choice relate to the associated Lie group $G$ in the problem set-up?
* Can you elaborate more on Line 400-404, specifically on how RSGM is a special case of the proposed framework?
* What's the connection between this work and diffusion models on general Lie groups, for example [1]?
* Given a data distribution, how do we know what Lie group $G$ to choose for using the proposed approach? And if the data distribution is invariant under the group action in $G$, does the proposed approach also guarantee to preserve such invariance?
* Would the proposed approach be applicable if the associated Lie group is high dimensional, say $G = SO(n)$ for $n > 3$?

* In the QM9 molecular conformer experiment, how would the proposed approach compare in terms of numerical performance with other methods, e.g. [2-4]?
* In the molecular docking experiment, how would the proposed approach compare in terms of numerical performance with other methods, e.g. [5]?

Reference:

[1] Zhu, Yuchen, et al. "Trivialized Momentum Facilitates Diffusion Generative Modeling on Lie Groups." arXiv preprint arXiv:2405.16381 (2024).

[2] Wang, Yuyang, et al. "Swallowing the Bitter Pill: Simplified Scalable Conformer Generation." Forty-first International Conference on Machine Learning.

[3] Xu, Minkai, et al. "Geodiff: A geometric diffusion model for molecular conformation generation." arXiv preprint arXiv:2203.02923 (2022).

[4] Jing, Bowen, et al. "Torsional diffusion for molecular conformer generation." Advances in Neural Information Processing Systems 35 (2022): 24240-24253

[5] Corso, Gabriele, et al. "Diffdock: Diffusion steps, twists, and turns for molecular docking." arXiv preprint arXiv:2210.01776 (2022).

---

> ### Author Response · Authors · 2024-11-20
> **Authors' response**
>
> First of all, we thanks the reviewer for taking the time to read and provide comments to our manuscript. We are looking forward to a constructive rebuttal period and we are confident that the paper will improve through the exchange.
>
> We will address in the following both the comments raised in the "Weaknesses" and "Questions" sections.
>
> **Weaknesses**
>
> -------------------
> *Motivation*
>
> **TLDR: We derive a diffusion process in Euclidean space that possesses all the features of (curved) Lie group diffusion. We thus obtain an unconstrained dynamics (as opposed to equivariant neural networks) that maintains the knowledge of the group action on the data.**
>
> The reviewer asks about the motivation of our work. Here we provide a more detailed explanation that hopefully makes the motivation behind our work intuitive.
>
> The idea is that while often the representation of the data is presented in Euclidean space (3d coordinates for point clouds, pixel values for images, etc.), the underlying "true" coordinates might live in a curved space (bond and torsion angles for molecules, global transformation of images/solids, etc.). However, performing diffusion in curved space, while possible, has notable challenges:
> - *Data transformation/preprocessing*. The data needs to be preprocessed and transformed to extract the curved coordinates. This is laborious and difficult to do in general
>  - *Diffusion is more complicated*. Diffusion on curved manifolds includes additional difficulties that related to the curvature of the manifold itself. This is a very interesting field, but it is at its infancy compared to Euclidean diffusion, where lots of theoretical and computation results are readily available and we can leverage them.
>  - *Need of projection*. In order to perform diffusion on a general manifold $X$, we need to impose the condition that the diffusion path is contained within $X$ at all times. This implies the need of a projection (the exponential map in case of Lie groups), which is in general very expensive to compute.
>   - *Need of sampling*. For general curved space there is no exact solution to the SDE. That means that the forward solution (in training) must also be simulated in order to obtain $p(x(t)|x(0))$. We instead derived an exact solution for **any** group $G$ (equation 8).
>
>
> Our approach, therefore, aims at obtaining an equivalent dynamics to curved space diffusion, while remaining in Euclidean space. Thus we have the property of the former, with all the benefits of the latter. In particular, the data does not need to be transformed, we can use all the algorithm developed for Euclidean diffusion, and there is no need for an expensive projection map.
>
> The "curved dynamics" determined by the Lie group enters in the SDE through a first-order term (the Lie algebra elements) and one second-order term (the Casimir element). In general, instead, the projection map is a Taylor expansion with infinite terms.
>
> Moreover, we wish to point out that **standard Euclidean diffusion also corresponds to a choice of Lie group**, so it can also be seen as part of our formalism. So, our proposal is not an alternative to standard Euclidean diffusion, but rather an extending framework.
>
> Finally, we point out that when the group satisfies the condition of Section 2.2, we obtain an **unconstrained** dynamics, that is, not restricted to invariant/equivariant representations of the data.  Our approach can therefore bring the benefits of group inductive bias into such models (for instance, aligning Lie group sub-factors to the desired degrees of freedom of the data, as in the CrossDocked experiment for global symmetries), without loosing the expressivity of unconstrained generative models. This is particularly relevant in light of recent work [1], where it has been shown that unconstrained model outperform equivariant ones. *Our framework unifies the best of both worlds: we obtain an unconstrained dynamics (the network is unconstrained) but we still keep track of the group action on the data.*
>
> [1] Abramson, Josh, et al. "Accurate structure prediction of biomolecular interactions with AlphaFold 3." Nature (2024): 1-3.

---

> ### Author Response · Authors · 2024-11-20
>
> ---------------------
> *How to effectively choose $G$?*
>
> We recall that if the conditions of Section 2.2 are satisfied, any choice of $G$ allows the learning of any distribution with support on $X$ (exactly like standard diffusion). So, our formalism is not a constrained learning.
>
> Practically, it makes sense to choose $G$ as a product $G=G_1\times G_2 \times G_3\times \cdots$, where the specific subgroups act on "meaningful underlying coordinates" of the data. For instance $G_1$ could be related to a specific bond angle (see Figure 4). This means that the corresponding component of the score $s_1$ will guide the learning of that specific angle (*all while remaining in Euclidean space!*).
>
> Another example for a useful choice of $G$ is if we wish to perform conditional learning. This is exemplified in the MNIST experiment. There we chose G=SO(2) to model global rotation of the images (note that these are not symmetries of the distributions). In this case we can learn the conditional distribution $p(MNIST | rotated MNIST)$. This is not possible with standard diffusion, as it necessitates to build a bridge between two non-trivial distributions. **To compare our framework with a known benchmark, we trained a Brownian Bridge Diffusion model (BBDM) [1] on the rotated MNIST dataset from Section 5.2.**
> The BBDM implements a continuous-time stochastic process in which the probability distribution during the diffusion process is conditioned on the start $x(0) \sim q(x_0)$ and end point $x(T) \sim p(x_T)$, where an intermediate point $x(t) \sim p(x_t| x_0, x_T, t) = N(x_t, \mu(t, x_0, x_T), \Sigma(t)$ can be simply drawn since the mean function depends on the pairs $(x_0, x_T)$, see Eq. 3 and 4 in reference [1].
>
> We observe that the BBDM model is in some cases not able to reconstruct a rotated MNIST digit, see Figure 6a, digits: (2,4,5,6,7) but removes/adds pixels in the image and in some cases transitions from a rotated digit to another digit as shown in the example from 9 to 4 in the last row of Figure 6b in the updated manuscript. This artifact is most likely due to the way the BBDM is trained by pinning two endpoints and allowing to change the (latent) representation in the full 784-dimensional space.
>
> We added this new experiment in the main text of the paper.
>
>
> [1] BBDM: Image-to-image Translation with Brownian Bridge Diffusion Models. Bo Li, Kaitao Xue, Bin Liu, Yu-Kun Lai, CVPR 2023
>
> --------------
>
> *Technical assumptions*
>
> The reviewer is correct that this distinction was not consistently emphasized throughout the work. As outlined earlier, our primary goal is to achieve (curved) Lie group dynamics within Euclidean space. However, in the process, we derived sufficient conditions for generalized score matching, which also hold when $X$ is curved. To clarify:
> - The conditions presented in Section 2.2, "Sufficient Conditions for Lie Group-Induced Generalized Score Matching", are valid for a curved $X$ and they are a novel and nontrivial result on its own;
> - However, the main results, including Theorem 3.1, as well as the focus of the rest of the paper, assume $X$ is flat/Euclidean.
>
> We have updated the introduction and relevant sections to make this distinction explicit, ensuring there is no further confusion.
>
> ---------------
>
> *Numerical comparison to existing methods*
>
> We are not exactly sure what the reviewer means by numerical performance, but in the QM9 experiments, we trained the standard Fisher $G_0=T(3)^N$ and proposed diffusion model on $G_1=(SO(3)\times \mathbb{R}^+)^N$ with the *same* network architecture (but different forward as well as backward SDE dynamics) to model molecules' conformers in $X=\mathbb{R}^{3N}$, where $N$ is the number of atoms in a molecule. Hence, the computational complexity with respect to runtime within the same model class should be around the same, where the standard diffusion model on $G_0$ is likely to have faster runtime since less FLOPs are required compared to $G_1$. The purpose of the QM9 experiment was to show that both approaches are applicable to train a generative model for conformer sampling and that therefore we can achieve an unconstrained dynamics with Lie group different than for standard diffusion.
>
> In regards to comparison with other methods:
> - **Compared to GeoDiff [1] our trained models are faster since both were trained on $T=100$ diffusion timesteps as opposed to GeoDiff that was trained with $T=5000$**;
> - Compared to the Torsional Diffusion model [2], their runtime is faster using 20 steps only since they operate on the curved manifold for the torsion angles $X=(\-\pi, \pi)^M$, reducing the dimensionality of the problem, where M is the number of rotatable bonds with $M<3N$ in a molecule. We instead perform diffusion in the whole $\mathbb{R}^{3N}$ space
>
> [1] Xu, Minkai, et al. "Geodiff: A geometric diffusion model for molecular conformation generation."
>
> [2] Jing, Bowen, et al. "Torsional diffusion for molecular conformer generation."

---

> ### Author Response · Authors · 2024-11-20
>
> *Paper writing clarity*.
>
> We thank the reviewer for carefully reading the paper and catching the missing figure caption. We also added in the same figure an experiment with a higher dimension Lie group, namely for $SO(4)$, showing that our methods trivially applies to Lie groups of any dimension.
>
> We also corrected some inconsistent notation, especially in the $SO(2)$ example in Section 2.
>
> -------------
>
> **Questions**
>
> *How to choose the drift function*.
>
> The drift function can be chosen according to the following criteria:
> - Choice of prior distribution: the prior distribution, from which we sample during inference, depends on the drift terms.
> - Tractability of score function: in order to be able to have a tractable score function, we can choose the drift term to be affine in the flow coordinates. This is the assumption we make around equation (10) and (11).
> - A more general drift function is also possible. However, in the score function is not tractable anymore, and we need to perform implicit score matching, along the lines of [1]
>
> In the experiments in Section 5.1, 5,3 and 5.4 we used the "variance-preserving" drift function which resembles an Ornstein-Uhlenbeck process. The forward SDE is stated in Eq. 53 in the Appendix of the submitted manuscript. The time-varying beta function is chosen from a cosine scheduler proposed in [2].
> By the choice of such drift function, the the distribution of the flow coordinates $\boldsymbol{\tau}$ will converge to a standard Gaussian, i.e. $p(\boldsymbol{\tau}_T | \boldsymbol{\tau}(\mathbf{x}_0)) \sim N(0, I)$.
>
>
> [1] Yang Song, Conor Durkan, Iain Murray, and Stefano Ermon. Maximum likelihood training of score-based diffusion models. Advances in neural information processing systems.
>
> [2] Prafulla Dhariwal, Alexander Quinn Nichol. Diffusion Models Beat GANs on Image Synthesis
>
> -------------
>
> *RSGM is a special case of the proposed framework?*.
>
> We think that the answer to this question is already implicit in our description of the motivation above, but we address it again here.
>
> Our framework is not a special case of the diffusion on Riemannian manifolds. If $M$ is a Riemannian manifold, it is more like that when $M=G$, a Lie group, we obtain a (curved) dynamics similar to diffusion on the Lie group
> $G$, but with the benefit of always staying on Euclidean space.
>
> We see therefore our method as a more practical framework than RSGM, since we do not need to involve the projection map and we can use all the results and techniques developed for Euclidean score matching/diffusion.
>
> We made this point clearer in the text when discussing our relationship with that line of work.
>
> -------------------
>
> *Connection between this work and diffusion models on general Lie groups.*
>
> The reviewer mentioned the paper
> [1] Zhu, Yuchen, et al. ”Trivialized Momentum Facilitates Diffusion Generative Modeling on Lie Groups.”, and we agree that we should include a discussion of how our work relates to it. A shorted version of our discussion here is added in the related work section.
>
> The method in that paper leverages the concept of trivialized momentum, where diffusion (noising) is performed on the Lie algebra of the group rather than the Lie group itself. This approach, as is the case for ours, also allows to reduce diffusion on an Euclidean space, since the Lie algebra, being a vector space, is isomorphic to $\mathbb{R}^n$, which also eliminates curvature terms. The diffusion is mapped back to the Lie group $G$ via the (derivative of the) left multiplication map, which connects tangent spaces at different points on $G$.
>
> They also propose a framework for dynamics that remain entirely on the curved manifold without, according to their claims, requiring projections. This is achieved by separating each step into two parts: updating the Lie algebra element $d\xi$ and updating the Lie group element $dg = g \xi$.
>
> This method has, however, severe drawbacks:
> - The main drawback lies in the non-Abelian case, which is the most relevant since the Abelian case essentially reduces to SO(2) or copies of it. For non-Abelian groups, the absence of a conditional transition probability necessitates implicit score matching, requiring divergence computation in the data space which is very expensive. Our method includes both theoretically and practically non-Abelian groups, as demonstrated in the QM9 and CrossDock examples (which include SO(3) factors).
> - Despite the authors' claim that their formalism avoids projection onto the manifold, this is however implicitly still the case. The equation $dg = g_0 \xi$ ($g\in G, \xi\in \mathfrak{g}$) operates on the tangent space of the group, and its solution (neglecting the variation of $\xi$ is $g = \exp(g_0 \xi$. Thus, we need to know and apply the exponential map $\exp: TG \to G$, which is precisely the projection onto the Lie group. Thus, their method also needs the projection onto the group. In our framework, this is not the case.

---

> > ### Author Response · Authors · 2024-11-20
> >
> > *Given a data distribution, how do we know what Lie group to choose for using the proposed approach? And if the
> > data distribution is invariant under the group action in , does the proposed approach also guarantee to preserve such
> > invariance?*
> >
> > (Remark: some of these points have been made above as well, but we address them here again for sake of completeness)
> >
> > If the conditions in Section 2.2 are met, any choice of $G$ enables the **unconstrained** learning of **any** distribution supported on $X$, just like standard diffusion methods (where we choose $G=T(N)$). Therefore, our formalism does not impose constraints on the learning process.
> >
> > In practice, it is often beneficial to select $G$ as a product of subgroups, $G = G_1 \times G_2 \times G_3 \times \cdots$, where each subgroup corresponds to meaningful coordinates in the data. For example, $G_1$ could represent a specific bond angle, as illustrated in Figure 4. This design ensures that the associated score component $s_1$ directly guides the learning of that angle, while operating entirely within Euclidean space.
> >
> > Another practical application of choosing $G$ arises in conditional learning. This is demonstrated in the MNIST experiment, where $G = SO(2)$ was chosen to model the global rotation of images (which are not symmetries of the distribution). This setup allowed us to learn the conditional distribution $p(\text{MNIST} | \text{rotated MNIST})$, a task that standard diffusion methods cannot achieve since they lack the capacity to connect two non-trivial distributions. We mentioned above already that we added a benchmark to our model by training a Brownian Bridge Diffusion model (BBDM).
> >
> > To the question of what happens when the distribution is invariant under a specific group $G_1$ with $\dim G_1 = m$, we can look at what happens in two scenarios:
> > - Standard Diffusion ($T(n)$). In this case we learn a $n=\dim X$ dimensional score, even though we know that the "true" underlying distribution has $n-m$-dimensional support (invariance under $G_1$ gives us $m$ constraints in a $n$-dimensional space). Thus, we overparametrized the space.
> > - We can instead choose $G = G_1 \times G_2$, where $G_2$ is "complementary" to $G_1$, meaning that the whole group satisfies the condition of Section 2.2. In this case we can simply drop $G_1$ because we know that the distribution is invariant under it. We therefore have a learning process in $n-m$ dimension (guided by $G_2$), which corresponds to the true dimensionality of the problem.
> >
> > Example for this can be found in the Experiment section 5.1. Figure 5 (d,e) show distribution that depends only on radius and angle respectively, that is, they are invariant with respect to SO(2) and $\mathbb{R_+}$ respectively. The generated distributions are learned only through a one-dimensional score corresponding to the complementary group in $G=SO(2) \times \mathbb{R_+}$ ($G_2=\mathbb{R_+}$ for Figure 5d and $G_2=SO(2)$ for Figure 5e). With traditional diffusion we are instead forced to learn a 2-dimensional vector field, not leveraging the symmetry property of the problem.
> >
> > -----------------------
> >
> > *Would the proposed approach be applicable if the associated Lie group is high dimensional, say SO(N) for N > 3*
> >
> > Yes, the proposed approach works with *any* Lie group.
> >
> > We added an experiment where we lean a 4d distribution (mixture of Gaussian) in $X=\mathbb{R}^4$ with the group $G=SO(4)\times\mathbb{R_+}$. The Lie algebra elements are described in Appendix A.5, and Figure 5f shows the true and the generated distributions. The figure is depicted as a matrix of all the combinations of 2-dimensional projections of the 4d space.

---

> > > ### Author Response · Authors · 2024-11-26
> > > **Additional baseline experiment for CrossDocked**
> > >
> > > Following the suggestion of the reviewer, we performed a **comparison between our proposed method and diffusion on Lie groups based on the strategy of "Riemannian score-based generative modelling" (RSGM) [1] in the CrossDocked experiment.** We re-implemented the strategy of [1,2,3] for modelling the $T(3)\times SO(3)$ dynamics corresponding to global transformations (translation and rotations) of the ligand. We compared the Root mean square deviation (RMSD) for all the 100 molecules in the test dataset from the two approaches. For each molecule we generated 5 poses and we compared the RMSD with respect to the ground-truth docked ligand. The results are presented in the updated Figure 8. In summary, **our framework achieves a lower (i.e., better) RMSD**, namely in average RMSD=$2.9\pm 1.0$ Å (for ours) vs RMSD=$5.6\pm 1.2$ Å for RSGM. Figure 8b depicts the distributions of the RMSD over the whole test dataset.
> > >
> > > [1] De Bortoli et al, ”Riemannian score-based generative modelling”
> > >
> > > [2] Huang, Chin-Wei, et al. ”Riemannian diffusion models.”
> > >
> > > [3] Corso, Gabriele, et al. "Diffdock: Diffusion steps, twists, and turns for molecular docking."
> > >
> > > ------------------
> > >
> > > Thank you again for your valuable feedback and the time you spent reviewing our submission. Your comments have been instructive and key in helping us improve the paper, and we feel we addressed carefully all the points you raised.
> > >
> > > We hope that our revisions answered your questions and resolve your concerns, and if so, we would greatly appreciate it if you could update your review to reflect this. Of course, we remain available throughout the rebuttal period to address any further questions or comments you may have.
> > >
> > > Thank you for your input and for helping us strengthen the paper.

---

> > > > ### Comment · Reviewer_3xv7 · 2024-12-02
> > > >
> > > > ## Response to rebuttal
> > > >
> > > > I appreciate the author's detailed response. While some of my concerns have been mitigated, I still have some major questions about the contribution and the task targeted by this work.
> > > >
> > > > 1. The author claimed that they derive a diffusion process in Euclidean space that possesses all the features of (curved) Lie group diffusion, and I don't think this is true. I do agree with the authors that when the given data (which is Euclidean) has some true coordinates liven in some known Lie group, the proposed approach could have some benefits. However, what I don't think would work (and is in fact achievable by Lie group diffusion) is that when you are directly given data that are elements in some known Lie groups, and the task would be to generate more of them. For example, if we are given some data that are n by n real matrices in SO(n) with any n, how would this method be able to generate more of these n by n matrices is unclear to me. However, this task that I proposed is potentially achievable by manifold diffusion models like RSGM. I believe the author should be clear about the advantages and disadvantages of their proposed approach and state them in the work without over-claiming their contribution. For example, the author claims in the updated manuscript that they provide "the first result of denoising score-matching result for general non-Abelian groups". To me this is not a valid claim, again since the proposed methods can't generate these manifold elements. I would appreciate if the authors could further clarify how their approach can address the task that I propose here.
> > > >
> > > > 2. The presentation issue is still quite severe in the updated manuscript. I spent quite a good amount of time reading this paper and its updated version, and still I am confused about the exact setup of each experiment. I believe you should at least explicitly describe the space $X$, the Lie group $G$, and what is the data distribution you try to generate, in **all the listed experiments**. In the current version, only very limited experiments have mentioned clearly the used $X$ and $G$ (such as QM9), and almost no experiments have all three elements well explained, especially regarding the data distribution that you try to sample from. For example, I am not sure if the paper has clearly mentioned what is the rotation in this rotated MNIST example, is it a random rotation and if so, what is the distribution of this rotation. Many important details like this are not listed in the main text or Appendix. I think this is extremely important since it relates to the reproducibility of this work, and unfortunately the current version still has lots to improve on this part.
> > > >
> > > > To sum up, while I acknowledged the contribution and some novelty of this work to modeling Euclidean data with some inherent structure characterized by known Lie groups, I don't agree with the reviewer's way of selling them as the proposed approach can handle the generative modeling of any Lie group data (in general, Lie group data does not have to be related to any point cloud or Euclidean data). Therefore, I feel like the discussion and criticism of RSGM, other manifold diffusion mentioned in the work is not completely fair. While there's some overlap in terms of the tasks that can be handled by this approach and RSGM, in general the targeted scenario is very different. The author should be clear about this distinction in the manuscript in order to better stress their own advantages and strengths. Besides, the experiments can be enhanced by including more details and more quantitative comparisons to show more promising results.

---

> > > > > ### Author Response · Authors · 2024-12-02
> > > > >
> > > > > Dear Reviewer 3xv7,
> > > > >
> > > > > Thank you very much for your response and for acknowledging the benefits of our work and that we have successfully addressed most of your concerns!
> > > > >
> > > > > In the following, we will address each point you raised individually.
> > > > >
> > > > > -----------------------
> > > > >
> > > > > The reviewer seem to be skeptical about our statement that our method performing diffusion in Euclidean space possesses all the features of curved Lie diffusion. I hope we can clarify here this misunderstanding.
> > > > >
> > > > > > However, what I don’t think would work (and is in fact achievable by Lie group diffusion)
> > > > > is that when you are directly given data that are elements in some known Lie groups, and the task would be to
> > > > > generate more of them. For example, if we are given some data that are n by n real matrices in SO(n) with any n,
> > > > > how would this method be able to generate more of these n by n matrices is unclear to me. However, this task
> > > > > that I proposed is potentially achievable by manifold diffusion models like RSGM.
> > > > >
> > > > >
> > > > > We claim that the experiment proposed by the reviewer falls within the class of problems that can be solved using our framework. Below, we will explain why this is the case, but first, we would like to clarify a few points.
> > > > >
> > > > > An abstract group is defined by a set of elements and a group operation that satisfies the properties of associativity, the existence of an identity element, and the existence of inverse elements. In most cases, however, whether explicitly or implicitly, a specific representation is chosen to describe the group. We recall that a representation is a map $\rho: G \rightarrow GL(V)$ from the group to a vector space $V$.
> > > > >
> > > > > For example, when the reviewer refers to general elements of $SO(n)$, he is referring to them as matrices, which assumes a representation $\rho: G \to \mathrm{GL}(V)$, where $V$ is a vector space whose dimension matches the representation. For instance, for $SO(3)$, the most commonly used representation is the 3-dimensional one with $3 \times 3$ orthogonal matrices. However, $SO(3)$ has infinitely many irreducible representations, each of dimension $2l+1$ for $l = 0, 1, 2, \ldots$, which would given matrices of dimensions $(2l+1)\times (2l+1))$. While the $l=1$ case is most familiar, other representations (e.g., for describing atomic orbitals with angular momentum quantum number $l$) are also important in specific contexts and use cases.
> > > > >
> > > > > Thus, even in the reviewer's argument, a vector space $V$ has been introduced implicitly through the representation. This vector space $V$ can directly serve as the space where Euclidean diffusion is performed in our framework.
> > > > >
> > > > > Even if we consider a Lie group without referencing a representation or action on a vector space, our framework still allows generation directly on the Lie group. To do this, we choose an "auxiliary" space $X$ such that the stabilizers of $G$ are trivial, meaning that the group action on $X$ retains all information about the group. Such representations are called "faithful." Using this setup, we can generate group elements in $G$ through the auxiliary space $X$ and the one-to-one correspondence provided by the group action. For example, for $\mathrm{SO}(3)$, we can use its action on $\mathbb{R}^3$ with Euler angle parametrization to generate samples directly from $\mathrm{SO}(3)$. This approach satisfies the necessary conditions for generating distributions on $\mathrm{SO}(3)$ using our framework.
> > > > >
> > > > > To make this more concrete, suppose the training set consists of elements from $\mathrm{SO}(3)$ sampled from an unknown distribution $p_{\mathrm{SO}(3)}(R)$. Let consider $\mathbb{R}^3$ with the usual basis frame given by the axis $x,y,z$. Let us call $f$ the frame, given by the collection of the three vectors defining the basis. For each group element in the training set, we apply its representation as a transformation of $f$, $R \mapsto R(f)$. $R(f)$ will then be a new frame of 3 orthonormal vectors. Note that $f$ is determined by 3 parameters (3 vectors have 9 coordinates, but we have 3 relations due to normality and further 3 given by orthogonality). Thus, these coordinates describe $X=\mathbb{R}^3$.
> > > > > This creates a setup suitable for our framework, where diffusion is performed on $R(f(x))$. To generate new samples, we sample a new frame $f'(x)$ and compute $R'$ such that $f' = R'(f)$. The resulting $R'$ is a new element of $\mathrm{SO}(3)$ generated by our method.

---

> > > > > > ### Author Response · Authors · 2024-12-02
> > > > > >
> > > > > > Perhaps a more conceptual description of how our algorithm works might be helpful to understand why the above works, that is, **how can we sample from $G$ while remaining always in Euclidean space.** The key idea is that we access the group indirectly through its action on the space (see Figure 2a). Formally, given $x = \rho_X(g)x_0$, where $x, x_0 \in X$, where $x_0$ is fixed and $\rho_X$ represents the action of $G$ on $X$, sampling $x$ is equivalent to sampling $g$. This works because every $x$ can be expressed in this form for a fixed $x_0$, leveraging the homogeneity property of $X$.
> > > > > >
> > > > > > While Lie group diffusion explicitly models $g$, our method indirectly models $g$ by capturing the infinitesimal action of $G$ on $X$. The updates follow paths determined by the Lie algebra action, ensuring that the diffusion trajectory mirrors the group’s intrinsic motion. This is significant not only because the final sample $x$ corresponds to a group element $g$ — which can be recovered by inverting the flow $\xi(\mathbf{x})$ induced by $\rho_X$ — but also because the entire diffusion trajectory aligns with the dynamics of group transformations. Figure 6b in the MNIST experiment demonstrates this: our model produces trajectories equivalent to Lie group transformations while remaining entirely in Euclidean space.
> > > > > >
> > > > > > This approach is demonstrated in our CrossDocked experiment, where, given a new ligand, our framework generates a trajectory in Euclidean space that corresponds to a trajectory on the Lie group. The final sample corresponds to the Lie group element that transforms the initial pose to the docked pose of the compound.
> > > > > >
> > > > > >
> > > > > > We acknowledge that for some cases our approach can be more complex compared to direct methods like RSGM, particularly for groups where exact computations are feasible. The cases where our approach has an advantage is if the data is in Euclidean space and there is some natural action on a group on it (which is often the case). However, it is important for us to ensure that the capabilities of our framework are properly understood. While we aim to avoid overstating our contributions, we also wish to ensure they are not undervalued.

---

> ### Author Response · Authors · 2024-12-02
>
> > I believe you should at least explicitly describe the space $X$, the Lie group $G$, and what is the data distribution you try to generate, in all the listed experiments.
>
> We thank the reviewer for reading the manuscript so carefully, and we definitely agree that we should be more explicit about the setup for each experiment. Here we list the changes that we will implement in the final version:
> - In Line 446, for the $d=2,3,4$ distributions, the data space is $X\in \{\mathbb{R}^2, \mathbb{R}^3, \mathbb{R}^4\}$ while the group we are dealing with is $G=SO(2) \times \mathbb{R_+}$,  $G=SO(3) \times \mathbb{R_+}$, and $G=SO(4) \times \mathbb{R_+}$.
> - For MNIST we make the following clarifications:
>     - MNIST is a dataset of pictures with distribution $p_{MNIST}(\mathbf{x})$, where $\mathbf{x}\in \mathbb{R}^{28\times 28}$. We call this data space $X_{MNIST} = \mathbb{R}^{28\times 28}$
>     - RotatedMNIST is a dataset of pictures with distribution $p_{RotMNIST}(\mathbf{x})$, where $\mathbf{x}\in \mathbb{R}^{28\times 28}$, whose elements $\mathbf{x}$ consists of random rotation of original MNIST picture, that is, $\mathbf{x} \in p_{RotMNIST}(\mathbf{x})$ iff there exists $R\in SO(2)$ such that $\mathbf{x}= R(\mathbf{x}')$ for $\mathbf{x}'\in p_{MNIST}(\mathbf{x'})$
>     - In our experiment, we want to learn a diffusion process that models the reverse process $\mathbf{x}'\rightarrow \mathbf{x}$, where $\mathbf{x}= R'(\mathbf{x}')\in p_{\text{MNIST}}$ for some $R'\in SO(2)$. Thus, the distribution from which we need to sample to achieve this, and therefore the distribution we wish to learn is the conditional distribution $p(\mathbf{x}| \mathbf{x}'), \mathbf{x}'\in p_{RotMNIST}, \mathbf{x} \in p_{MNIST}$. Note that this corresponds to learn (generate) the $SO(2)$ element $R_{\mathbf{x}'\rightarrow\mathbf{x}}$ that transforms $\mathbf{x}'$ into $\mathbf{x}$. We model this by a one-dimensional process, thus $X_{diff}=\mathbb{R}^1$ and $G=SO(2)$.
> - Thus: Line 461: In this experiment we have $X=\mathbb{R}^1$, $G=SO(2)$ and $p(\mathbf{x} | \mathbf{x}'),\mathbf{x}' \in p_{RotMNIST}, \mathbf{x} \in p_{MNIST}$.
> \item For QM9 we compare two models: for both $X=\mathbb{R}^{3N}$ and with groups $G=T(3)^N$ and $G=(SO(3)\times \mathbb{R_+})^N$. The distribution is $p_{QM9}(x)$.
> - CrossDocked. Line 517: In this experiment we choose $X=\mathbb{R}^6$ and $G=SE(3)$, while the distribution is $p(\mathbf{x}| \mathbf{x}')$ $\mathbf{x}'$ is a ligand in a random pose and position and the protein in its fixed place, $\mathbf{x}$ is the docked ligand into the pocked together with the pocket.

---

> > ### Comment · Reviewer_3xv7 · 2024-12-03
> >
> > Thank you for the detailed clarifications provided in the previous response! I still have some questions and confusion regarding your response, as are described below:
> >
> > 1. In the explanation for generating SO(3) elements from an unknown distribution, can you write everything in detail so that the idea becomes clearer for me? For example, what would be the training dataset that is used in practice (in practice means that they directly relate to the input to the score network)? Is the training dataset just the matrix respresentation of SO(3) elements (so in $\mathbb{R}^{3 \times 3} = \mathbb{R}^{9}$), or is it something else? How do you simulate the backward process, and how do you recover the rotation using sampled data? Also, in the explanation, what is this $x$ in $f(x)$, where $f$ is the frame called by the authors?
> >
> >     I have a guess regarding how this generation of SO(3) works, please correct me if I am wrong. Each $x$ refers to a special choice of a frame, which is a set of 3 orthonormal vectors. The proposed approach can generate a new frame $f'(x)$ conditioned on the value of $x$, then one SO(3) element can be recovered by comparing the frame $f(x)$ and $f'(x)$. The training dataset consisted of many pairs of $(x, f'(x))$ that correspond to different $x$ and different random rotations (do we even need different $x$? Can we just fix one constant frame and generate the rotated frames by these random rotations, then recover the elements by comparing with this fixed frame?). I am not sure if my understanding makes sense but I feel like my guess presented here is a little bit weird, probably due to the fact that I still haven't fully comprehended the meaning of space $X$ and the idea of frames here.
> >
> >     Can you also comment on the practical algorithms for simulating the backward diffusion process in this example? Based on my understanding, the backward diffusion can't be exactly simulated due to the presence of a nonlinear score network, but why wouldn't the numerical error destroy the Lie group structures that you try to somehow preserve?
> >
> > 2. In the rotated MNIST example, I think now I understand that the authors tried to generate a rotation that turned the rotated MNIST into its usual position (am I right?), but what's the input to the neural network (which is 1D as the authors suggest), and what is the relation between this 1D input to the unrotated image $x$ and rotated image $x'$? I guess this is similar to part of the questions in point 1, in the sense that I am still confused by what is $X_{diff}$ (I thought this is the space of data on which group action is performed, but if this is the case, shouldn't this be the space of images which are 784 dimensional?)
> >
> > I really appreciate the authors' effort in providing a better explanation and I am also trying hard to understand the applicable scenarios of this proposed approach. I hope that the authors can clarify these aspects for me.

---

> > > ### Author Response · Authors · 2024-12-03
> > >
> > > The reviewer intuition is pretty much correct about the $SO(3)$ experiment, but we will describe the whole set up in detail.
> > >
> > > Let the dataset $D$ consists of $3\times3$ $SO(3)$ matrices, that is, matrices that satisfy
> > >  $O^\top = O^{-1}$. We can parametrize a given matrix $O\in SO(3)$ by means of 3 orthonormal vectors: let $v_1, v_2, v_3$ be such vectors such that $||v_i||=1$ and $v_i \times v_j =0$, where we denote with $\times$ the vector product, and $O = (v_1, v_2, v_3)$, where $v_i$'s constitute the columns of the matrix $O$.
> > > An interpretation of this is a rotation that bring the frame (i.e., a basis of vectors) (1,0,0), (0,1,0), (0,0,1) to the basis defined by $v_i, v_2, v_3$ (note that $v_3$ is determined from $v_1, v_2$ and the orthonormality conditions.). This is a perfectly good basis since of the orthonormal properties.
> > >
> > > A suitable "sampling distribution" would be $N((1,0,0), \boldsymbol{1})$ for $v_1$, $N((0,1,0), \boldsymbol{1})$ for $v_2$, and $N((0,0,1), \boldsymbol{1})$ for $v_3$ (again this is determined from the other two). A suitable SDE can be of the Ornstein-Uhlenbeck type, which the appropriate initial conditions. Practically, we have the solution of the forward SDE at time $t$ given by
> > > $$
> > > O(t)= R(t) I$$
> > > where
> > > $$
> > > R(t)=\exp(\phi(t) A_\phi + \theta(t) A_\theta)
> > > $$
> > > where $A_\phi$ and $A_\theta$ are given in equation (23). Note that our requirement from above implies $\mathbb{E}[R(T)]=$Identity, where $T$ is the final time step (that is, at $t=T$ we sample around the standard frame (1,0,0), (0,1,0), (0,0,1)), therefore $\theta(T),\phi(T)\in N(0,1)$, and therefore the Ornstein-Uhlenbeck system is suitable, since the drift terms there drive the corresponding coordinates to zero.
> > >
> > > Now, to generate the noised samples, we generate the initial (from the point of view of the forward process, that is, from each sample in the dataset) $\theta(0), \phi(0)$
> > > through the parametrization $v_1=\sin \theta(0)\cos \phi(0)$ and $v_2=\sin \theta(0) \sin \phi(0)$ and $v_3 = \cos\theta(0)$, and we compute/sample $R(t)$ according to the above. $\theta(t)$ and $\phi(t)$ determine the scores (technically, we learn the noise we add when sampling from the corresponding distribution) the network needs to learn according to equations (11) and (12).
> > >
> > > Specifically, the network $\Phi$ takes as inputs
> > > - The noised matrix $O(t)=R(t) I$
> > > - the time step $t$
> > >
> > > and outputs two scores, corresponding to the update according to $A_\phi$ and $A_\theta$.
> > >
> > > Now, during sampling we sample a frame according to the distributions above, that is, we have $O(T)$ determined by vectors $v_1(T), v_2(T), v_3(T)$, these can be fed to the network, which will predict the scores, and we update these vectors (and therefore the matrix they build) according to (for ease of simplicity we left out the Casimir and divergence terms, which would also play a role and are included in our experiments):
> > > $$
> > > v_1(t-1) = \beta(t) (f_\phi A_\phi v_1(t) + f_\theta A_\theta v_1(t)) + \sqrt{\beta(t)} (\eta_\phi A_\phi v_1(t) + \eta_{\theta} A_\theta v_1(t))
> > > $$
> > > and similarly for $v_2$, where $\eta_{\phi,\theta}\in N(0,1)$ are the noise terms, and $\beta(t)$ is the time-scheduler function.
> > >
> > > We hope that this explicit description of the thought experiment proposed by the reviewer elucidates how our algorithm works in practice and clarifies how our method can be used to sample points from a Lie group.

---

> > > ### Author Response · Authors · 2024-12-03
> > > **Comment on the rotated MNIST experiment**
> > >
> > > >In the rotated MNIST example, I think now I understand that the authors tried to generate a rotation that turned the rotated MNIST into its usual position (am I right?), but what's the input to the neural network (which is 1D as the authors suggest), and what is the relation between this 1D input to the unrotated image
> > >  and rotated image
> > > ? I guess this is similar to part of the questions in point 1, in the sense that I am still confused by what is
> > >  (I thought this is the space of data on which group action is performed, but if this is the case, shouldn't this be the space of images which are 784 dimensional?)
> > >
> > > The data space is described by the MNIST image (rotated or unrotated) which  has $28\times28=784$ dimensions determined by the height and width, while each pixel is either set off or on, i.e. (0, 1). Therefore the grid is described in $(x, y)$ coordinates, where each grid-point has a function value $f(x,y) \in$ {0, 1} for $x, y=(1, 2, ..., 28)$. Hence, $\dim(X_\text{data}) = 784$.
> > >
> > > The network receives as input (a rotated or unrotated) MNIST image with that spatial resolution. Notice that the rotation only affects the pixel location $(x,y)$ but not the off/on label. Hence, $X_\text{data} \in \mathbb{R}^{784}$. Since the rotation matrix from $\text{SO}(2)$ is determined by **one** rotation angle, the diffusion dimension is $\dim(X_\text{diffusion}) = 1$.
> > > During training, for a specific timestep $t$, we sample a random noise $\epsilon \sim N(0, 1)$ and scale this noise sample to obtain the perturbed angle $\theta_t= \sigma(t)\epsilon$.
> > >
> > > By scaling we conclude that $\theta_t \sim N(0, \sigma(t)^2)$ which follows the forward SDE as explained Eq. 55 in Appendix E.1 - MNIST. We design the noise schedule in such way so that the terminal variance converges to $\sigma^2(T) = \frac{\pi}{2}$. Notice that $\epsilon$ reveals as (unscaled) 1-dimensional regression target which the score network needs to predict. Hence, for $\sigma(t)$ with $t\approx 0$, the noisy angle $\theta_{t\approx 0} = \epsilon  \sigma(t)$ is almost 0, i.e., the MNIST image is not rotated.
> > >
> > > To obtain a randomly rotated MNIST image, we rotate the original image by that angle $\theta_t$, first converting from radian into degree, and using the torchvision library to create the randomly rotated MNIST digit, which still has the data dimension $28^2=784$. Now the score model (named $s_\theta$) takes as inputs the (rotated) image and the timestep $t$. Based on this inputs the score model needs to **predict** a 1-dimensional score, which we obtain by applying spatial pooling in the last layers of the CNN, to minimize the loss $l = (s_{\theta}(x_\text{MNIST, t}, t) - (-1) \frac{\epsilon}{\sigma(t)})^2$ see Eq. 12 in the manuscript. Hence the output dimension of the score network equals the dimension of the diffusion space. Note we did not include the notation of $X_\text{diff}$ in the current manuscript, since we are not allowed to upload a newer version.
> > >
> > > Now for sampling: we start with a random angle $\theta_T$ from the prior distribution which we determined through the SDE in Appendix E.1 and in our experiments follows $\theta_T \sim N(0, \frac{\pi}{2})$.
> > > We convert this angle into radian and perform the rotation to obtain $x_\text{MNIST, rotated at t=T}$.
> > > Next, the score network inputs that $28\times 28$ image and outputs a 1-dimensional score that goes into the update equation to change the rotation of the current image. We provided the code for this experiment
> > > in \texttt{sism.mnist.train}.
> > >
> > > We want to emphasize that the CrossDocked2020 experiment is similar in the setting. The diffusion space is 6-dimensional since we consider one global rotation and one global translation through the group $G=(\text{SO}(3) \times T(3))$ but the data dimensionality is much larger comprising the protein pocket and the ligand with $3M+3N$ dimension, where $M$ and $N$ are the number of atoms in the protein and ligand. Therefore, $\dim(X_\text{data}) = 3(M+N)$. In similar fashion, the score network inputs the entire point cloud (where only the ligand is SE(3)-transformed, i.e. globally rotated and translated), and the network predicts 6 scores that update the ligand point cloud. Hence, $\dim(X_\text{diffusion})=6$.
> > >
> > > The global translation dynamics is equal to the standard Euclidean diffusion, but the global rotation dynamics is described in Appendix A.4. We provided the code for this experiment
> > > in \texttt{sism.plcomplex.model}.
> > >
> > > We hope that this description of the MNIST experiment explains how our algorithm works in this setting.

---

> > > > ### Author Response · Authors · 2024-12-03
> > > >
> > > > > but why wouldn't the numerical error destroy the Lie group structures that you try to somehow preserve?
> > > >
> > > > This is an excellent point raised by the reviewer.
> > > >
> > > > To address this, let us consider the motion along a single Lie algebra element, say $A_\phi$. From the Lie group action, $\phi$ describes an $\mathrm{SO}(2)$ subgroup, and the full space can be decomposed into $\phi$-orbits—circles parameterized by $\phi$, where the other parameter $\theta$ remains fixed.
> > > >
> > > > When we perform an "update along the $\phi$ direction" with $A_\phi$, the result will slightly deviate from the original orbit. Perhaps this is what the reviewer refers to as numerical error? This is not an issue for us for two main reasons:
> > > > - There is no systematic drift away from the orbit because of the Casimir terms, which compensate for the drift induced by the curvature of the Lie algebra update (this is illustrated in Figure 3).
> > > > - Since we operate in Euclidean space, there is no intrinsic Lie group structure to preserve.
> > > >
> > > > In other words, **our approach can be understood as a second-order approximation of Lie group diffusion mapped onto Euclidean space,** where *the Lie algebra update provides the first-order approximation and the Casimir terms provide the second-order correction.*
> > > >
> > > > It is important to note that, despite this approximation, our method still successfully learns distributions on Lie groups (as demonstrated in the example above and in the experiments in the paper). The trajectories generated by our method follow paths that are induced by Lie group elements. While these trajectories approximate those "exact" of Lie group diffusion, our approach offers significant advantages:
> > > > - It enables an exact solution of the forward SDE, eliminating the need for sampling trajectories during training.
> > > > - It leverages the full power of score matching for the backward process, avoiding the need for implicit score matching.
> > > >
> > > >
> > > > We will clarify this point more explicitly in the main text when discussing the relationship between our work and Lie group diffusion.

---

> > ### Author Response · Authors · 2024-12-03
> > **More details on the datasets used in the Experimental Section**
> >
> > Dear reviewer 3xv7, we'd like to take the chance to further answer your request in explaining the datasets used during our experiments.
> >
> > **Dataset setup**
> >
> > We apologize in advance for the math-latex formatting on the Openreview
> >
> >
> > ---
> >
> > For the 2d distributions in Figure 5a-5c the target distribution to learn is in the case of Figure 5(a,b,c) a mixture of $k=6$ Gaussians each with diagonal covariances. The means are listed column-wise as
> > $
> > \\boldsymbol{\mu}_1 = \\begin{pmatrix}
> > -6.5,  _\\
> > -6.0
> > \\end{pmatrix}^\top
> > $
> > $
> > \\boldsymbol{\mu}_2= \\begin{pmatrix}
> > -2.5, _\\
> > 2.5
> > \\end{pmatrix}^\top
> > $
> > $
> > \\boldsymbol{\mu}_3= \\begin{pmatrix}
> > -0.5,  _\\
> > -0.1
> > \\end{pmatrix}^\top
> > $
> > $
> > \\boldsymbol{\mu}_4= \\begin{pmatrix}
> >  5.5 , _\\
> >  -5.5
> > \\end{pmatrix}^\top
> > $
> > $
> > \\boldsymbol{\mu}_5= \\begin{pmatrix}
> > 6.0 , _\\
> > 6.0
> > \\end{pmatrix}^\top
> > $
> > $
> > \\boldsymbol{\mu}_6 = \\begin{pmatrix}
> > -10.0, _\\
> > 10.0
> > \\end{pmatrix}^\top
> > $
> >
> > with the following diagonal variances
> >
> > ${\boldsymbol{\sigma}}_{11,22,1}= \begin{pmatrix}
> > 1.3823, \\
> > 1.4150
> > \end{pmatrix}^\top
> > $
> >
> > ${\boldsymbol{\sigma}}_{11, 22, 2}= \begin{pmatrix}
> > 0.8829, \\
> > 1.4593
> > \end{pmatrix}^\top
> > $
> >
> > ${\boldsymbol{\sigma}}_{11,22, 3}= \begin{pmatrix}
> >  0.8904, \\
> >  1.1009
> > \end{pmatrix}^\top
> > $
> >
> > ${\boldsymbol{\sigma}}_{11,22, 4}= \begin{pmatrix}
> >  0.7566, \\
> >  1.2936
> > \end{pmatrix}^\top
> > $
> >
> > ${\boldsymbol{\sigma}}_{11,22,5}= \begin{pmatrix}
> > 1.4408,\
> > 0.6332
> > \end{pmatrix}^\top
> > $
> >
> > ${\boldsymbol{\sigma}}_{11,22,6}= \begin{pmatrix}
> > 1.4346,\\
> > 1.0936
> > \end{pmatrix}^\top
> > $
> >
> > Note we created the dataset that each Gaussian only has a diagonal covariance, i.e. $\sigma_{1,2,k}=\sigma_{2,1,k} = 0 ~~\text{for }{k=1,2,3,4,5,6}$.
> >
> > Each Gaussian is weighted by the values $\boldsymbol{\pi} = \begin{pmatrix}
> > 0.5523 & 0.0321 & 0.0963 & 0.0793 & 0.1036 & 0.1364\\
> > \end{pmatrix}^\top$.
> >
> > The motivation for this toy experiment is to showcase that we can learn any 2d distributions and the dataset creation is implemented in \texttt{sism.datasets.generate-mog-2d}
> >
> > ---
> >
> > Figure 5(d) is a radial distribution with 2 concentric circles. Each circle follows a radial distribution $p_k(x) = \frac{1}{2\pi\sigma_k^2}\exp(-\frac{1}{2\sigma_k^2} (||\mathbf{x}||_2 - r_k)^2)$ for $k=1,2$ and $||\mathbf{x}|| = \sqrt{\mathbf{x}^\top \mathbf{x}}$. We choose the radii $r_1=4.0$ and $r_2=8.0$ with variances $\sigma_1^2=\sigma_2^2=0.5$. We equally weigh these two distributions, hence $\boldsymbol{\pi} = \begin{pmatrix}
> >    0.5, & 0.5
> > \end{pmatrix}^\top$. With this experiment we aim to show that the intrinsic dimension is $1$, since the distribution only depends on the norm of the data point $\mathbf{x} \in \mathbb{R}^2$.
> >
> > The 2d dataset creation is implemented in \texttt{sism.datasets.generate-concentric-circle-dataset}
> >
> > ---
> >
> > Figure 5(e): Showcases an angular line distribution where the radius increases from $0.1$ until $3.0$ with angles (in radian) $\boldsymbol{\theta} = \begin{pmatrix}
> >     0.0, & 1.26, & 2.51, & 3.77, & 5.03
> > \end{pmatrix}^\top$ where a random angle is added drawn from $N(0, 0.05^2)$.
> >
> > Therefore the $n=5$ lines showcase the modes with the corresponding angles $\theta_i$.
> >
> > The code for creating the 2d dataset can be found in the Supplementary code under \texttt{sism.datasets.generate-line-distribution}
> >
> > ---
> >
> > Figure 5(f): Showcases the 3-torus distribution in $ \mathbb{R}^3$ which we parameterize with minor/inner radius $r=1.0$ and major/outer radius $R=3.0$. We parameterize the Cartesian coordinates as
> > $\begin{pmatrix}
> >    x(\theta, \varphi),~ \\
> >    y(\theta, \varphi),~  \\
> >    z(\theta, \varphi)
> > \end{pmatrix}^\top =
> >  \begin{pmatrix}
> >     (R + r\cos \theta) \cos \varphi,~ \\
> >     (R + r \cos \theta) \sin \varphi,~ \\
> >     r \sin \theta
> > \end{pmatrix}^\top$
> >
> > using angular coordinates $\theta, \varphi \in [0, 2\pi)$ and the constraint with $r\leq R$.
> >
> > We sample uniformly from $p(\varphi) = \frac{1}{2\pi}$ while to sample uniformly from $p(\theta) = \frac{R + r \cos \theta}{2\pi R}$ we perform rejection sampling as explained in \url{https://math.stackexchange.com/questions/2017079/uniform-random-points-on-a-torus}.
> >
> > We provide the code to generate this 3d dataset in  \texttt{sism.datasets.generate-samples-from-3d-torus}
> >
> > ---
> >
> > Figure 5(g): Showcases the Möbius strip distribution in $ \mathbb{R}^3$. We parameterize the Cartesian coordinates using
> > $\begin{pmatrix}
> >    x(u, v),~  \\
> >    y(u, v),~  \\
> >    z(u, v)
> > \end{pmatrix}^\top= \begin{pmatrix}
> >     (1 + \frac{v}{2}\cos \frac{u}{2})\cos u,~ \\
> >     (1 + \frac{v}{2}\cos \frac{u}{2})\sin u,~ \\
> >     \frac{v}{2}\sin \frac{u}{2},
> > \end{pmatrix}^\top$
> >
> > for $0\leq u \leq 2\pi$ and $-1 \leq v \leq 1$.
> >
> > We unifomly sample the coordinates $(u,v)$ within their ranges and apply the transform to create the 3d Cartesian coordinates.
> >
> > We provide the code to generate this 3d dataset in  \texttt{sism.datasets.generate-samples-from-mobius-strip}
> >
> > ---
> >
> > We will update the dataset section and details in the Appendix of the final version of this with details for the SO(4) experiment.

---

### Official Review · Reviewer_Wa2S · 2024-11-04

**Soundness:** 4
**Presentation:** 3
**Contribution:** 3
**Rating:** 8
**Confidence:** 4

**Summary:**

In this paper, the authors present a general framework for construction diffusion processes on spaces $X$ that are acted on by Lie groups $G$.
Specifically, they show that the Lie algebra infinitesimal actions on that space induce a fundamental vector field $\Pi$ and its associated differential operator $\mathcal{L} = \Pi \cdot \nabla$.
They then define a noising processes on $X$ (Eq 5) via $\Pi$ whose solution can be easily computed, and whose time-reversal is given in Eq 6 and also involves $\Pi$ as well as the generalised score $\mathcal{L} \log p_t$.
They show that generalised score can be learned by minimising the generalised Fisher divergence (Eq 4) akin to standard score matching.
Setting $X=\mathbb{R}^n$ and $G=\mathrm{T}(n)$, one recovers the standard score matching framework.

**Strengths:**

The manuscript is generally well written and organised. I liked that $(X=\mathbb{R}^2, G=\mathrm{SO}(2))$ was used through the manuscript as a running example.

The framework presented is fairly general and elegant.

Although I did not check the proof, the assumptions and results presented seem sound.

The framework is applied to several tasks, including in molecular conformation modelling and docking.

**Weaknesses:**

Not sure whether this is a 'weakness' per se, but this framework feels pretty similar to building a generative model on the group $G$ (_flow coordinates_) itself using the reparameterisation $x(\tau)$ via the flow map (e.g. Cartesian to polar coordinates).
Why not directly construct a generative model on this space by first mapping sample into the Lie algebra? Is the difference that such a map would need to be _global_ whist the flow map here is _local_ (i.e. depends on an $x_0$)?

Could still construct such a generative model without relying on a global diffeomorphism (e.g. Corso et al. (2022), De Bortoli et al. (2022)). These formalisms actually feel very similar to the one presented here.
For the Rotated MNIST experimentation of Section 5.2 it feels pretty natural, and similarly for the docking task of Section 5.4.
What is the advantage in not doing so?

**Questions:**

- Theorem 3.1: The $\Pi$ term is akin to covariance operator term right? (as in e.g. 'Score-based generative modeling through stochastic differential equations' Song et al 2021).
- 273: 'specific order is irrelevant since the Lie algebra generators commute.' -> how restrictive is this? Is this equivalent to assuming commutativity of the Lie group? E.g. does $\mathrm{SO}(3)$ satisfy this?
- 281: Does this mean that one needs to go back and forth the flow coordinates $\tau$ and the original $x$ at sampling time? Or only at the last iteration?
- 291: How efficiently can one compute the flow map? I assume that the action itself is cheaper than computing the exponential map in general?
- Section 5.3: Is the difference in energy differences significative here? What is the reasoning behind this group being beneficial? I would suggest evaluating the samples via PoseCheck/PoseBuster, which would be quite more indicative and robust than just looking at the Rdkit energy.

---

> ### Author Response · Authors · 2024-11-21
>
> First of all, we thanks the reviewer for taking the time to read and provide comments to our manuscript. We are looking forward to a constructive rebuttal period and we are confident that the paper will improve through the exchange.
>
> We will address in the following both the comments raised in the "Weaknesses" and "Questions" sections.
>
> --------------
> **Weaknesses**
> ------------
> ___________
>
>
> *Relation with Lie group/Riemannian diffusion*
>
> **TL;DR: We derive a diffusion process in Euclidean space that possesses all the features of (curved) Lie group diffusion. We thus obtain an unconstrained dynamics (as opposed to equivariant neural networks) that maintains the knowledge of the group action on the data.**
>
> The main motivation for this work was to derive a pure Euclidean dynamics, with all its benefits, having the properties and features of a diffusion on Lie groups. In other words: *Our approach achieves equivalent dynamics to curved space diffusion while staying in Euclidean space.*
>
> The key idea is that, while data is often represented in Euclidean space (e.g., 3D point clouds or pixel values), the underlying "true" coordinates may reside in a curved space (e.g., bond angles for molecules or global transformations of images). Performing diffusion directly on curved spaces, as proposed in the papers the reviewer mentioned, presents significant challenges:
> - *Data transformation*: Extracting curved coordinates requires pre-processing, which is labor-intensive and generally difficult.
> - *Diffusion dynamics is complicated*: Diffusion on curved manifolds is more complex due to curvature-related challenges. While promising, this field is far less developed compared to Euclidean diffusion, where extensive theoretical and computational tools are available.
> - *Projection requirement*: To constrain diffusion to a manifold $X$, projection (e.g., the exponential map for Lie groups) is needed, which is computationally expensive.
> - *Need of sampling*. For general curved space there is no exact solution to the SDE. That means that the forward solution (also in training) must also be simulated in order to obtain $p(x(t)|x(0))$. We derived an exact solution for **any** group $G$ (equation 8).
>
>
> Our framework realizes a curved space dynamics while staying in Euclidean space. This provides the benefits of curved dynamics without requiring data transformation, enables the use of established Euclidean diffusion algorithms, and avoids costly projections.
>
> Curved dynamics emerge through a first-order term (Lie algebra elements) and a specific second-order term (Casimir element), without the need of the infinite Taylor expansion typically needed for projection maps. Furthermore, standard Euclidean diffusion is simply a special case of our framework, corresponding to a specific Lie group choice. Thus, our proposal extends, rather than replaces, traditional Euclidean diffusion.
>
> Finally, we point out that when the group satisfies the condition of Section 2.2, we obtain an unconstrained dynamics, that is, not restricted to invariant/equivariant representations of the data. Our approach can therefore bring the benefits of group inductive bias into such models (for instance, aligning Lie group sub-factors to the desired degrees of freedom of the data, as in the CrossDocked experiment for global symmetries), without loosing the expressivity of unconstrained generative models. This is particularly relevant in light of recent work [1], where it has been shown that unconstrained model outperform equivariant ones. Our framework unifies the best of both worlds: we obtain an unconstrained dynamics (the network is unconstrained) but we still keep track of the group action on the data.
>
> [1] Abramson, Josh, et al. "Accurate structure prediction of biomolecular interactions with AlphaFold 3." Nature (2024): 1-3.
>
> -----------
>
> *Global vs local diffeomorphism*
>
> The reviewer is absolutely correct that for our framework only a local diffemeorphism is necessary. This is explicitly stated in Condition 3 (lines 227-236), i.e., that $\boldsymbol{\Pi}(\boldsymbol{x})$ must form only a locally commuting frame of vector fields.

---

> > ### Author Response · Authors · 2024-11-21
> >
> > **Questions**
> > ------------
> > -----------
> >
> > *Theorem 3.1: The term is akin to covariance operator term right?*
> >
> > There are two "new" terms that appear in our SDEs in Theorem 3.1
> > - The term $\boldsymbol{\Pi} \nabla \boldsymbol{\Pi}$, which appear only in the time-reversed SDE, corresponds to the "divergence" term of [1]. Perhaps this is what the reviewer meant by covariance operator term, since it involves the diffusion covariance matrix $G^\top G$ (in [1]'s notation)?
> > - The truly new term is the Casimir operator. This is new to our formalism and its related to the curved nature of the geodesics of the flow coordinates. This is a second order effect (the Lie algebra is first-order) and compensate the deviation from the orbit due to a displacement along the tangent vector (as depicted in Figure 3).
> >
> >
> > [1] Score-based generative modeling through stochastic differential equations’ Song et al 2021
> >
> > -------------
> >
> > *281: Does this mean that one needs to go back and forth the flow coordinates and the original at sampling time? Or
> > only at the last iteration?*
> >
> > One does not need to go back and forth between the flow and Cartesian coordinates. This is needed only at the first step (during sampling), that is, we sample $\tau(T)$, we thus obtain $x(T)$, and then the whole dynamics is done on the $x(t)$ (recall $x(0)$ is the "denoised" data distribution).
> >
> > -------------
> >
> > *291: How efficiently can one compute the flow map?*
> >
> > Computing the flow map has essentially no cost, as it is a known and hard-coded map. For $SO(2)$, for instance, it involves extracting $r,\theta$ from $x_1, x_2$. The exponential map, however, involves (in general) summing an infinite number of terms, each being a monomial of increasing power in the Lie algebra matrices. For Lie groups of higher dimensions it becomes increasingly more expensive.
> >
> > -----------
> >
> > *Section 5.3: Is the difference in energy differences significative here? What is the reasoning behind this group being
> > beneficial?*
> >
> > The energy difference is not significative, and this is actually the point we are trying to prove. For this example, we are not claiming that $G=(SO(3)\times \mathbb{R_+})^N$ leads to a better learning than $G=(T(3N))$ (standard diffusion). This is instead to provide evidence that  our formalism allows us to learn **any uncostrained and unconditional} distribution** through **any group** satisfying the conditions of Section 2.2. That is, it is not the case that $G$ needs necessarily to align with some internal degrees of freedom in order for us to be able to learn the distribution, but, as long as the conditions of Section 2.2 are satisfied, we can learn any distribution with any choice for $G$.
> >
> > This flexibility in choosing $G$ has however, benefits.
> > In the other experiments (MNIST and CrossDocked) we show the advantage of choosing a specific $G$, namelyto be able to learn a 1-dimensional conditional distribution (MNIST) and to model rigid transformations on SE(3) through a global rotation and translation (CrossDocked).
> >
> > To make our point clear, we added an experiment to our updated manuscript. Namely, for the MNIST experiment, **we trained a Brownian Bridge Diffusion model (BBDM) [1] on the rotated MNIST dataset**  in a $784$ dimensional space. Note that this is the only option since we cannot apply standard diffusion since we are trying to learn a map between two non-trivial distributions.
> > The BBDM implements a continuous-time stochastic
> > process in which the probability distribution during the diffusion process is conditioned on the start $x(0) \sim q(x_0)$ and end point $x(T) \sim p(x_T)$, where an intermediate point $x(t) \sim p(x_t| x_0, x_T, t) = N(x_t, \mu(t, x_0, x_T), \Sigma(t)$ can be simply drawn since the mean function depends on the pairs $(x_0, x_T)$, see Eq. 3 and 4 in reference [1].
> >
> > We observe that the BBDM model is, in some cases, not able to reconstruct a rotated MNIST digit, (see Figure 6a, digits: (2,4,5,6,7) )but removes/adds pixels in the image.  In some cases it even transitions from a rotated digit to another digit as shown in the example (from 9 to 4) in the last row of Figure 6b in the updated manuscript. This artifact is most likely due to the way the BBDM is trained by pinning two endpoints and allowing to change the (latent) representation in the full 784-dimensional space. Thus we explicitly show the advantage of our framework that allows us to work on the exact dimensionality of the problem.
> >
> > [1] BBDM: Image-to-image Translation with Brownian Bridge Diffusion Models. Bo Li, Kaitao Xue, Bin Liu, Yu-Kun Lai, CVPR 2023

---

> > > ### Author Response · Authors · 2024-11-26
> > > **Extra experiment for CrossDocked, comparison with Riemannian diffusion**
> > >
> > > Since the reviewer mentioned the relation of our method with Lie group/Riemannian diffusion, we performed a **comparison between our proposed method and diffusion on Lie groups based on the strategy of "Riemannian score-based generative modelling" (RSGM) [1] in the CrossDocked experiment.** We re-implemented the strategy of [1,2,3] for modelling the $T(3)\times SO(3)$ dynamics corresponding to global transformations (translation and rotations) of the ligand. We compared the Root mean square deviation (RMSD) for all the 100 molecules in the test dataset from the two approaches. For each molecule we generated 5 poses and we compared the RMSD with respect to the ground-truth docked ligand. The results are presented in the updated Figure 8. In summary, **our framework achieves a lower (i.e., better) RMSD**, namely in average RMSD=$2.9\pm 1.0$ Å (for ours) vs RMSD=$5.6\pm 1.2$ Å for RSGM. Figure 8b depicts the distributions of the RMSD over the whole test dataset.
> > >
> > > [1] De Bortoli et al, ”Riemannian score-based generative modelling”
> > >
> > > [2] Huang, Chin-Wei, et al. ”Riemannian diffusion models.”
> > >
> > > [3] Corso, Gabriele, et al. "Diffdock: Diffusion steps, twists, and turns for molecular docking."
> > >
> > > ------------------
> > >
> > > Thank you again for your valuable feedback and the time you spent reviewing our submission. Your comments have been instructive and key in helping us improve the paper, and we feel we addressed carefully all the points you raised. We hope that our revisions answered your questions and, of course, we remain available throughout the rebuttal period to address any further questions or comments you may have.
> > >
> > > Thank you for your input and for helping us strengthen the paper.

---

### Author Response · Authors · 2024-11-26

We thank the reviewers for taking the time to read our paper and provide valuable feedback. We believe that their input has significantly improved our work, both in terms of clarity and experimental results. For ease of readability, we summarize below the main discussion points raised by the reviewers and the key revisions made to the paper, which are highlighted in blue in the updated manuscript.

-------------

**Motivation/Scope of the paper**

**We propose a generative modeling framework in Euclidean space with dynamics analogous to those on a Lie group manifold. Our approach enables unconstrained dynamics (i.e., it is not limited to being equivariant or invariant under the group $G$ while preserving knowledge of the group action on the data space.**

The main advantage of our framework with respect to Lie group diffusion can be summarize as follows:
- *Data transformation*: Extracting curved coordinates requires pre-processing, which is labor-intensive and generally difficult.
- *Diffusion dynamics is complicated*: Diffusion on curved manifolds is more complex due to curvature-related challenges. While promising, this field is far less developed compared to Euclidean diffusion, where extensive theoretical and computational tools are available.
 - *Projection requirement*: To constrain diffusion to a manifold $X$, projection (e.g., the exponential map for Lie groups) is needed, which is computationally expensive.   Our dymanics is completely in Euclidean space and does not required projections.
- *Need of sampling*. For general curved space there is no exact solution to the SDE. That means that the forward solution (also in training) must also be simulated in order to obtain $p(x(t)|x(0))$. We derived an exact solution for **any** group $G$ (equation 8).
- *It works for any Lie group, both Abelian and not-Abelian*. This is the first methods are are aware of that realizes simulation-free training of manifold-like diffusion models, and the first result of denoising score-matching result for general non-Abelian groups (unlike [1] and [2]). To demonstrate this point, we included an experiment with $G=SO(4)$.

[1] De Bortoli et al, "Riemannian score-based generative modelling"

[2] Huang, Chin-Wei, et al. "Riemannian diffusion models."


Some reviewers pointed out that the assumption about the curvature of the manifold $X$ was not consistently emphasized throughout the work. As clarified above, our primary goal/motivation is to achieve (curved) Lie group dynamics within Euclidean space, thus achieving the features of Riemannian/Lie group diffusion while maintaining the advantages of Euclidean space. However, in the process, we derived sufficient conditions for generalized score matching, which also hold when $X$ is curved. To clarify:
- The conditions presented in Section 2.2, "Sufficient Conditions for Lie Group-Induced Generalized Score Matching", are valid for a curved $X$ and they are a novel and nontrivial result on its own;
- The main results, including Theorem 3.1, as well as the focus of the rest of the paper, assume $X$ is flat/Euclidean.

We have updated the introduction and relevant sections to make this distinction explicit, ensuring there is no further confusion.

---

> ### Author Response · Authors · 2024-11-26
>
> --------------------
>
> **Additional Experiments**
>
>
> - We **compared our method on the rotated MNIST dataset against the Brownian Bridge Diffusion Model (BBDM) [1]**, as traditional diffusion models are unsuitable for this task, which involves learning a map between two non-trivial distributions. In some cases, BBDM fails to reconstruct a rotated MNIST digit correctly, as shown in Figure 6a (e.g., digits 2, 4, 5, 6, and 7). Instead, it adds or removes pixels, and in some instances, transitions from one digit to another (e.g., from 9 to 4, as seen in the last row of Figure 6b in the updated manuscript). This artifact likely arises from BBDM’s training process, which pins two endpoints while allowing the (latent) representation to change in the full 784-dimensional space.
>     *Our model can effectlively learn just the one-dimensional SO(2) transformations while remaining in pixel Euclidean space.*
>
>     [1] BBDM: Image-to-image Translation with Brownian Bridge Diffusion Models. Bo Li, Kaitao Xue, Bin Liu, Yu-Kun Lai, CVPR 2023
>
> - We performed a **comparison between our proposed method and diffusion on Lie groups "Riemannian score-based generative modelling" (RSGM) [2] in the CrossDocked experiment.** We re-implemented the strategy of [2,3,4] for modeling the $T(3)\times SO(3)$ dynamics corresponding to global transformation of the ligand. We compared the Root mean square deviation (RMSD) for all the 100 molecules in the test dataset from the two approaches. For each molecule we generated 5 poses and we compare the RMSD with respect to the ground-truth docked ligand. The results are presented in the updated Figure 8. In summary, *our framework achieves a better RMSD, namely in average RMSD=$2.9\pm 1.0$ Å vs RMSD=$5.6\pm 1.2$ Å  for RSGM.*
>
>    [2] De Bortoli et al, ”Riemannian score-based generative modelling”
>
>    [3] Huang, Chin-Wei, et al. ”Riemannian diffusion models.”
>
>    [4] Corso, Gabriele, et al. "Diffdock: Diffusion steps, twists, and turns for molecular docking."
>
> - We added an **$G=SO(4)$ experiment to demonstrate that our method works with any Lie group of any dimensions**, beyond the standard SO(2) and SO(3), without the need of sampling during training and/or performing implicit score matching. The theoretical results are presented in the newly added Appendix A.5, A.6, and the experimental result can be found in the updated Figure 5.

---

### Comment · Area_Chair_ANQP · 2024-11-29
**Discussion reminder**

Dear reviewers,
If you haven't already done so, could you please engage in discussion with the authors? Thanks a lot!
-AC

---

### Meta-Review · Area_Chair_ANQP · 2024-12-19

**Metareview:**

This paper proposes a new method for score-based generative modeling of data from Lie groups. Given a Lie group, the interesting idea is to consider the action of that group on an appropriately chosen vector space, and conduct generative modeling in that flat space instead. The majority of reviewers are experts in related areas but their opinions were bimodal. Although the assessments remained divergent till the end, upon carefully reading the original and rebuttal versions, reviewer-author interactions, and additional discussions, it seems there are still significant concerns about whether empirical results sufficiently support the strong claims, the presentation, and whether existing approaches are adequately discussed and experimentally compared against. I do appreciate the efforts of the authors as well as the hard work of the reviewers, and I feel that incorporating the post rebuttal discussions into a revision would make the paper much stronger. Recognizing the potential of the idea beyond the scores, I encourage the authors to submit again.

**Additional Comments On Reviewer Discussion:**

The authors and reviewers had extensive discussions, but two of the expert reviewers (75NJ, 3xv7) still felt their concerns (e.g., empirical results may not sufficiently support the strong claims, and unclear presentation) had not been fully resolved. Reviewer T87W on the other hand gave a high score but the arguments seem slightly more superficial than other reviews.

---

### Decision · Program_Chairs · 2025-01-22

Reject